# iHyperTime: Interpretable Time Series Generation with Implicit Neural Representations

## Abstract

Implicit neural representations (INRs) have emerged as a powerful tool that provides an accurate and resolution-independent encoding of data. Their robustness as general approximators has been shown across diverse data modalities, such as images, video, audio, and 3D scenes. However, little attention has been given to leveraging these architectures for time series data. Addressing this gap, we propose an approach for time series generation based on two novel architectures: TSNet, an INR network for interpretable trend-seasonality time series representation, and iHyperTime, a hypernetwork architecture that leverages TSNet for time series generalization and synthesis. Through evaluations of *fidelity* and *usefulness* metrics, we demonstrate that iHyperTime outperforms current state-of-the-art methods in challenging scenarios that involve long or irregularly sampled time series, while performing on par on regularly sampled data. Furthermore, we showcase iHyperTime fast training speed, comparable to the fastest existing methods for short sequences and significantly superior for longer ones. Finally, we empirically validate the quality of the model's unsupervised trend-seasonality decomposition by comparing against the well-established STL method.

Code available at: `https://anonymous.4open.science/r/iHyperTime-8186/README.md`

## 1 Introduction

Modeling time series data has been a key topic of research for many years, constituting a crucial component in a wide variety of areas such as climate modeling, medicine, biology, retail and finance (Lim & Zohren, 2021). Traditional methods for time series modeling have relied on parametric models informed by expert knowledge. However, the development of modern machine learning methods has provided purely data-driven techniques to learn temporal relationships. In particular, neural network-based methods have gained popularity in recent times, with applications to a wide range of tasks, such as time series classification (Ismail Fawaz et al., 2020), clustering (Alqahtani et al., 2021), segmentation (Zeng et al., 2022), anomaly detection (Choi et al., 2021), upsampling (Oh et al., 2020), imputation (Cao et al., 2018), forecasting (Torres et al., 2021) and generation (Coletta et al., 2023). In particular, generation of synthetic time series has recently gained attention due to the large number of applications in medical and financial fields, where data cannot be shared, either due to privacy reasons or proprietary restrictions (Jordon et al., 2021; Assefa et al., 2020). Moreover, synthetic time series can be used to augment training datasets to improve model generalization on downstream tasks, such as classification (Fons et al., 2021), forecasting and anomaly detection. In these fields, having a disentangled representation of time series can be critical for applications with regulatory focus, which often require transparency and interpretability of proposed machine learning solutions as well as injection of expert knowledge as constraints into the training process (Vyetrenko & Xu, 2019).

The task of generating realistic time-series data poses considerable challenges, particularly due to the diverse nature of time series, which may vary along dimensions such as univariate vs. multivariate, short vs. long, and regularly vs. irregularly sampled. Although numerous solutions have been proposed to address the problem of time-series data generation (Alaa et al., 2021; Yoon et al., 2019; Esteban et al., 2017), the focus has predominantly been on generating data for well-structured scenarios, such as short and regularly sampled time series. Such an approach is often conflicting with the complexities of real-world time-series data, where irregularities, missing values and diverse

sequence lengths are commonplace (Fang & Wang, 2020). Recent methods have addressed some of these shortcomings (Jeon et al., 2022; Zhou et al., 2023; Coletta et al., 2023). However, no single approach has shown a consistent performance across all these challenging scenarios. Furthermore, the scalability of many existing methods to longer sequences is constrained by escalating computational costs that correlate with series length.

In recent years, implicit neural representations (INRs) have gained popularity as an accurate and flexible method to parameterize signals from diverse sources, such as images, video, audio and 3D scene data (Sitzmann et al., 2020b; Mildenhall et al., 2020). Conventional methods for data encoding often rely on discrete representations, such as data grids, which are limited by their spatial resolution and present inherent discretization artifacts. In contrast, INRs encode data in terms of continuous functional relationships between signals, and thus are uncoupled to spatial resolution. In practical terms, INRs provide a data representation framework that is resolution-independent, which makes them ideally suited to deal with the aforementioned challenges in real-world time series. While there have been a few recent works exploring the application of INRs to time series data (Jeong & Shin, 2022; Woo et al., 2023), there is no work on leveraging these architectures for generating synthetic time series.

In this paper, we propose a novel method for time series generation based on two novel architectures: 1) TSNet, an INR tailored for resolution-agnostic encoding of time series data, offering a trend-seasonality-residual disentangling of single time series. 2) iHyperTime, a hypernetwork architecture for generalization of time series datasets, that leverages TSNet to produce interpretable latent representations of the signals. Together, these architectures form a unified approach for disentangled representation and generation of multiple forms of time series data, including challenging cases such as multivariate, irregularly sampled, and long time series.

**Generation quality** Through empirical evaluations, we demonstrate that iHyperTime outperforms existing state-of-the-art methods for time series generation. Our method excels in complex scenarios such as irregularly sampled or long sequences, while performing on par with state-of-the-art for regularly sampled time series.

**Efficiency** We show that our architecture achieves rapid training speeds, comparable to the fastest methods for short sequences, and substantially faster for longer ones.

**TSR Decomposition** We validate the unsupervised decomposition capabilities of our method by benchmarking it against the widely-used STL decomposition technique.

## 2 RELATED WORK

**Implicit Neural Representations** Implicit Neural Representations (INRs) provide a continuous representation of multidimensional data, by encoding a functional relationship between input co-ordinates and signal values, avoiding possible discretization artifacts. They have recently gained popularity in visual computing (Mescheder et al., 2019; Mildenhall et al., 2020) due to the key development of positional encodings (Tancik et al., 2020) and SIREN periodic activations (Sitzmann et al., 2020b), which have proven to be critical for the learning of high-frequency details. Whilst INRs have been shown to produce accurate reconstructions in a wide variety of data sources, such as video, images and audio (Sitzmann et al., 2020b; Chen et al., 2021; Rott Shaham et al., 2021), few works have leveraged them for time series representation (Jeong & Shin, 2022; Woo et al., 2023), and none have focused on interpretability and generation.

**Hypernetworks** Hypernetworks are neural network architectures that are trained to predict the parameters of secondary networks, referred to as hyponetworks (Ha et al., 2017; Sitzmann et al., 2020a). In the last few years, some works have leveraged different hypernetwork architectures for the prediction of INR weights, in order to learn priors over image data (Skorokhodov et al., 2021) and 3D scene data (Littwin & Wolf, 2019; Sitzmann et al., 2019; Sztrajman et al., 2021). Sitzmann et al. (2020b) leverage a set encoder and a hypernetwork decoder to learn a prior over SIRENs encoding image data, and apply it for image in-painting.

**Time Series Generation** Synthesis of time series data using deep generative models has been previously studied in the literature. Examples include the TimeGAN architecture (Yoon et al., 2019), GT-GAN (Jeon et al., 2022), and QuantGAN (Wiese et al., 2020). More recently, as an

alternative to GAN-based time series generation, Desai et al. (2021) proposed TimeVAE, based on a variational autoencoder, while Coletta et al. (2023) proposed a diffusion model architecture called DiffTime. Alaa et al. (2021) introduced Fourier Flows, a normalizing flow model for time series data that leverages the frequency domain representation, which is currently considered together with TimeGAN as state-of-the-art for time series generation. In the last few years, multiple methods have used INRs for data generation, with applications on image synthesis (Skorokhodov et al., 2021), super-resolution (Chen et al., 2021) and panorama synthesis (Anokhin et al., 2021). However, there are currently no applications of INRs on the generation of time series data.

**Interpretable Time Series**  Seasonal-trend decomposition is a standard tool in time series analysis. The trend encapsulates the slow time-varying behavior of the signal, while seasonal components capture periodicity. These techniques introduce interpretability in time series, which plays an important role in downstream tasks such as forecasting and anomaly detection. The classic approaches for decomposition are the widely used STL algorithm (Cleveland et al., 1990), and its variants (Wen et al., 2019; Bandara et al., 2022). Relevant to this work is the recent N-BEATS architecture (Oreshkin et al., 2020), a deep learning-based model for univariate time series forecasting that provides interpretability capabilities. The model explicitly encodes seasonal-trend decomposition into the network by defining separate trend and seasonal blocks, which fit a low degree polynomial and a Fourier series.

## 3   FORMULATION

In this Section, we describe the TSNet network architecture for time series representation and TSR decomposition, and the iHyperTime network leveraged for generalization and new data generation.

### 3.1   TIME SERIES REPRESENTATION

We consider a time series signal encoded by a discrete sequence of $N$ observations $\mathbf{y} = (\mathbf{y}_1, ..., \mathbf{y}_N)$ where $\mathbf{y}_i \in \mathbb{R}^m$ is the $m$-dimensional observation at time $t_i$. This time series defines a dataset $\mathcal{D} = \{(t_i, \mathbf{y}_i)\}_{i=1}^{N}$ of time coordinates $t_i$ associated with observations $\mathbf{y}_i$. We want to find a continuous mapping $f : \mathbb{R} \to \mathbb{R}^m, t \to f(t)$ that parameterizes the discrete time series, so that $\mathbf{y}_i = f(t_i)$ for $i = 1, \ldots, N$. The function $f$ can be approximated by an implicit neural representation (INR) architecture conditioned on the training loss $\mathcal{L} = \sum_i \|\mathbf{y}_i - \hat{f}(t_i)\|^2$. Input and output of the INR are of dimensions 1 and $m$, corresponding to the time coordinate $t$ and the prediction $\hat{f}(t)$. After training, the network encodes a continuous representation of the functional relationship $f(t)$ for a single time series.

#### 3.1.1   TSNET

We propose an interpretable architecture to encode time series that reuses the described INR. In particular, we assume that our INR follows a classic time series additive decomposition, i.e.,

$$f(t) = f_T(t) + f_S(t) + f_R(t), \tag{1}$$

where $f_T$, $f_S$, $f_R$ correspond to the trend, seasonality and residual components of $f(t)$. Note that this is a standard assumption for time series decomposition techniques, such as STL and others (Cleveland et al., 1990). We elaborate on our modeling of these three components in the following.

**Trend and Seasonality Blocks**  Following the work by Oreshkin et al. (2020), we model trend and seasonality via basis decompositions with coefficients learned by fully-connected networks. The trend component of a time series aims to model slow-varying (and occasionally monotonic) behavior, thus we consider a polynomial regressor, i.e.,

$$f_T(t) = \sum_{p=0}^{P} \mathbf{w}_p^{(T)} t^p, \tag{2}$$

where $P$ denotes the degree of the polynomial, and $\mathbf{w}_p^{(T)}$ denotes the learned weight associated with the $p$th degree. In practice, $P$ is chosen to be small (e.g., $P = 2$) to capture low frequency behavior.

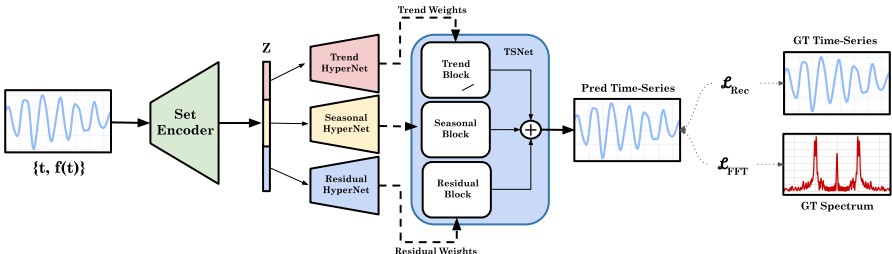

Figure 1: iHyperTime architecture. The Set Encoder processes a set of tuples $\{(t_i, \mathbf{y}_i)\}_{i=1}^N$ representing a time series, and encodes it as embeddings $Z_T$, $Z_S$, $Z_R$ associated to the components of the TSR decomposition. The hypernetwork decoders learn to predict the weights of their corresponding TSNet blocks from the embeddings. During training, the output of the hypernetworks is used to instantiate a TSNet hyponetwork, and the loss is computed as a difference between $\mathbf{y}$ and the output of TSNet $\hat{f}(t)$, in terms of signal and spectral distribution.

The seasonal component of the time series $f_S(t)$ aims to capture the periodic behavior of the signal, and thus we leverage a learnable Fourier decomposition:

$$f_S(t) = \sum_{i=0}^{N/2-1} \left( \mathbf{w}_i^{(S)} \cos\left(2\pi it\right) + \mathbf{w}_{N/2+i}^{(S)} \sin\left(2\pi it\right) \right) \tag{3}$$

where $\mathbf{w}_i^{(S)}$ are the weights predicted by the network.

**Residual Block**  The residual of a time series comprises the high-frequency non-periodic components of the signal. In order to model it, we leverage a fully-connected network of $K$ layers with sine activations (SIREN), as defined by Sitzmann et al. (2020b):

$$\mathbf{q}_{k+1} = \sin\left(\omega_0 \mathbf{w}_k^{(R)} \mathbf{q}_k + \mathbf{b}_k^{(R)}\right), \qquad k = 0, ..., K-1 \tag{4}$$

$$f_R(t) = \mathbf{w}_K^{(R)} \mathbf{q}_K + \mathbf{b}_K^{(R)} \tag{5}$$

where $\mathbf{w}_k^{(R)}$, $\mathbf{b}_k^{(R)}$ and $\mathbf{q}_k$ are the weights, biases, and outputs of the $k$ layer, with $\mathbf{q}_0 = t$ corresponding to the input of the network. A general factor $\omega_0$ multiplying the network weights determines the order of magnitude of the frequencies that will be used to encode the signal. As shown by Sitzmann et al. (2020b), SIRENs mitigate the spectral bias of regular fully-connected networks, and thus are well suited for learning and representation of high-frequencies. We refer to Appendix D.2 for the TSNet model implementation details.

### 3.2 TIME SERIES GENERATION WITH IHYPERTIME

In Fig. 1, we show a diagram of our iHyperTime architecture, which can be used to learn a prior over implicit neural representations (TSNet) of time series data. Next, we will detail its components and describe how iHyperTime can be used for time series generation. Additional details on the model implementation can be found in Appendix D.2.

**Set Encoder**  The set encoder is composed of SIREN layers (Sitzmann et al., 2020b) and takes as input an arbitrary set of tuples $\{(t_i, \mathbf{y}_i)\}_{i=1}^N$, where $t$ denotes the time-coordinate and $\mathbf{y}_i$ the corresponding univariate or multivariate time series value. Each tuple is encoded into a fixed-size embedding $Z_i = g(t_i, \mathbf{y}_i)$, and the sample set is reduced to a single embedding $Z$ by applying a symmetric operation $\oplus$ (e.g., averaging): $Z = \bigoplus_{i=1}^N Z_i$, where the function $g : \mathbb{R} \times \mathbb{R}^m \to \mathbb{R}^{d_Z}$, used to determine the embedding of each tuple, is parameterized by the SIREN layers (model details in Appendix D.2). The use of a set encoder introduces permutation invariance in the computation, and provides a high degree of flexibility in terms of the input (Zaheer et al., 2017), enabling the encoding of data with missing values or irregular sampling, which are common occurrences in time series.

**Hypernetwork decoders**  The embedding $Z$ is modeled as a concatenation of three sub-embeddings $Z_T$, $Z_S$, and $Z_R$ with $Z_T \in \mathbb{R}^{d_T}$ denoting the trend embedding, $Z_S \in \mathbb{R}^{d_S}$ denoting the seasonality embedding, and $Z_R \in \mathbb{R}^{d_R}$ denoting the residual embedding with $d_Z = d_T + d_S + d_R$. Each

embedding component is pass through its own hypernetwork decoder, which outputs the weights of its corresponding block in the TSNet INR. For example, $Z_T$ is passed into the Trend Hypernetwork to output the weights of the trend block in TSNet. The output of TSNet sums the three signals from each block into a single predicted time series, which is compared against the ground truth signal via the reconstruction and spectral losses ($\mathcal{L}_{Rec}$ and $\mathcal{L}_{FFT}$) during training.

**iHyperTime training**   During training, we use the weights predicted by the hypernetwork decoders to instantiate a TSNet hyponetwork and evaluate it on the input time-coordinate $t$, to produce the predicted time series value $\hat{f}(t)$. We then compare the TSNet hyponetwork prediction with the ground truth signal via the reconstruction and spectral losses $\mathcal{L}_{Rec}$ and $\mathcal{L}_{FFT}$.

The training of iHyperTime is performed in three stages, in order to improve stability: 1) we train the Trend networks (Trend hypernetwork, Trend block) for 100 epochs, computing the MSE loss between the ground truth time series $\mathbf{y}$ and the output of the block: $\mathcal{L}_1 = \sum_i \|\mathbf{y}_i - \hat{f}_T(t_i)\|^2$. This leads to a smooth approximation of the time series, which we use as initial guess for the second stage. 2) We then train the Trend and Seasonality blocks together, computing the MSE reconstruction loss $\mathcal{L}_{rec}$ and the FFT loss $\mathcal{L}_{FFT}$ between the ground truth and the added output of both TSNet blocks. 3) Finally, we train the three blocks together.

**Training Loss**   The training of iHyperTime is done by optimizing the following loss, which contains an MSE reconstruction term $\mathcal{L}_{rec}$, a spectral loss $\mathcal{L}_{FFT}$ and two regularization terms $\mathcal{L}_{weights}$ and $\mathcal{L}_{latent}$, for the network weights and the latent embeddings, respectively:

$$\mathcal{L} = \underbrace{\frac{1}{N}\sum_{i=1}^{N}\left\|\mathbf{y}_i - \hat{f}(t_i)\right\|^2}_{\mathcal{L}_{rec}} + \lambda_1 \underbrace{\frac{1}{W}\sum_{j=1}^{W} w_j^2}_{\mathcal{L}_{weights}} + \lambda_2 \underbrace{\frac{1}{Z}\sum_{l=1}^{Z} z_l^2}_{\mathcal{L}_{latent}} + \lambda_3 \underbrace{\frac{1}{N}\sum_{k=0}^{N-1}\left\|F_k - \hat{F}_k\right\|}_{\mathcal{L}_{FFT}}. \tag{6}$$

where $F_k = [\mathcal{F}_T\{\mathbf{f}\}]_k$ corresponds to the coefficient of the $k$th frequency in the discrete Fourier transform (DFT) of the time series. The term $\mathcal{L}_{FFT}$ penalizes deviations of the signal's frequency spectrum with respect to ground truth. Thus, we ensure a high-fidelity reconstruction not only of the time series values, but also of its spectral composition. We refer to Appendices C and D for further details on $\mathcal{L}_{FFT}$ and the implementation details of the iHyperTime architecture.

**Time Series Generation**   After training, we leverage the hypernetwork architecture to generate latent representations of the time series from our training set. Generation of new time series is produced by randomly selecting pairs of time series, and performing linear a interpolation between their embeddings $Z^{(1)}$ and $Z^{(2)}$:

$$Z^{gen} = Z^{(1)} + \lambda\left(Z^{(2)} - Z^{(1)}\right) \tag{7}$$

where $\lambda$ is also sampled randomly. Optionally, the interpolation can be performed on individual components $Z_T$, $Z_S$, $Z_R$ of the embeddings, enabling the conditional generation of time series.

## 4 EXPERIMENTS

We present our evaluation of time series generation on regular and irregular data, covering time series of diverse lengths and numbers of channels. Additionally, we perform an analysis of our model's TSR decomposition, and we compare training and inference times with previous works.

### 4.1 BASELINES AND EVALUATION

**Datasets**   We test the performance of iHT using multiple datasets with varying characteristics such as periodicity, level of noise, number of features and length of the series. Stock corresponds to Google stock price data from 2004 to 2019, where each observation has 6 features. Energy is a UCI appliance prediction dataset (Candanedo et al., 2017) with 28 features. Additionally, we also consider Monash dataset (Godahewa et al., 2021), from which we choose FRED-MD, NN5 Daily, Temperature Rain, and Solar Weekly datasets. A complete description of the datasets can be found in Appendix B.

Additionally, we conduct experiments on irregularly sampled time series, achieved by randomly removing fixed percentages of values from each time series. We create the datasets by removing 30, 50 and 70% of each time series.

**Baselines**   We compare our method with TimeGAN (Yoon et al., 2019), GT-GAN (Jeon et al., 2022), Fourier Flows (FF) (Alaa et al., 2021), LS4 (Zhou et al., 2023), DiffTime (Coletta et al., 2023), and RCGAN (Esteban et al., 2017). TimeGAN and GT-GAN have shown strong performance on multivariate time series with short sequence lengths and are able to handle missing data. LS4, DiffTime, and Fourier Flows have shown strong performance on time series with longer sequence length, generating distributions of frequencies that closely resemble the original data. We refer to Appendix D.1 for further details on baselines and the adjusted DiffTime and RCGAN architectures, introduced to deal with missing data and longer time series, respectively.

**Evaluation metrics**   To asses the quality of the synthesized data, we adopt the predictive and discriminative scores used in TimeGAN (Yoon et al., 2019). The **predictive score** measures the *usefulness* of the generated data by using a *train on synthetic, test on real* (TSRT) approach: a model is trained using the synthetic data to predict the next step in a sequence, and then it is evaluated using the real data. The mean absolute error (MAE) between the predicted values and the ground truth is used for the evaluation. The **discriminative score** serves as a measurement of *fidelity* of the generated data, where the aim is to assess if the synthetic data is indistinguishable from real data. For this purpose, a discriminative model is trained to classify real and fake samples, and then used to test whether the original and generated data are correctly classified. The discriminative score is computed as $|\text{Accuracy} - 0.5|$, where a low value means that the classification is challenging, and therefore, the model cannot tell which samples are real and which are generated. For a **qualitative evaluation**, we analyze the synthesized and original time series by employing t-SNE visualizations which project the data into a two-dimensional space (van der Maaten & Hinton, 2008). Additionally, we perform a kernel density estimation (Jeon et al., 2022) to compare the data distributions. For the experiments on longer real-world time series from the Monash dataset, we also consider the **marginal score** (Zhou et al., 2023) that computes the absolute difference between the real and synthetic empirical probability density functions.

## 4.2 Experimental results on regular time series synthesis

Performance results for regularly sampled time series are provided in Table 1, for univariate and multivariate datasets of varying lengths. The results indicate that iHT outperforms all methods in terms of the predictive score. This highlights the usefulness of the data generated by our method as a source of synthetic data for learning. In terms of discriminative score, iHT is competitive with state-of-the-art methods across all datasets, although no method emerges as a definitively superior approach. In Figure 2, the t-SNE visualizations show that the time series generated by iHT closely resemble the ground truth data distribution. We refer to Appendix I for additional qualitative comparisons.

Table 1: Regular time series generation performance in terms of predictive and discriminative scores.

| | Method | Energy24 | Stock24 | Stocks72 | Stock360 |
|---|---|---|---|---|---|
| | iHT | **.251 ± .000** | **.037 ± .000** | **.188 ± .000** | **.168 ± .000** |
| Predictive Score | GT-GAN | .321 ± .002 | .040 ± .000 | .207 ± .000 | .188 ± .000 |
| | TimeGAN | .273 ± .004 | .038 ± .001 | .226 ± .002 | .206 ± .000 |
| | RCGAN | .292 ± .005 | .040 ± .001 | .192 ± .001 | .189 ± .000 |
| | DiffTime | .252 ± .000 | .038 ± .001 | .213 ± .000 | .215 ± .000 |
| | LS4 | .295 ± .001 | .103 ± .001 | .194 ± .000 | **.168 ± .000** |
| | FF | **.251 ± .000** | .076 ± .001 | .191 ± .000 | .169 ± .000 |
| | Original | .250 ± .003 | .036 ± .001 | .186 ± .001 | .167 ± .001 |
| | iHT | .245 ± .019 | **.044 ± .011** | .014 ± .009 | .018 ± .015 |
| Discriminative Score | GT-GAN | **.221 ± .068** | .077 ± .031 | .058 ± .017 | .085 ± .064 |
| | TimeGAN | .236 ± .012 | .102 ± .021 | .073 ± .047 | .042 ± .074 |
| | RCGAN | .336 ± .017 | .196 ± .027 | **.012 ± .09** | **.014 ± .007** |
| | DiffTime | .445 ± .004 | .097 ± .016 | .097 ± .012 | .101 ± .018 |
| | LS4 | .499 ± .000 | .363 ± .027 | .089 ± .081 | .088 ± .081 |
| | FF | .499 ± .001 | .349 ± .113 | .016 ± .018 | .015 ± .014 |

Table 2: Irregular time series generation performance: predictive and discriminative scores. 30% missing data.

| | Method | Energy24 | Stock24 | Stocks72 | Stock360 |
|---|---|---|---|---|---|
| | iHT | **.049 ± .001** | **.013 ± .001** | **.188 ± .000** | **.168 ± .000** |
| Pred Score | GT-GAN | .066 ± .001 | .021 ± .003 | .206 ± .000 | .196 ± .000 |
| | DiffTime | .052 ± .001 | .019 ± .006 | .200 ± .000 | .188 ± .000 |
| | LS4 | .063 ± .001 | .022 ± .005 | .198 ± .000 | .229 ± .000 |
| | FF | .148 ± .007 | .137 ± .029 | .210 ± .000 | .184 ± .000 |
| | Original | .045 ± .001 | .011 ± .002 | .186 ± .001 | .167 ± .001 |
| | iHT | .452 ± .003 | **.059 ± .046** | **.017 ± .007** | **.014 ± .010** |
| Disc Score | GT-GAN | .333 ± .063 | .251 ± .097 | .068 ± .007 | .111 ± .026 |
| | DiffTime | **.298 ± .010** | .215 ± .010 | .110 ± .045 | .057 ± .070 |
| | LS4 | .500 ± .000 | .495 ± .004 | .203 ± .028 | .067 ± .016 |
| | FF | .500 ± .000 | .497 ± .005 | .223 ± .092 | .156 ± .102 |

## 4.3 Experimental results on irregular time series synthesis

Tables 2, 3 and 4 shows the results for the irregular time series generation with different percentages of missing values. iHT outperforms all methods in terms of predictive score across all datasets. In regards to *fidelity* (predictive score), our method shows the best performance in 3 out of 4 datasets. In particular, it shows the best scores for long time series datasets, showing its versatility on time

Table 3: Irregular time series generation performance: predictive and discriminative scores. 50% missing data.

| | Method | Energy24 | Stock24 | Stocks72 | Stock360 |
|---|---|---|---|---|---|
| **Pred Score** | **iHT (Ours)** | **.051 ± .002** | **.014 ± .001** | **.187 ± .000** | **.168 ± .000** |
| | GT-GAN | .064 ± .001 | .018 ± .002 | .195 ± .000 | .195 ± .000 |
| | DiffTime | .057 ± .001 | .024 ± .002 | .278 ± .000 | .186 ± .000 |
| | LS4 | .065 ± .002 | .033 ± .005 | .212 ± .000 | .197 ± .000 |
| | FF | .227 ± .004 | .169 ± .018 | .236 ± .000 | .215 ± .000 |
| | Original | .045 ± .001 | .011 ± .002 | .186 ± .001 | .167 ± .001 |
| **Disc Score** | **iHT (Ours)** | .472 ± .004 | **.102 ± .051** | **.011 ± .003** | **.004 ± .003** |
| | GT-GAN | **.317 ± .010** | .265 ± .073 | .026 ± .012 | .081 ± .023 |
| | DiffTime | .422 ± .011 | .332 ± .034 | .284 ± .137 | .110 ± .061 |
| | LS4 | .500 ± .000 | .498 ± .000 | .144 ± .034 | .027 ± .015 |
| | FF | .500 ± .002 | .498 ± .003 | .376 ± .130 | .422 ± .058 |

Table 4: Irregular time series generation performance: predictive and discriminative scores. 70% missing data.

| | Method | Energy24 | Stock24 | Stocks72 | Stock360 |
|---|---|---|---|---|---|
| **Pred Score** | **iHT (Ours)** | **.053 ± .000** | **.014 ± .013** | **.187 ± .000** | **.168 ± .000** |
| | GT-GAN | .076 ± .001 | .020 ± .005 | .205 ± .000 | .196 ± .000 |
| | LS4 | .084 ± .003 | .024 ± .002 | .188 ± .000 | .188 ± .000 |
| | DiffTime | .065 ± .001 | .068 ± .063 | .284 ± .000 | .196 ± .000 |
| | FF | .304 ± .005 | .205 ± .001 | .267 ± .000 | .245 ± .000 |
| | Original | .045 ± .001 | .011 ± .002 | .186 ± .001 | .167 ± .001 |
| **Disc Score** | **iHT (Ours)** | .482 ± .003 | **.115 ± .052** | **.020 ± .019** | **.011 ± .012** |
| | GT-GAN | **.325 ± .047** | .230 ± .053 | .058 ± .002 | .091± .013 |
| | LS4 | .499 ± .002 | .455 ± .011 | .036 ± .026 | .183 ± .017 |
| | DiffTime | .444 ± .001 | .421 ± .003 | .436 ± .009 | .148 ± .018 |
| | FF | .500 ± .003 | .498 ± .008 | .424 ± .083 | .369 ± .163 |

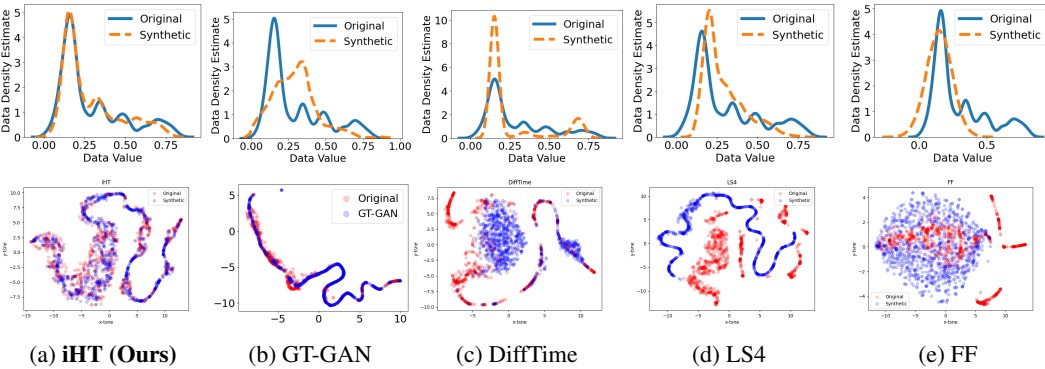

(a) **iHT (Ours)**    (b) GT-GAN    (c) TimeGAN    (d) RCGAN    (e) DiffTime    (f) LS4    (g) FF

Figure 2: t-SNE visualizations on Stock24 data, where a greater overlap of blue and red dots shows a better distributional-similarity between the original and generated data. Our approach shows the best performance. (See Appendix 21 for high resolution charts)

series beyond 24 time steps. Furthermore, the performance of iHT does not degrade significantly with the percentage of missing values, even in the extreme case of 70% missing data. The top row in Figure 3 compares the distributions of original and synthetic data for the Stock24 dataset with 50% of missing values. iHT shows the best performance, with the closest match to the original data distribution. The bottom row shows the corresponding t-SNE visualizations, where we can see that iHT and GT-GAN show the best overlap between original and generated data, with iHT showing more data diversity, covering a wider area across the original data. Additional plots for other missing rates are shown in Appendix I, where we observe a similar behavior with iHT showing the best overlap in both distribution and t-SNE visualization.

(a) **iHT (Ours)**    (b) GT-GAN    (c) DiffTime    (d) LS4    (e) FF

Figure 3: (Top) Data distribution on irregular Stock24 data (Missing 50%). (Bottom) t-SNE visualizations on irregular Stock24 data (Missing 50%), where a greater overlap of blue and red dots shows a better distributional-similarity between the generated data and original data. Our approach shows the best performance.

## 4.4 EXPERIMENTAL RESULTS ON LONGER REAL-WORLD TIME SERIES SYNTHESIS

Table 5 shows additional generation results for 4 real-world datasets from the Monash dataset, where 3 of the datasets contain time series with lengths over 700 time steps. We report comparisons with LS4, which has shown state-of-art performance on these datasets (Zhou et al., 2023), while we leave the full evaluation table in Appendix E. In this scenario, the Classification (*fidelity*) and Prediction

(*usefulness*) scores are computed using a 1-layer S4 model (Zhou et al., 2023). The results show the ability of iHT to deal with long time series, with superior performance in 3 out of 4 datasets w.r.t. the state-of-art LS4.

Table 5: Generation results on Monash datasets.

| Data | Metric | LS4 | iHT (Ours) | Data | Metric | LS4 | iHT (Ours) |
|---|---|---|---|---|---|---|---|
| FRED-MD | Marginal ↓ | 0.0221 | **0.0177** | Temp Rain | Marginal ↓ | **0.0834** | 0.2978 |
| | Class ↑ | 0.544 | **1.3278** | | Class ↑ | 0.976 | **11.2493** |
| | Prediction ↓ | 0.0373 | **0.0181** | | Prediction ↓ | 0.521 | **0.132** |
| NN5 Daily | Marginal ↓ | **0.00671** | 0.00893 | Solar Weekly | Marginal ↓ | 0.0459 | **0.03273** |
| | Class ↑ | **0.636** | 0.4982 | | Class ↑ | 0.683 | **1.2413** |
| | Prediction ↓ | 0.241 | **0.2349** | | Prediction ↓ | 0.141 | **0.0739** |

## 4.5 RUNTIME

In Figure 4, we show that the strong performance of iHT on long time series does not impact its computational time. We consider a set of synthetic datasets with lengths {80, 320, 1280, 5120, 20480} and we evaluate the training time for 100 iterations, and the inference time on one batch (Zhou et al., 2023). The figure shows that iHT has among the lowest computational times w.r.t. existing approaches. Moreover, iHT training times are almost unaffected by the length of the time series, with negligible changes even for 20,480 time steps, making it the fastest method for long sequences. Additional details and the overall training times are presented in Appendix F.

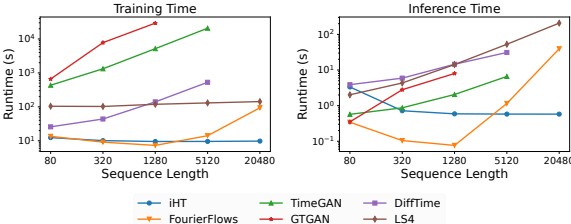

Figure 4: Training and inference time comparison for time series of different lengths.

## 4.6 TREND-SEASONALITY DECOMPOSITION ANALYSIS

iHT provides a controllable method for time series generation based on an interpretable trend-seasonality-residual decomposition of latent embeddings. We explore the interpretable decomposition of iHT on analytically generated time series datasets that have trend (T), seasonal (S) and noise (R) components, or a combination of two of them. The synthetic datasets are generated by uniformly sampling trend values, frequencies and levels of noise. We compute the decomposition error as the MSE between the output of each block of iHT and the corresponding TSR analytic component. In table 6, we compare against the traditional STL method, which requires the period of seasonality as an additional parameter. In *STL (exact)*, we compute STL with the exact period of the analytic signal. In *STL (approx)*, we provide STL with an approximate period estimated by analyzing the Fourier spectrum of the signal, a more realistic setting for time series decomposition. Our method shows the lowest trend error, with a small standard deviation with respect to *STL (approx)*, and shows comparable results with *STL (exact)*. Additionally, our method shows similar performance on the seasonality component when there is seasonality present in the dataset with respect to *STL (approx)*, and shows much better agreement when there is no seasonality present (T+R dataset).

In Figure 5 (a) and (c), we show the distribution of the trend, and residuals outputs from iHT against the ground truth components for the T+S+R dataset. In the case of the seasonality component (b), we plot the histogram for the *frequencies* of the time series, where we estimate the frequency of each time series by finding the dominant frequency component in the discrete Fourier transform. The trend and seasonality histograms show very good agreement with the ground truth, while the residuals show slightly wider tails. Finally, we visualize the learned representations via t-SNE of the embeddings. Figure 5 (d) and (e) shows that iHT is able to learn the trend and seasonal patterns

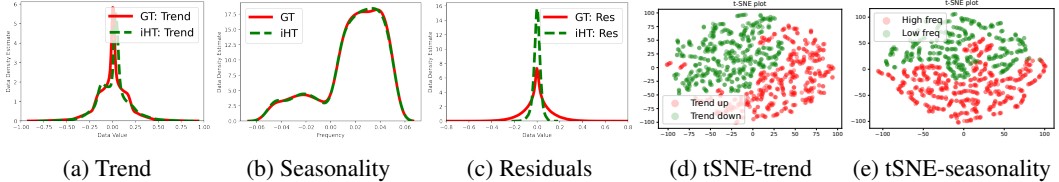

| (a) Trend | (b) Seasonality | (c) Residuals | (d) tSNE-trend | (e) tSNE-seasonality |

Figure 5: (a,b,c): Distribution of the trend, seasonality and residual outputs of iHT vs ground truth. (d,e): t-SNE visualization of trend and seasonality embeddings in iHT. All cases correspond to the T+S+R dataset.

Table 6: Ablation study for trend-seasonality-residual decomposition of time series. We compare iHT against STL decomposition, for three dataset configurations: trend+seasonality (T+S), trend+residual (T+R) and all three components (T+S+R).

| MSE ($\times 10^{-3}$) | T+S+R | | | T+R | | | T+S | | |
|---|---|---|---|---|---|---|---|---|---|
| | STL (exact) | STL (approx) | iHT (Ours) | STL (exact) | STL (approx) | iHT (Ours) | STL (exact) | STL (approx) | iHT (Ours) |
| Trend | $0.46 \pm 2.08$ | $1.04 \pm 4.69$ | $0.60 \pm 0.50$ | $0.07 \pm 0.07$ | $1.26 \pm 5.70$ | $0.20 \pm 0.27$ | $0.46 \pm 2.10$ | $1.17 \pm 5.12$ | $0.46 \pm 0.48$ |
| Seasonality | $0.57 \pm 2.10$ | $1.19 \pm 4.73$ | $1.21 \pm 0.68$ | $0.03 \pm 0.02$ | $1.35 \pm 5.74$ | $0.17 \pm 0.29$ | $0.44 \pm 2.09$ | $1.18 \pm 5.11$ | $1.25 \pm 0.77$ |
| Residuals | $0.17 \pm 0.15$ | $0.18 \pm 0.12$ | $1.29 \pm 0.62$ | $0.15 \pm 0.09$ | $0.18 \pm 0.13$ | $0.41 \pm 0.26$ | $0.03 \pm 0.10$ | $0.04 \pm 0.07$ | $1.25 \pm 0.71$ |

from the dataset. In plot (d) the color separation corresponds to positive and negative trend, while plot (e) shows the separation in frequencies. We refer to Appendix G and H for further details on the synthetic datasets and for additional results.

### 4.7 ABLATION STUDIES

In Table 7, we change the architecture of iHT to create simpler ablation models, and report predictive and discriminative metrics for multiple datasets. **iHT** corresponds to our full proposed model. In **iHT (no FFT)** we have removed the FFT loss from the training. In **iHT-SIREN** we remove the TSR decomposition from iHT, replacing TSNet with a SIREN network. We observe that the predictive scores are comparable for all configurations, while discriminative scores improve for all datasets when we incorporate the interpretable decomposition. The error reduces further when we add the FFT loss to the training process.

Table 7: Ablation study for model architecture: comparison of iHT against simpler configurations: *iHT (no FFT)* without FFT loss, and *iHT-SIREN* without TSR decomposition.

| | Energy24 | Stock24 | Stock72 | Stock360 | | Energy24 | Stock24 | Stock72 | Stock360 |
|---|---|---|---|---|---|---|---|---|---|
| *Predictive Score* | | | | | *Disc. Score* | | | | |
| **iHT** | 0.047 | 0.013 | 0.188 | 0.168 | **iHT** | 0.245 | 0.044 | 0.014 | 0.009 |
| **iHT (no FFT)** | 0.046 | 0.014 | 0.188 | 0.168 | **iHT (no FFT)** | 0.278 | 0.073 | 0.015 | 0.011 |
| **iHT-SIREN** | 0.048 | 0.013 | 0.188 | 0.169 | **iHT-SIREN** | 0.341 | 0.108 | 0.022 | 0.024 |

## 5 DISCUSSION AND CONCLUSIONS

We presented iHyperTime, a versatile and efficient framework for generating time series with a wide range of characteristics. Unlike existing generative models that excel either in short or long sequences, our model demonstrates superior performance across both types of datasets. Our evaluations reveal its efficacy in handling irregularly sampled data, where it consistently surpasses current benchmarks. For regularly sampled sequences, iHyperTime's performance is competitive with the best available models, regardless of time series length. One of the model's notable strengths is its rapid training speed, which is not only comparable to the quickest existing methods for short sequences but also significantly faster for longer ones. Importantly, our architecture incorporates inductive biases that facilitate unsupervised decomposition of time series into trend, seasonality, and residual components, a capability we validated against the established STL decomposition method. Additionally, we illustrated iHyperTime's ability to learn semantically meaningful representations, opening the door for applications that involve generating time series conditioned on interpretable factors.

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

# A  ADDITIONAL RELATED WORK

**Implicit Neural Representations**    INRs (or coordinate-based neural networks) have recently gained popularity in computer vision applications. The usual implementation of INRs consists of a fully-connected neural network (MLP) that maps coordinates (*e.g.* xyz-coordinates) to the corresponding values of the data, essentially encoding their functional relationship in the network. One of the main advantages of this approach for data representation, is that the information is encoded in a continuous/grid-free representation, that provides a built-in non-linear interpolation of the data. This avoids the usual artifacts that arise from discretization, and has been shown to combine flexible and accurate data representation with high memory efficiency (Sitzmann et al., 2020b; Tancik et al., 2020). Whilst INRs have been shown to work on data from diverse sources, such as video, images and audio (Sitzmann et al., 2020b; Chen et al., 2021; Rott Shaham et al., 2021), their recent popularity has been motivated by multiple applications in the representation of 3D scene data, such as 3D geometry (Park et al., 2019; Mescheder et al., 2019; Sitzmann et al., 2020a; 2019) and object appearance (Mildenhall et al., 2020; Sztrajman et al., 2021). In early architectures, INRs showed a lack of accuracy in the encoding of high-frequency details of signals. Mildenhall et al. (2020) proposed positional encodings to address this issue, and Tancik et al. (2020) further explored them, showing that by using Fourier-based features in the input layer, the network is able to learn the full spectrum of frequencies from data. Concurrently, Sitzmann et al. (2020b) tackled the encoding of high-frequency data by proposing the use of sinusoidal activation functions (SIREN: Sinusoidal Representation Networks), and **?** showed the equivalence between Fourier features and single-layer SIRENs. Our INR architecture for time series data (Section 3) is based on the SIREN architecture by Sitzmann *et al*.

**Hypernetworks**    A hypernetwork is a neural network architecture designed to predict the weight values of a secondary neural network, denominated a hyponetwork (Sitzmann et al., 2020a). The concept of hypernetwork was formalized by Ha et al. (2017), drawing inspiration from methods in evolutionary computing (Stanley et al., 2009). Moreover, while convolutional encoders have been likened to the function of the human visual system (Skorokhodov et al., 2021), the analogy cannot be extended to convolutional decoders, and some researchers have argued that hypernetworks much more closely match the behavior of the prefrontal cortex (Russin et al., 2020). Hypernetworks have been praised for their expressivity, compression due to weight sharing, and for their fast inference times(Skorokhodov et al., 2021). They have been leveraged for multiple applications, including few-shot learning (Rusu et al., 2019; Zhao et al., 2020), continual learning (von Oswald et al., 2020) and architecture search (Zhang et al., 2019; Brock et al., 2018). Moreover, in the last two years some works have started to leverage hypernetworks for the training of INRs, enabling the learning of latent encodings of data, while also maintaining the flexible and accurate reconstruction of signals provided by INRs. This approach has been implemented with different hypernetwork architectures, to learn priors over image data (Sitzmann et al., 2020b; Skorokhodov et al., 2021), 3D scene geometry (Littwin & Wolf, 2019; Sitzmann et al., 2019; 2020a) and material appearance (Sztrajman et al., 2021). Tancik et al. (2021) leverage hypernetworks to speed-up the training of INRs by providing learned initializations of the network weights. Sitzmann et al. (2020b) combine a set encoder with a hypernetwork decoder to learn a prior over INRs representing image data, and apply it for image in-painting. Our hypernetwork architecture from Section 3 is similar to Sitzmann *et al.*'s, however we learn a prior over the space of time series and leverage it for new data synthesis through interpolation of the learned embeddings.

**Interpretable Time Series**    Seasonal-trend decomposition techniques are standard tools in time series analysis used to decompose a time series into trend, seasonal, and remainder components. The trend component encapsulates the slow time-varying behavior of the time series, while seasonal components capture recurring (i.e., periodic) fluctuations in the data. These techniques enable an intuitive and interpretable analysis of time series data which play an important role in a variety of downstream tasks, including forecasting and anomaly detection. The classic approach for performing the decomposition is the widely used STL algorithm (Cleveland et al., 1990). To account for outliers and distributional shifts, a robust version of the algorithm, called Robust STL, has also been proposed (Wen et al., 2019). Additional challenges in seasonal-trend decomposition involve dealing with complex time series data that exhibit multiple seasonal components, to which techniques such as multiple STL (MSTL) have been proposed (Bandara et al., 2022). The ability to break

time series into interpretable components has been a topic of recent interest in the context of anomaly detection, forecasting, and generation. Relevant to this work is the recently proposed N-BEATS architecture (Oreshkin et al., 2020), a deep learning-based univariate time series forecasting solution that provides time series interpretability capabilities without considerable loss in predictive performance. The N-BEATS architecture explicitly encodes seasonal-trend decomposition into the network by defining two blocks: a trend block which uses a small ordered polynomial to capture slow varying behaviors, and a seasonality block which uses a Fourier series to capture cyclical patterns. Little work, however, has been done in the design of generation schemes that allow for decomposition of time series data into interpretable components. While TimeVAE (Desai et al., 2021) proposes a VAE architecture where the decoder has trend and seasonality blocks to allow for interpretable generation, no results highlighting the advantage of this capability were demonstrated.

## B  DATASETS

Here we introduce in detail the datasets used in our evaluation. We use publicly available Google stocks data from Yahoo finance, and the UCI Energy dataset (Candanedo et al., 2017). Google stock dataset contains daily observations from 2004 to 2019 with 6 features, namely *open, high, low, close, adjusted close, and volume*. The energy data contains 28 features with 10-minute resolution. Finally, we consider 4 datasets with longer real-world time-series from Monash repository (Godahewa et al., 2021), namely FRED-MD, NN5 Daily, Temperature Rain, and Solar Weekly. The first three datasets have time-series of length of around 700, while the latter one has time-series with length of 52. The datasets characteristics are summarized in Table 8.

In order to make a fair comparison with current state-of-the-art methods, we process the Stock and Energy in two different ways: in line with Yoon et al. (2019) and Jeon et al. (2022), we slice the data using a window of 24 time steps, corresponding to datasets Stock24 and Energy24. Following Coletta et al. (2023), we select one feature per dataset (univariate) and slice it using windows of 72 and 360 time steps, which correspond to Stock72 and Energy360.

Table 8: Main characteristics of the datasets used.

| Dataset | Number of Samples | Length of Time series | No of Features | Source |
|---|---|---|---|---|
| Stock24 | 3661 | 24 | 6 | |
| Stock72 | 3613 | 72 | 1 | Link |
| Stock360 | 3325 | 360 | 1 | |
| Energy | 19635 | 24 | 28 | Link |
| FRED-MD | 107 | 728 | 1 | Link |
| NN5 Daily | 111 | 791 | 1 | Link |
| Temp Rain | 32072 | 725 | 1 | Link |
| Solar Weekly | 137 | 52 | 1 | Link |

## C  FOURIER-BASED LOSS

As part of the training of our iHyperTime architecture, we propose a Fourier spectrum reconstruction loss. For a discrete-time signal $\mathbf{f} = \{f_0 = f(0), f_1 = f(1), \ldots, f_N = f(N)\}$, the $N$-point discrete Fourier transform (DFT) is utilized to obtain the corresponding frequency domain representation of $\mathbf{f}$ through the following operation:

$$F_k = [\mathcal{F}_T\{\mathbf{f}\}]_k = \sum_{n=0}^{N-1} f_n e^{-2\pi i\left(\frac{kn}{N}\right)}, \quad 0 \leq k \leq N-1,$$

where $i = \sqrt{-1}$ corresponds to the imaginary unit of a complex number. The coefficient $F_k \in \mathbb{C}$ quantifies the strength in representation of the $k$th frequency component of the signal. The DFT has a time complexity of $\mathcal{O}(N^2)$. In practice, an algorithm called the fast Fourier transform (FFT) is used to compute the DFT due to its lower time complexity (i.e., $\mathcal{O}(N \log N)$). Using the FFT to obtain the

frequency domain representations of two discrete-time signals $\mathbf{f}$ and $\hat{\mathbf{f}}$, we introduce a Fourier-based reconstruction loss as follows:

$$\mathcal{L}_{\text{FFT}} = \frac{1}{N} \sum_{k=0}^{N-1} \|F_k - \hat{F}_k\|.$$

Here, we utilized the PyTorch implementation of the FFT to obtain the DFT for each signal. It is important to note that the DFT is only well-defined for regularly sampled signals. In the case of this work, the discrete-time signal $\mathbf{f}$ is obtained by deterministically sampling the function $f(t)$ via a discretized grid of time steps $t \in \{0, 1, \ldots, N\}$.

## D    IMPLEMENTATION & REPRODUCIBILITY DETAILS

### D.1    BASELINES

We use the following methods with publicly available code as benchmark for our method:

- Fourier Flows (Alaa et al., 2021): `https://github.com/ahmedmalaa/Fourier-flows`

- TimeGAN (Yoon et al., 2019): `https://github.com/jsyoon0823/TimeGAN`

- GT-GAN(Jeon et al., 2022): `https://github.com/Jinsung-Jeon/GT-GAN`

- RCGAN (Esteban et al., 2017): `https://github.com/3778/Ward2ICU`

- LS4(Zhou et al., 2023): `https://github.com/alexzhou907/ls4/tree/main`

- DiffTime(Coletta et al., 2023): `https://arxiv.org/abs/2307.01717`

We adapted *TimeGAN*, *RCGAN*, and *GT-GAN* for longer time-series by setting the hidden dimensions to be equal to the time-series length, as suggested by authors and empirically evaluated. Moreover, we improve RCGAN discriminator to handle longer time-series more effectively using the CSDI transformer architecture (Tashiro et al., 2021). For DiffTime we reach out the authors to get access to their code, and to handle missing data we dynamically mask the input time-series to let the model learn to reconstruct the whole original time-series, similarly to CSDI approach for imputation (Tashiro et al., 2021).

### D.2    IMPLEMENTATION DETAILS

iHyperTime is composed of a set encoder, three decoder (hypernetworks) whose outputs corresponds to the weights of each of the blocks in TSNet. Below we explain each component in detail.

**Set Encoder**    The set encoder is a SIREN with two hidden layers of 128 neurons and an output layer (embedding) of 40 neurons. It takes as input an arbitrary set of tuples $\{(t_i, \mathbf{y}_i)\}_{i=1}^N$, where $t$ denotes the time-coordinate and $\mathbf{y}_i$ the corresponding univariate or multivariate time series value. We use a single floating point value as temporal coordinate (time $t$). As data pre-processing, the values are scaled to the interval $[-1, 1]$, with a common global factor for all time series of a dataset. MinMax scaling is also applied to the time series amplitudes, although in the interval $[0, 1]$. In all cases, regardless of sequence length, the time series is fed to the set encoder as a single set, and is hence converted into a single embedding $Z$.

**Decoder (Hypernetwork)**    Each decoder block (hypernetwork) is a one-layer MLP with ReLU activations, with a hidden layer of dimension 128. The output of each hypernetwork is a vector that contains the weights of its corresponding decomposition block. Table 9 shows the dimension details, where $n_t$, $n_S$ and $n_R$ correspond to the number of weights in the trend, seasonality and residuals blocks, which form TSNet.

Table 9: Architecture of the hypernetworks in the decoder.

| Hypernet block | Design | Input size | Output size |
|---|---|---|---|
| Trend Hypernet | Relu | 10 | 128 |
| | Linear | 128 | $n_T$ |
| Season Hypernet | Relu | 15 | 128 |
| | Linear | 128 | $n_S$ |
| Res. Hypernet | Relu | 15 | 128 |
| | Linear | 128 | $n_R$ |

**TSnet Architecture**   TSnet is an implicit neural representation of univariate/multivariate time series data. It is composed of three distinctive blocks that perform a trend-seasonality-residual additive decomposition of the time series signal. Table 10 shows the network details of each component of TSNet. In the trend block, $p$ corresponds to the degree of the polynomial, $L$ corresponds to the max length of the time series, and $m$ corresponds to the number of features.

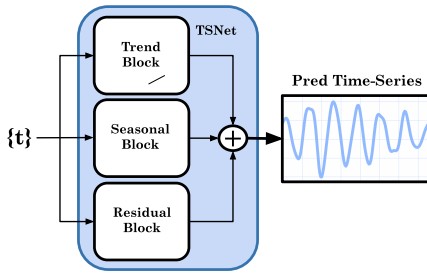

Figure 6: Diagram of the TSnet architecture.

Table 10: Architecture of TSNet.

| | Layer | Design | Input size | Output size |
|---|---|---|---|---|
| Trend block | 1 | Linear | $p$ | $m$ |
| Seasonality block | 1 | Linear | $L$ | $m$ |
| Residual block | 1 | Sine(Linear) | 1 | 60 |
| | 2 | Sine(Linear) | 60 | 60 |
| | 3 | Sine(Linear) | 60 | 60 |
| | 4 | Sine(Linear) | 60 | 60 |
| | 5 | Linear | 60 | $m$ |

**Training**   The training of iHT is performed in three stages to improve stability: 1) we train the Trend HyperNetwork, Trend Block for 100 epochs, computing the MSE loss between the ground truth time series $\mathbf{y}$ and the output of the block: $\mathcal{L}_1 = \sum_i \|y_i - \hat{f}_{\mathrm{tr}}(t)\|^2$. This leads to a smooth approximation of the time series, which we use as initial guess for the second stage. 2) We then train the Trend and Seasonality blocks together, computing the MSE reconstruction loss $\mathcal{L}_{\mathrm{rec}}$ and the FFT loss $\mathcal{L}_{\mathrm{FFT}}$ between the ground truth and the added output of both TSnet blocks. 3) Finally, we train the three blocks together.

- Stage 1 training:
    - Number of epochs: 100
    - Learning rate: $1e-3$
- Stage 2 training:
    - Number of epochs: 150
    - Learning rate: $5e-5$

- Stage 3 training:
    - Number of epochs: 150
    - Learning rate: $5e - 5$
- Batch size: 256
- $\lambda_1 = 1.0 \times 10^{-3}$
- $\lambda_2 = 1.0$
- $\lambda_3 = 1.0 \times 10^{-2}$

We train Energy24, Stock24, Stock72, Stock360 and Solar Weekly datasets for 400 epochs, with Adam optimizer. For the NN5 daily, and Fred MD datasets we trained for 500 epochs, and for the Temperature Rain dataset we train iHT for 1500 epochs.

**Hardware and Software**   We implement our method in Python and the experiments are ran using a *g4dn.2xlarge* AWS instance with a NVIDIA T4 GPU, 8 CPU and 32gb of RAM.

# E   ADDITIONAL EXPERIMENTAL RESULTS ON REAL-WORLD TIME SERIES SYNTHESIS

We show additional comparisons of time series synthesis the four Monash datasets in Table 11. Out method still shows competitive results across most datasets, with best predictive score in three cases, only loosing against Latent ODE in the FRED-MD dataset.

| Data | Metric | RNN-VAE | GP-VAE | ODE$^2$VAE | Latent ODE | TimeGAN | SDEGAN | SaShiMi | LS4 | iHT (Ours) |
|---|---|---|---|---|---|---|---|---|---|---|
| FRED-MD | Marginal ↓ | 0.132 | 0.152 | 0.122 | 0.0416 | 0.0813 | 0.0841 | 0.0482 | 0.0221 | **0.0177** |
| | Class. ↑ | 0.0362 | 0.0158 | 0.0282 | 0.327 | 0.0294 | 0.501 | 0.00119 | 0.544 | **1.3278** |
| | Prediction ↓ | 1.47 | 2.05 | 0.567 | **0.0132** | 0.0575 | 0.677 | 0.232 | 0.0373 | 0.0181 |
| NN5 Daily | Marginal ↓ | 0.137 | 0.117 | 0.211 | 0.107 | 0.0396 | 0.0852 | 0.0199 | **0.00671** | 0.00893 |
| | Class. ↑ | 0.000339 | 0.00246 | 0.00102 | 0.000381 | 0.00160 | 0.0852 | 0.0446 | **0.636** | 0.4982 |
| | Prediction ↓ | 0.967 | 1.169 | 1.19 | 1.04 | 1.34 | 1.01 | 0.849 | 0.241 | **0.2349** |
| Temp Rain | Marginal ↓ | 0.0174 | 0.183 | 1.831 | **0.0106** | 0.498 | 0.990 | 0.758 | 0.0834 | 0.2978 |
| | Class. ↑ | 0.00000212 | 0.0000123 | 0.0000319 | 0.0000419 | 0.00271 | 0.0169 | 0.0000167 | 0.976 | **11.2493** |
| | Prediction ↓ | 159 | 2.305 | 1.133 | 145 | 1.96 | 2.46 | 2.12 | 0.521 | **0.132** |
| Solar Weekly | Marginal ↓ | 0.0903 | 0.308 | 0.153 | 0.0853 | 0.0496 | 0.147 | 0.173 | 0.0459 | **0.03273** |
| | Class. ↑ | 0.0524 | 0.000731 | 0.0998 | 0.0521 | 0.6489 | 0.591 | 0.00102 | 0.683 | **1.2413** |
| | Prediction ↓ | 1.25 | 1.47 | 0.761 | 0.973 | 0.237 | 0.976 | 0.578 | 0.141 | **0.0739** |

Table 11: Generation results on FRED-MD, NN5 Daily, Temperature Rain, and Solar Weekly.

# F   TRAINING TIME COMPARISON

Table 12 shows the training time of iHT and all the other baselines for the Energy and Stock datasets. iHT has the lowest training time across all datasets, with Fourier Flows having a similar performance in the Stock datasets. In the case of Energy, given that Fourier Flows trains on each feature separately, this sequential training increases the computational time because of the large number of features present in the dataset. The training times of TimeGAN and GTGAN are orders of magnitude larger for the datasets with the longest time series, in the case of TimeGAN because it based on RNNs, whilst GTGAN's needs to solve various differential equations.

# G   ADDITIONAL TREND-SEASONALITY DECOMPOSITION ANALYSIS

In this section we provide further details of the analysis of iHT decomposition.

**Synthetic dataset**   We generated time series datasets that have trend (T), seasonal (S) and noise (R) components, or a combination of two of them. The trend was generated by randomly choosing the degree of the polynomial, a sign and a slope. A code example with the parameters is shown in Code Snippet 1. To model the seasonality component we use a Sine function with the frequency sampled uniformly within [1,10]. Finally, for the residual component we used Gaussian noise, with

Table 12: Comparison of training time. iHT shows the shortest training time on all datasets.

| Training Time (HH:MM) | Energy24 | Stock24 | Stock72 | Stock360 |
|---|---|---|---|---|
| **iHT (Ours)** | **00:15** | **00:03** | **00:03** | **00:04** |
| GTGAN | 10:39 | 12:20 | 04:32 | 21:23 |
| TimeGAN | 12:28 | 11:40 | 34:30 | 65:00 |
| FourierFlows | 02:48 | 00:07 | **00:03** | 00:05 |
| LS4 | 04:19 | 00:57 | 01:16 | 02:09 |
| DiffTime | 17:03 | 02:52 | 01:42 | 02:13 |

the standard deviation sampled between 0 and 0.2. For each dataset, we generated 2000 time series of 200 time steps. Figure 7 shows examples of each dataset.

Code Snippet 1: Trend generation

```python
def generate_trend():
    trend_slope = np.random.uniform(1.5,2)
    degree = np.random.choice([1,2,3])
    sign = np.random.choice((-1, 1))
    trend = sign * (trend_slope*regular_time_samples)**degree
    return trend
```

Table 13, we compare iHT with *STL (exact)* and *STL (approx)*, in the previous dataset TSR, T+R, T+S the additional dataset of S+R dataset. We can observe that in the S+R dataset iHT shows the worst performance in all three components. Interestingly, both STL (exact) and STL (approx) show higher errors than in the other datasets, showing that this is a more challenging case.

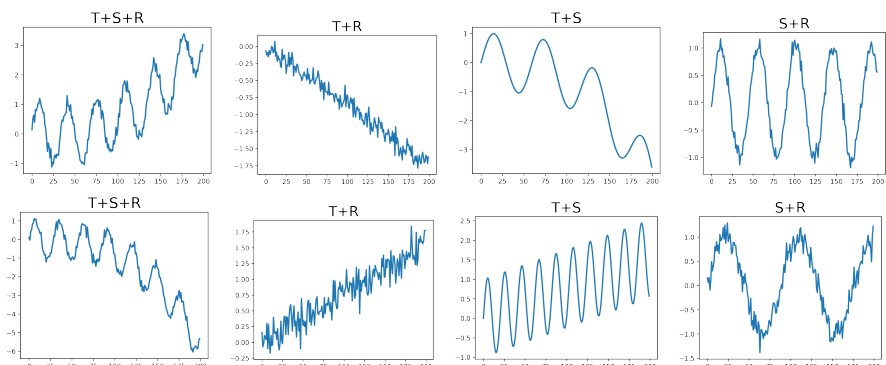

Figure 7: Example of synthetic datasets with trend (T), seasonal (S) and noise (R) components, or a combination of two of them.

| MSE ($\times 10^{-3}$) | TSR STL (exact) | STL (approx) | iHT (Ours) | T+R STL (exact) | STL (approx) | iHT (Ours) | T+S STL (exact) | STL (approx) | iHT (Ours) | S+R STL (exact) | STL (approx) | iHT (Ours) |
|---|---|---|---|---|---|---|---|---|---|---|---|---|
| Trend | $0.46 \pm 2.08$ | $1.04 \pm 4.69$ | $0.6 \pm 0.5$ | $0.07 \pm 0.07$ | $1.26 \pm 5.7$ | $0.2 \pm 0.27$ | $0.46 \pm 2.1$ | $1.17 \pm 5.12$ | $0.46 \pm 0.48$ | $0.18 \pm 0.19$ | $0.71 \pm 1.9$ | $8.11 \pm 8.31$ |
| Seasonality | $0.57 \pm 2.1$ | $1.19 \pm 4.73$ | $1.21 \pm 0.68$ | $0.03 \pm 0.02$ | $1.35 \pm 5.74$ | $0.17 \pm 0.29$ | $0.44 \pm 2.09$ | $1.18 \pm 5.11$ | $1.25 \pm 0.77$ | $2.54 \pm 2.68$ | $3.89 \pm 3.93$ | $14.86 \pm 11.49$ |
| Residuals | $0.17 \pm 0.15$ | $0.18 \pm 0.12$ | $1.29 \pm 0.62$ | $0.15 \pm 0.09$ | $0.18 \pm 0.13$ | $0.41 \pm 0.26$ | $0.03 \pm 0.1$ | $0.04 \pm 0.07$ | $1.25 \pm 0.71$ | $2.64 \pm 2.73$ | $3.45 \pm 3.07$ | $13.98 \pm 5.71$ |

Table 13: Ablation study for trend-seasonality-residual decomposition of time series. We compare iHT against STL decomposition, for three dataset configurations: trend+seasonality (T+S), trend+residual (T+R), seasonality+residual (S+R) and all three components (TSR).

In Figures 8, 9, 10 and 11 we show the evaluation of the output of iHT for each dataset. The first three plots on each row correspond to the distribution of trend, seasonality and residuals outputs from iHT against the ground truth components. We can see in Figures 9 and 10 that for the T+S and T+R datasets, the trend shows good agreement, and in the case of no residuals, we observe a narrow distribution close to zero, while in the case of no seasonality, the frequency histogram is also very narrow around zero. In Figure 11, which corresponds to S+R we can observe that the distribution of trend is quite broad, and this is in line with the results on Table 13. Even in the seasonality distribution

we can observe a slightly worse match with regards to the other cases. The plots on the right show the learned representations via t-SNE of the embeddings. We can still observe a separation in trend up and down on the T+R dataset, although the separation in high and low frequency in the S+R plot is less obvious.

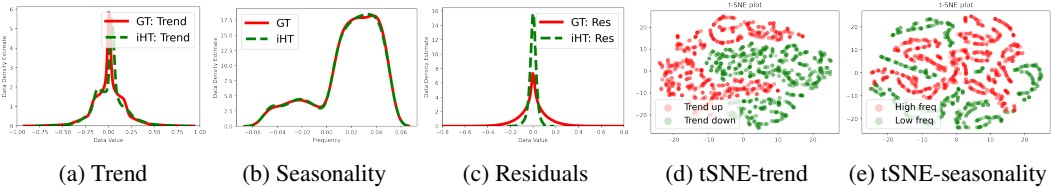

| (a) Trend | (b) Seasonality | (c) Residuals | (d) tSNE-trend | (e) tSNE-seasonality |

Figure 8: (a,b,c): Distribution of the trend, seasonality and residual outputs of iHT vs ground truth. (d,e): t-SNE visualization of trend and seasonality embeddings in iHT for the TSR dataset.

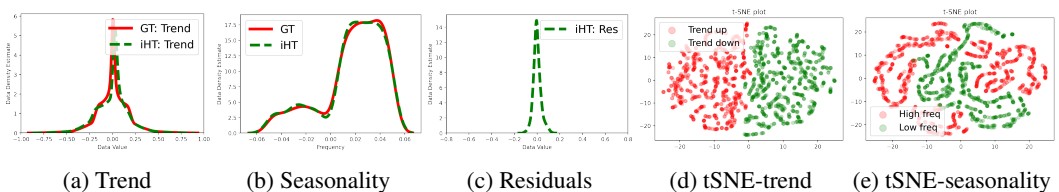

| (a) Trend | (b) Seasonality | (c) Residuals | (d) tSNE-trend | (e) tSNE-seasonality |

Figure 9: (a,b,c): Distribution of the trend, seasonality and residual outputs of iHT vs ground truth. (d,e): t-SNE visualization of trend and seasonality embeddings in iHT for the T+S dataset.

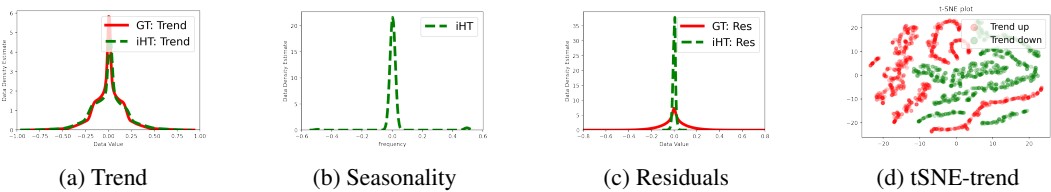

| (a) Trend | (b) Seasonality | (c) Residuals | (d) tSNE-trend |

Figure 10: (a,b,c): Distribution of the trend, seasonality and residual outputs of iHT vs ground truth. (d): t-SNE visualization of trend embeddings in iHT for the T+R dataset.

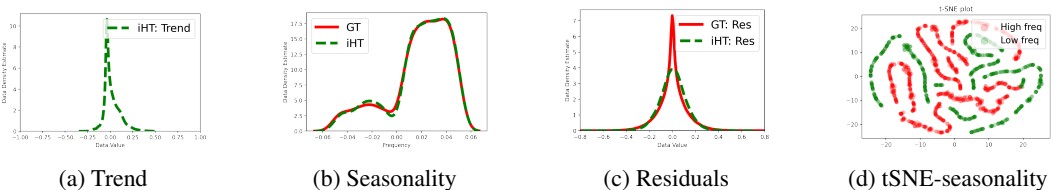

| (a) Trend | (b) Seasonality | (c) Residuals | (d) tSNE-seasonality |

Figure 11: (a,b,c): Distribution of the trend, seasonality and residual outputs of iHT vs ground truth. (d): t-SNE visualization of seasonality embeddings in iHT for the S+R dataset.

## H    TIME SERIES GENERATION USING TREND COMPONENT

Here we consider a peculiar capability of iHT that enables the user to provide an input trend component to generate time-series accordingly. iHT generates new time series by performing interpolation in the embedding space between two time series, and then generating the novel weights of TSNet that represents the novel time series. This allows us to control the generation by projecting a desired trend pattern in iHT to use in the generation process. To evaluate the performance of this guided generation, we use iHT trained on Stock24. We provide an input trend and generate 1000 time series using iHT. In this experiment we consider the following baselines: DiffTime, RCGAN, TimeGAN, GT-GAN. While DiffTime (Coletta et al., 2023) is naturally designed for constrained time-series generation, we adapted RCGAN, TimeGAN, GT-GAN architectures to deal with the input trend. In detail, we re-trained them as conditioned models using an additional input trend, computed as a polynomial interpolation from original data during the training. In Figure 12 we show how the generated time-series follow the trend. For each approach we generate 1000 samples and we plot their 5-95th percentile values as the light-blue shaded area, while trend is the dotted orange line. In Table 14 we report the quantitative metrics, which evaluate how much each generated time-series deviate from the input trend by computing the L2 distance and Dynamic-Time-Warping (DTW) distance between the generated sample and the input trend. The results show that iHT has among the best performance and it is competitive w.r.t. to DiffTime, which is specifically designed to incorporate such trend constraints.

Table 14: Time-Series Trend Generation on Stock24 dataset.

| Algo | L2 Distance | DTW Distance |
|---|---|---|
| iHT (Ours) | 23.68±18.6 | **14.43±13.60** |
| DiffTime | **19.83±5.40** | 15.42±4.79 |
| GT-GAN | 1304.2±1026.9 | 1303.9±1303.1 |
| TimeGAN | 88.18±12.10 | 87.29±12.35 |
| RCGAN | 60.56±9.20 | 32.94±6.05 |

Figure 12 shows additional qualitative results using iHT trained on Stock72 with a wide diversity of input trends. We can see that in all cases, the input trend is within the 5-95th percentile values of the generated time series, showing a good agreement of the synthetic time series with the input trend.

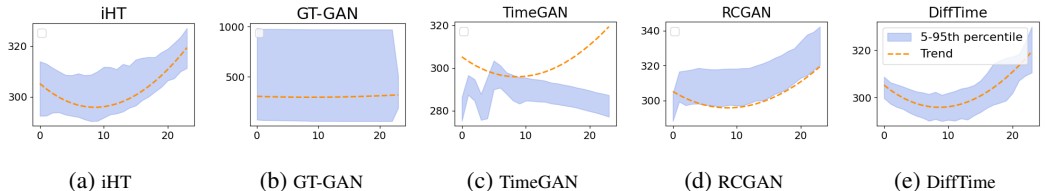

(a) iHT          (b) GT-GAN          (c) TimeGAN          (d) RCGAN          (e) DiffTime

Figure 12: A visualizations of time-series generated according the input *Trend*. The orange dotted time-series is the trend, and the shaded blue area shows the 5% and 95% percentiles of the generated synthetic time-series. Our approaches show among the best performance with time-series closer to the input trend.

## I    VISUALIZATIONS WITH TSNE AND DATA DISTRIBUTIONS

In this section we report an additional evaluation of the synthetic and real data distributions. We evaluate the synthetic distributions on Stock24, including missing data from 30% to 70%; then we analyse synthetic data on longer stock time-series, i.e., stock72 and stock360; and finally we evaluate the synthetic distributions for Energy data, which has 28 dimensions.

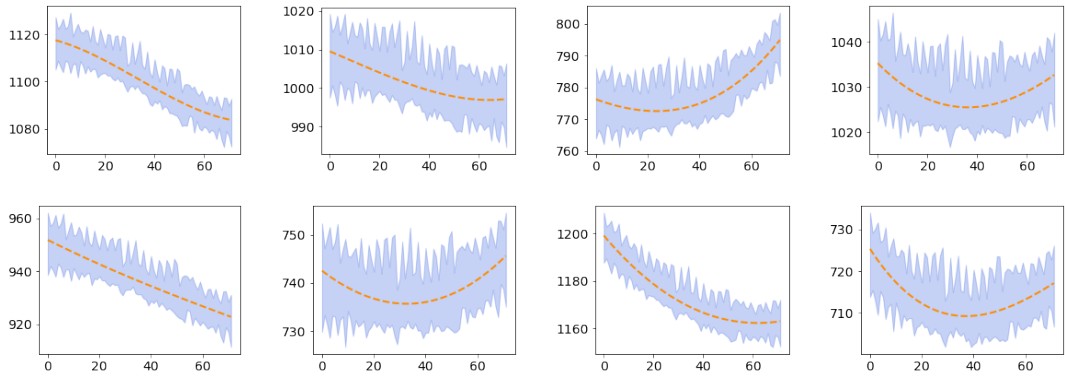

Figure 13: A visualizations of time-series generated by iHT according to an input *Trend*, on the Stock72 dataset. The orange dotted time-series is the trend, and the shaded blue area shows the 5% and 95% percentiles of the generated synthetic time-series.

## I.1 DATA DISTRIBUTION

First we plot the real (blue) and synthetic (orange) distributions empirically evaluated through a kernel-density estimation of real and generated data. Figure 14 shows the empirical distributions for Stock24, where iHT has among the closest match with original data. The superior performance of iHT is more evident with irregular data, from Figure 15 to Figure 17, where iHT is always able to closely resemble the real data distributions.

The performance of iHT is consistent with longer time-series (Figure 18 and Figure 19) and highly dimensional data like Energy in Figure 20.

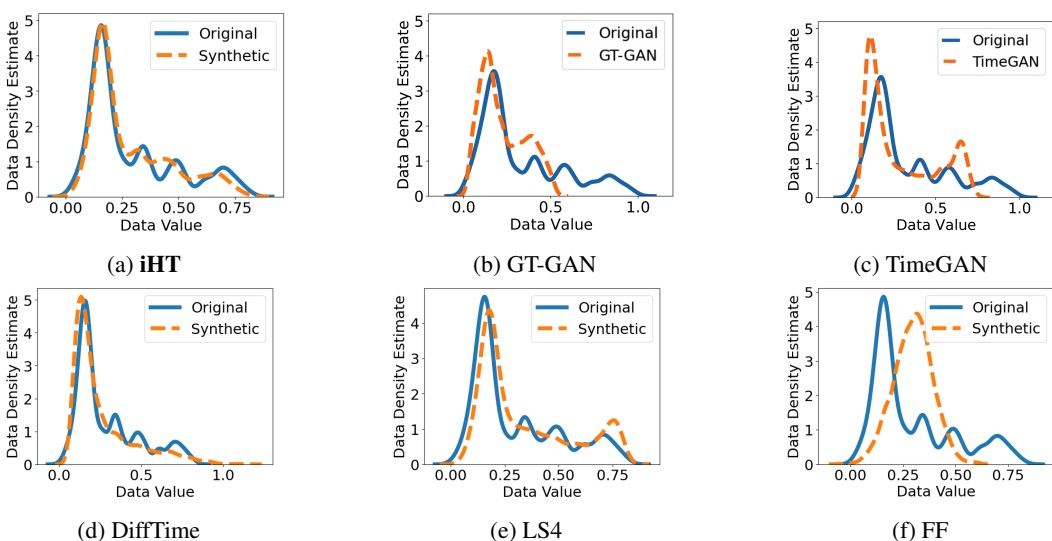

Figure 14: Data distribution on Stock24 data.

## I.2 TSNE VISUALIZATION

We now evaluate the real (blue) and synthetic (red) distributions through t-SNE visualizations. Figure 21 shows the t-SNE plots for Stock24, where iHT has among the best performance (i.e., the synthetic data almost completely overlap with real data). As mentioned in the previous section, the superior performance of iHT is more evident with irregular data, from Figure 22 to Figure 24, where iHT is the only method to closely resemble the real data distribution. While GT-GAN reproduces

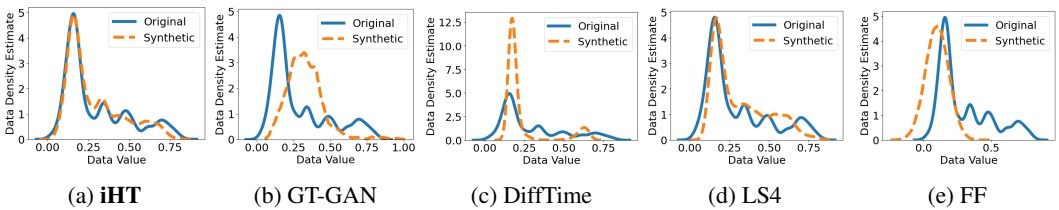

Figure 15: Data distribution on irregular Stock24 data (Missing 70%).

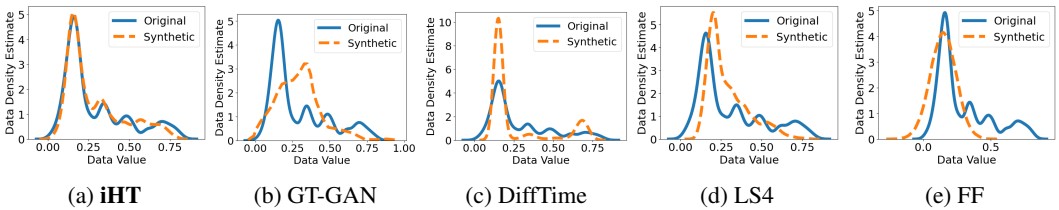

Figure 16: Data distribution on irregular Stock24 data (Missing 50%).

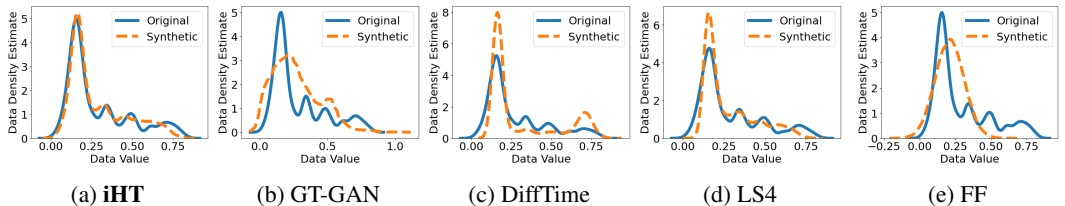

Figure 17: Data distribution on irregular Stock24 data (Missing 30%).

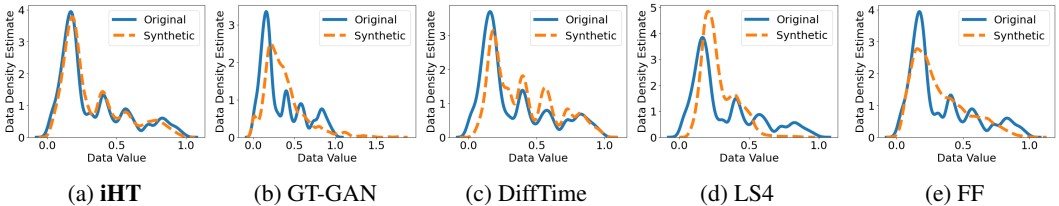

Figure 18: Data distribution on regular Stock72 data.

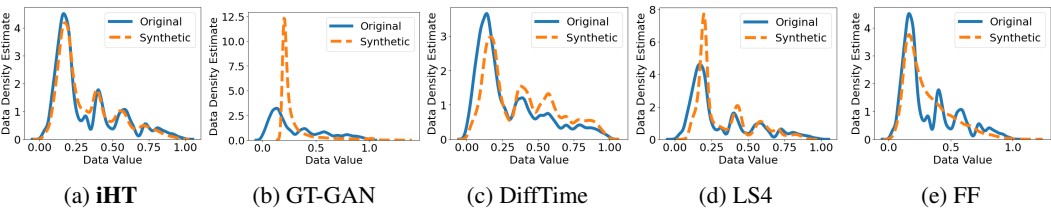

Figure 19: Data distribution on regular Stock360 data.

similarly the real data distributions, the synthetic distributions are more condensed around the original data, and don't cover the full space.

For longer time-series (Figure 25 and Figure 26) the t-SNE plots show that iHT is able to better reproduce the original data distributions. Finally, Figure 27 shows the t-SNE plots for energy data where with consistent performance for iHT.

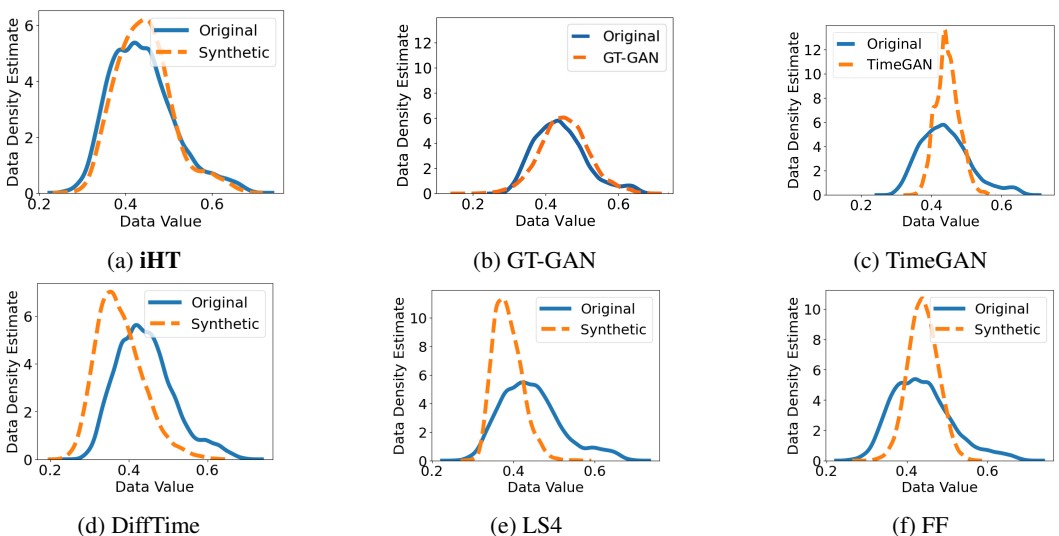

Figure 20: Data distribution on Energy data.

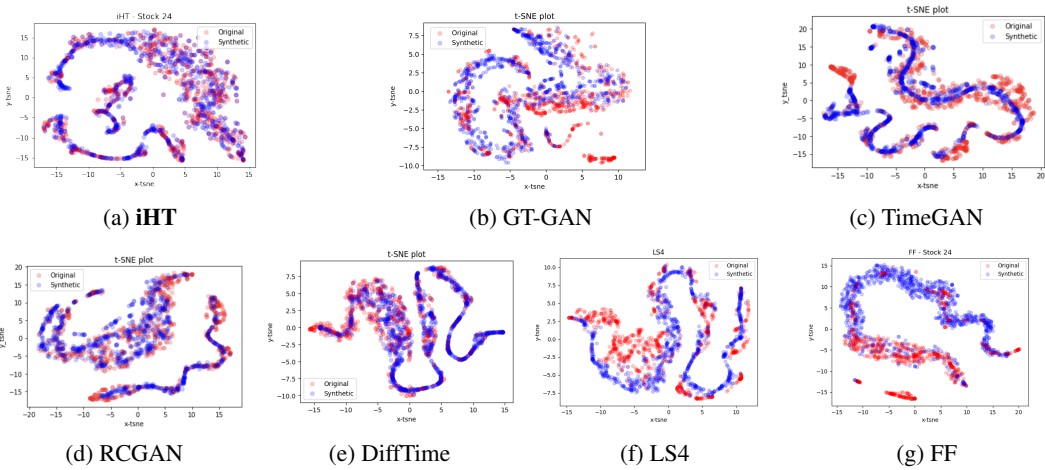

Figure 21: t-SNE visualizations on Stock24 data, where a greater overlap of blue and red dots shows a better distributional-similarity between the generated data and original data. Our approach shows the best performance.

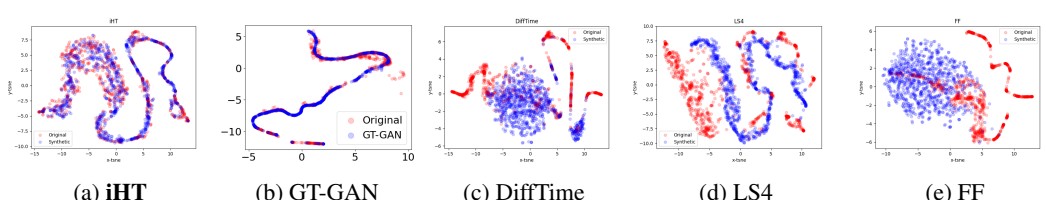

Figure 22: t-SNE visualizations on irregular Stock24 data (Missing 70%), where a greater overlap of blue and red dots shows a better distributional-similarity between the generated data and original data. Our approach shows the best performance.

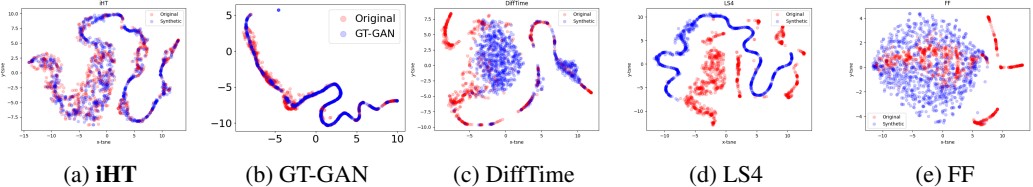

Figure 23: t-SNE visualizations on irregular Stock24 data (Missing 50%), where a greater overlap of blue and red dots shows a better distributional-similarity between the generated data and original data. Our approach shows the best performance.

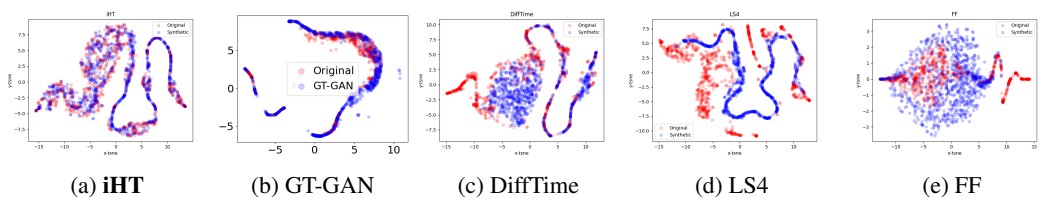

Figure 24: t-SNE visualizations on irregular Stock24 data (Missing 30%), where a greater overlap of blue and red dots shows a better distributional-similarity between the generated data and original data. Our approach shows the best performance.

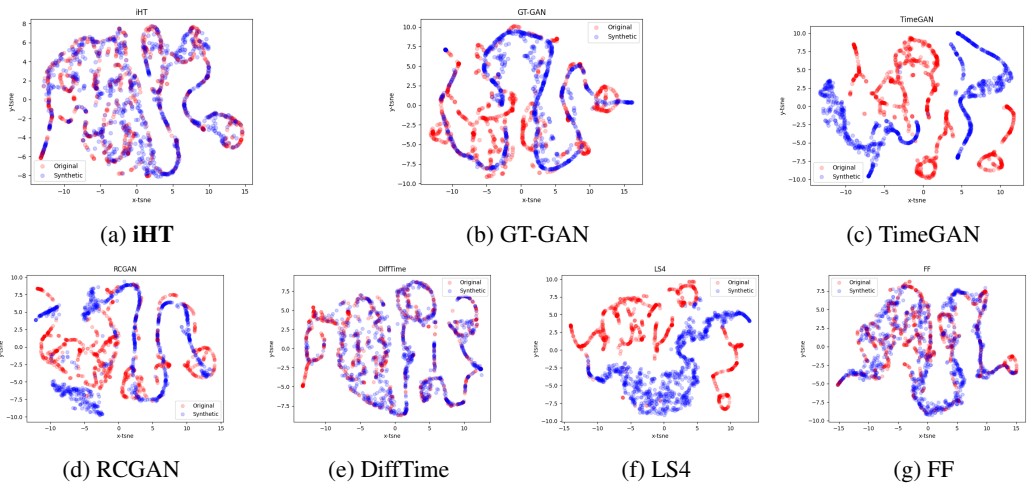

Figure 25: t-SNE visualizations on Stock72 data, where a greater overlap of blue and red dots shows a better distributional-similarity between the generated data and original data. Our approach shows the best performance.

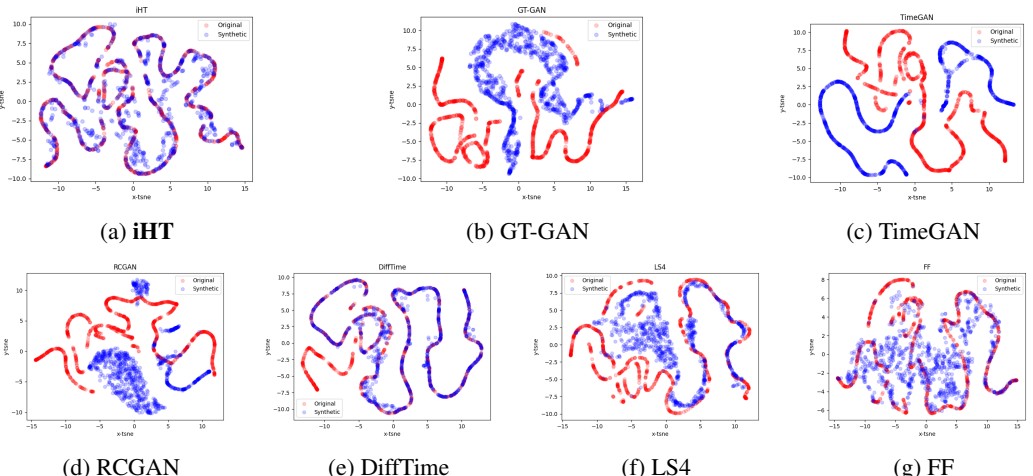

(a) **iHT**     (b) GT-GAN     (c) TimeGAN

(d) RCGAN     (e) DiffTime     (f) LS4     (g) FF

Figure 26: t-SNE visualizations on Stock360 data, where a greater overlap of blue and red dots shows a better distributional-similarity between the generated data and original data. Our approach shows the best performance.

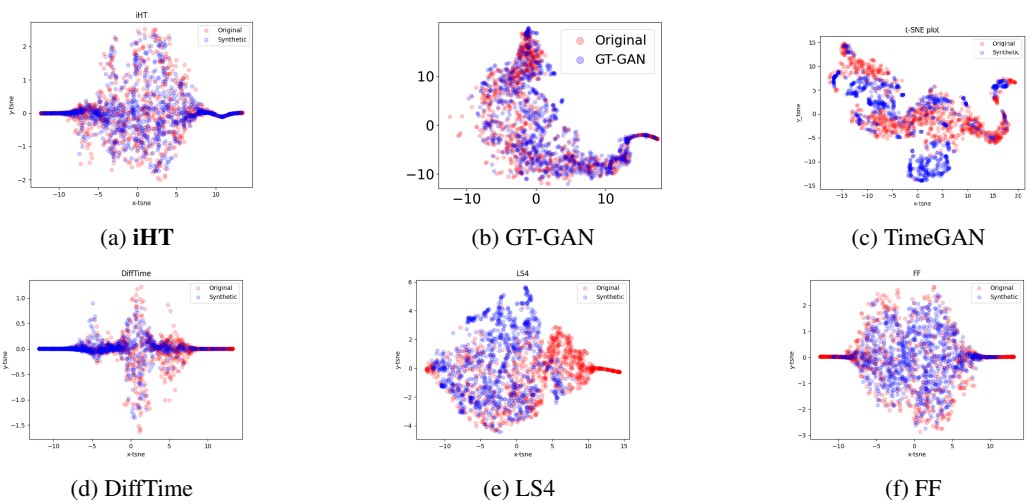

(a) **iHT**     (b) GT-GAN     (c) TimeGAN

(d) DiffTime     (e) LS4     (f) FF

Figure 27: t-SNE visualizations on Energy data, where a greater overlap of blue and red dots shows a better distributional-similarity between the generated data and original data. Our approach shows the best performance.

# J  FINANCIAL RETURNS AND AUTOCORRELATION

We now evaluate the distributions of two well known properties (i.e., *stylized facts*) of financial time-series, namely the returns and autocorrelation. We consider stock uni-variate data with length of 72. Figure 28 and Figure 29 show the *returns* and the *autocorrelation of returns* distributions, respectively. The two figures confirm the ability of iHT to learn the real data properties (i.e., the real and synthetic distributions mostly overlap).

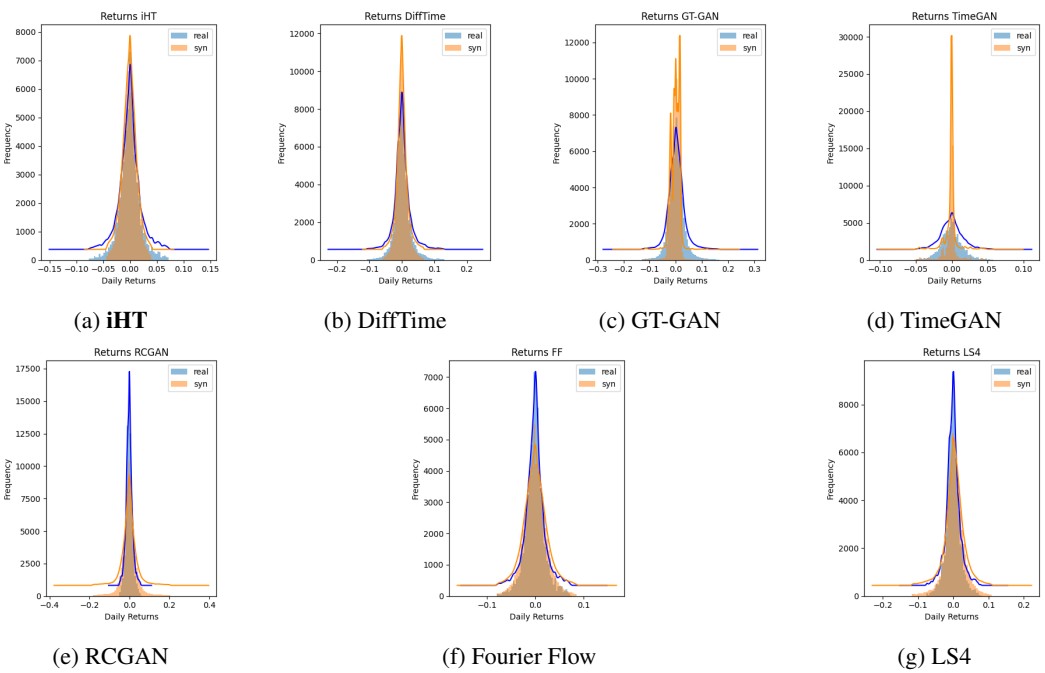

Figure 28: Returns distribution of stock time-series with length 72.

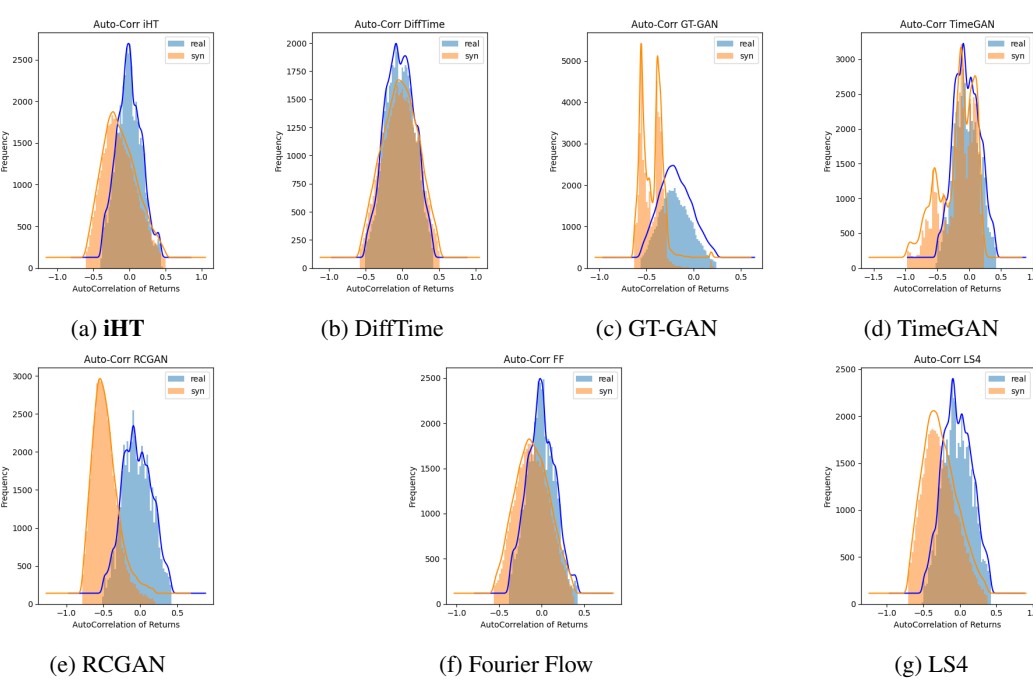

Figure 29: Autocorrelation of returns distributions of stock time-series with length 72.

## K VISUALIZATIONS OF GENERATED SAMPLES

Finally we report examples of generated time-series. It is worth to mention that in Figure 30 the synthetic time-series from iHT effectively respect the open-high-low-close relationship from the real data – high (low) is the highest (lowest) series. Such data property is preserved also when the model is trained on missing data, as shown in Figure 31, Figure 32, and Figure 33. With the exception of DiffTime, which is specifically designed for constrained time-series generation, most of the existing approaches do not preserve such property.

Energy data is shown in Figure 34 for regular time-series, and in Figure 35, Figure 36, and Figure 37 for irregular time-series. Considering that Energy has 28 features, with different scales, we plot only the first 5 normalized features.

Finally, we plot longer-times for regular stock72 in Figure 38. While we plot the irregular stock72 in Figure 39, Figure 40, and Figure 41.

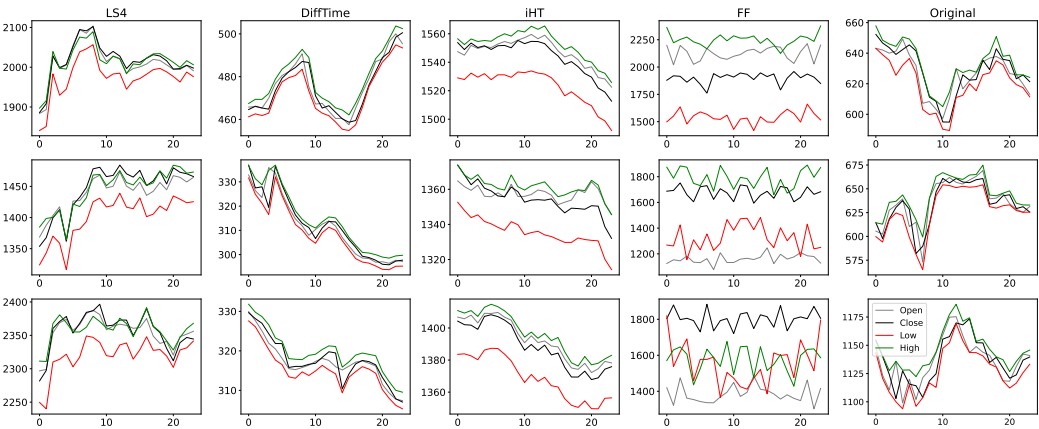

Figure 30: An example of Regular Stock24 samples.

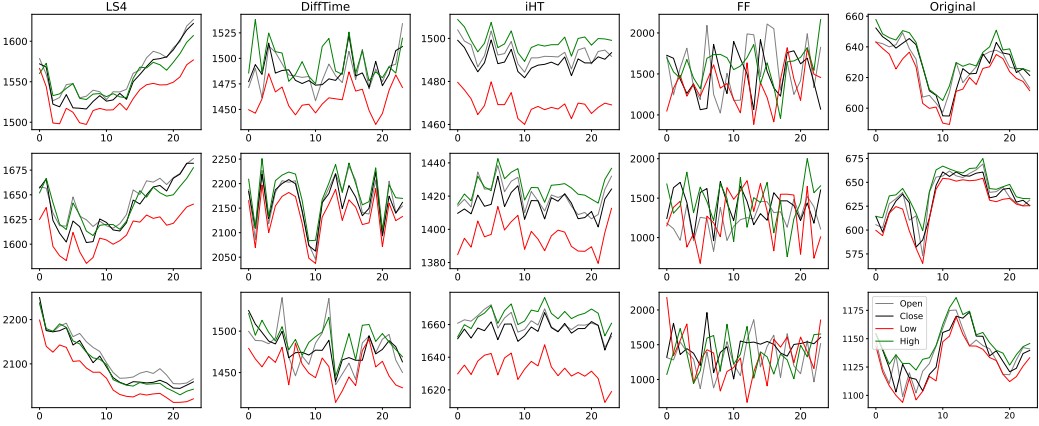

Figure 31: An example of Irregular Stock24 samples (70% missing data).

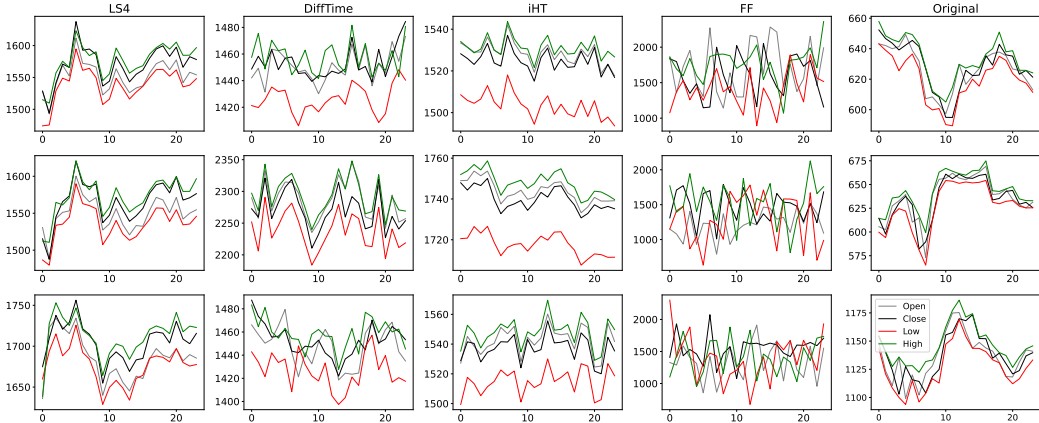

Figure 32: An example of Irregular Stock24 samples (50% missing data).

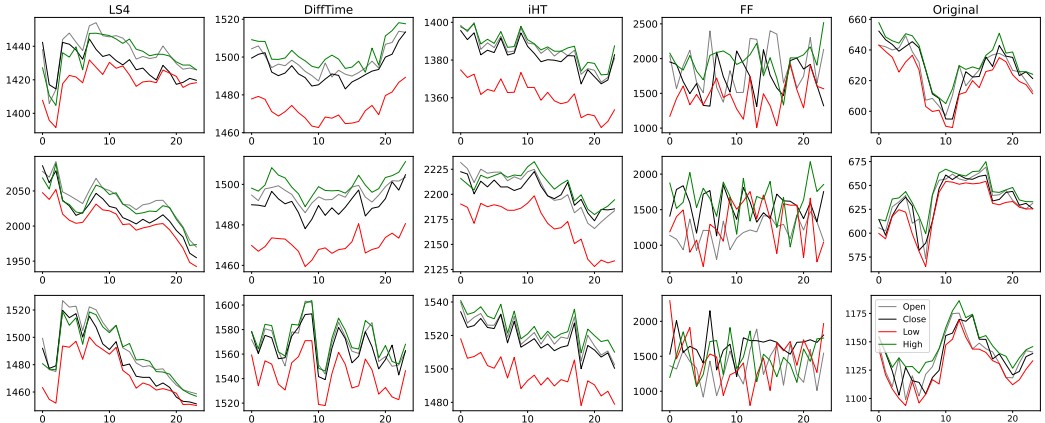

Figure 33: An example of Irregular Stock24 samples (30% missing data).

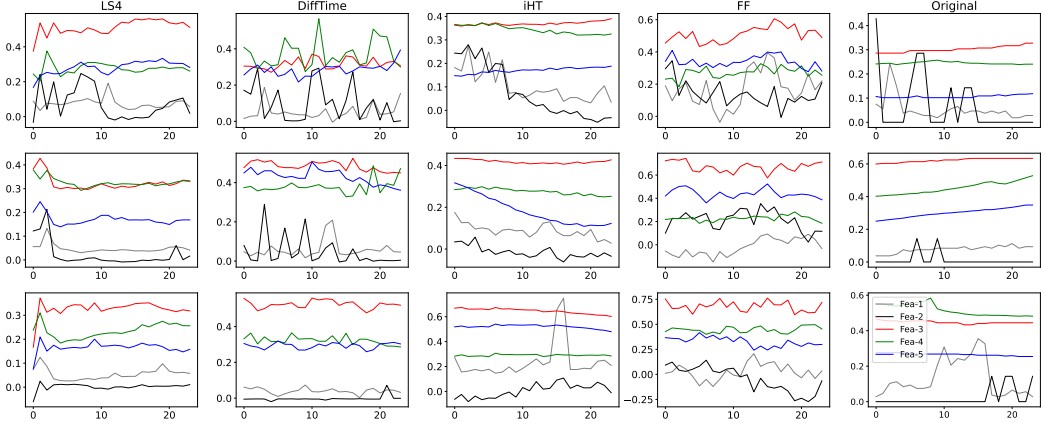

Figure 34: An example of Regular Energy24 samples.

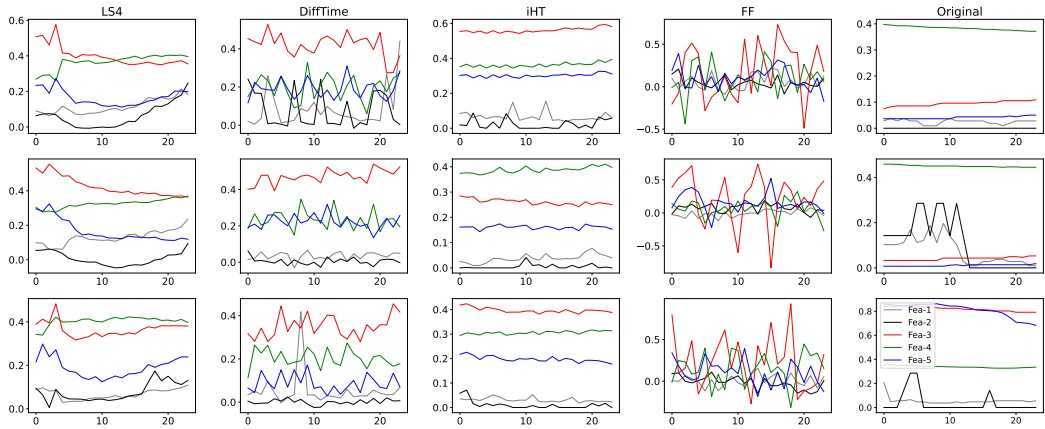

Figure 35: An example of Irregular Energy24 samples (70% missing data).

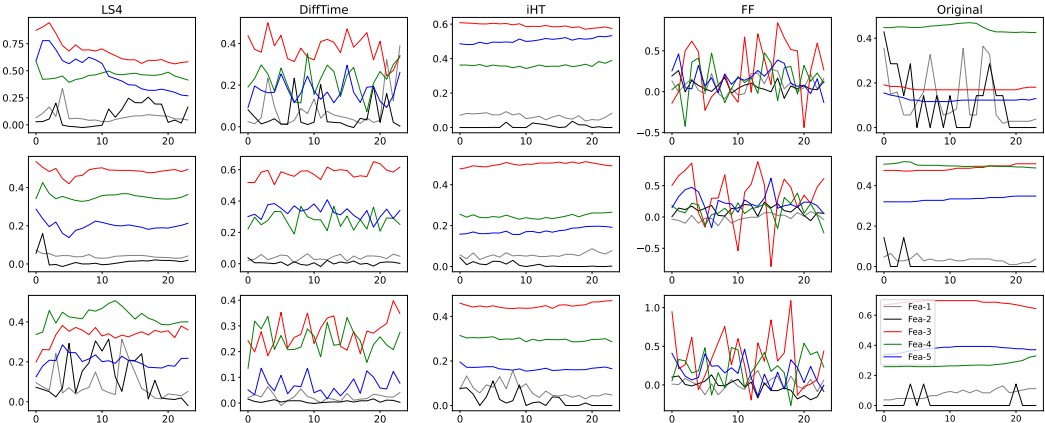

Figure 36: An example of Irregular Energy24 samples (50% missing data).

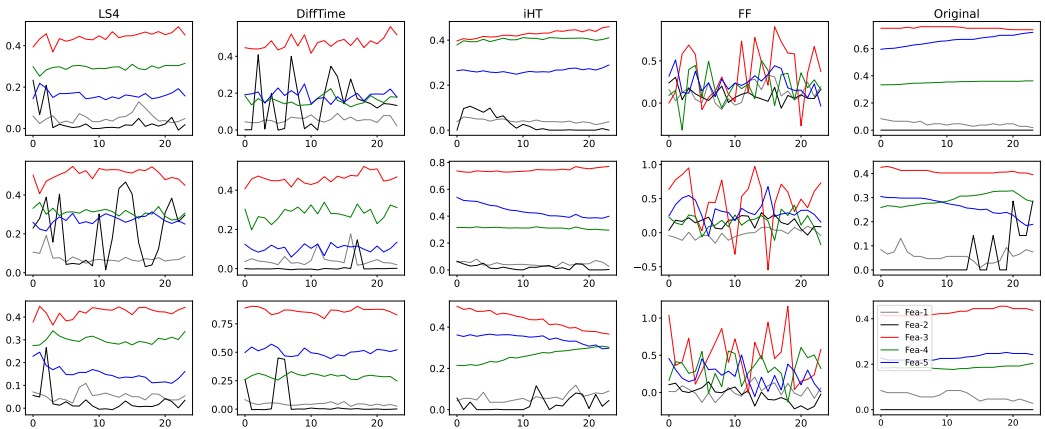

Figure 37: An example of Irregular Energy24 samples (30% missing data).

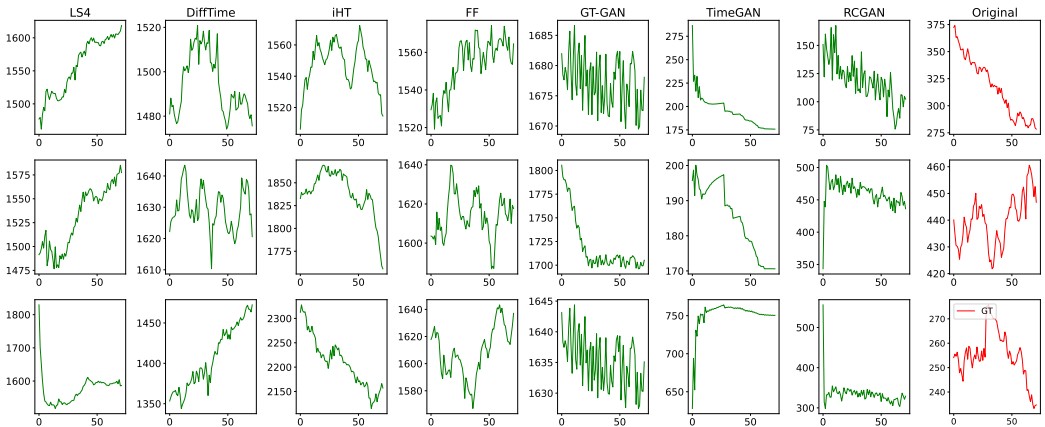

Figure 38: An example of Regular Stock72 samples.

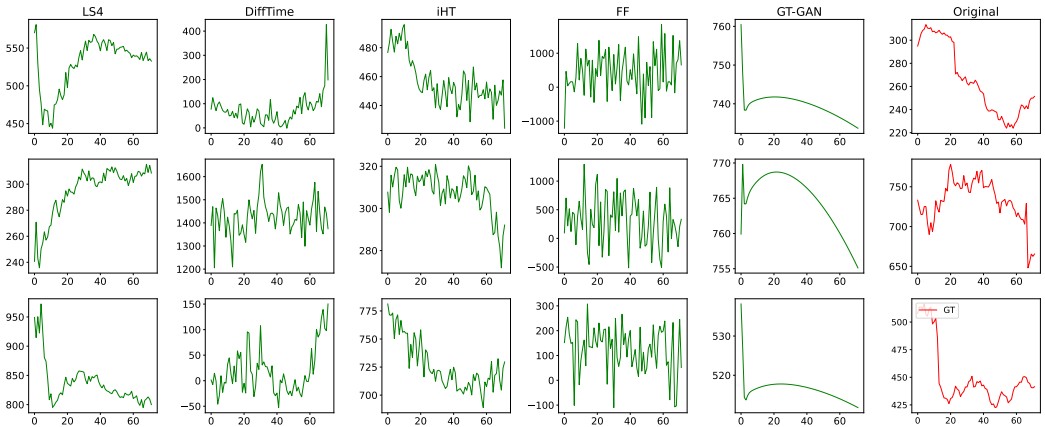

Figure 39: An example of Irregular Stock72 samples (70% missing data).

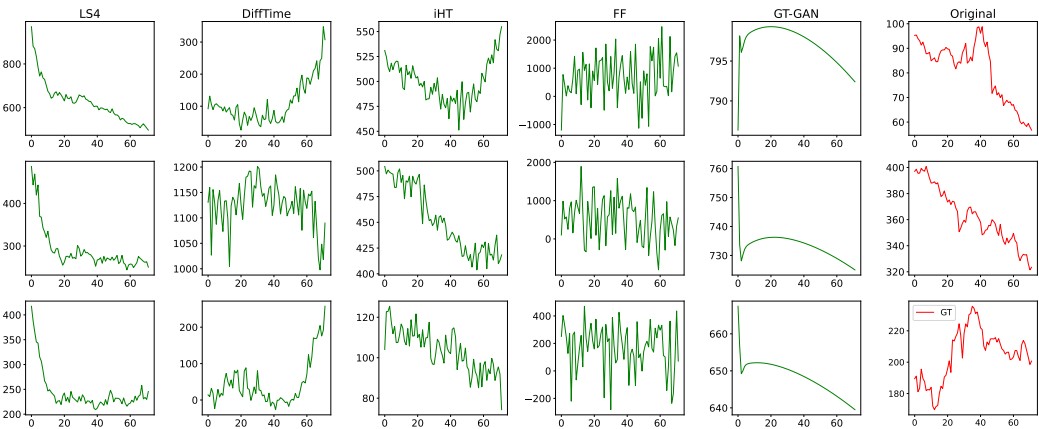

Figure 40: An example of Irregular Stock72 samples (50% missing data).

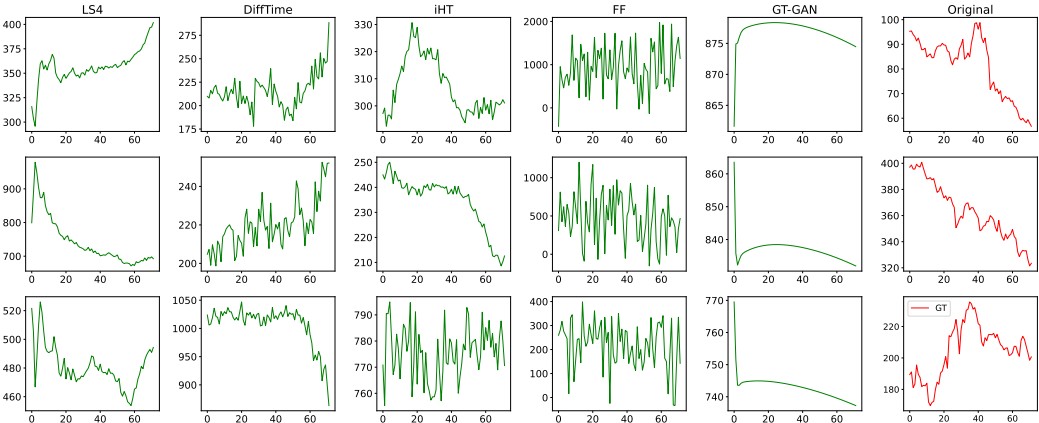

Figure 41: An example of Irregular Stock72 samples (30% missing data).

## SUPPLEMENTAL REFERENCES

Andrew Brock, Theo Lim, J.M. Ritchie, and Nick Weston. SMASH: One-shot model architecture search through hypernetworks. In ICLR, 2018.

Jeong Joon Park, Peter Florence, Julian Straub, Richard Newcombe, and Steven Lovegrove. Deepsdf: Learning continuous signed distance functions for shape representation. In CVPR, June 2019.

Jacob Russin Russin, Randall O'Reilly, and Yoshua Bengio Bengio. Deep learning needs a prefrontal cortex. In Bridging AI and Cognitive Science ICLR 2020 Workshop, 2020.

Andrei A. Rusu, Dushyant Rao, Jakub Sygnowski, Oriol Vinyals, Razvan Pascanu, Simon Osindero, and Raia Hadsell. Meta-learning with latent embedding optimization. In ICLR, 2019.

Kenneth O. Stanley, David B. D'Ambrosio, and Jason Gauci. A Hypercube-Based Encoding for Evolving Large-Scale Neural Networks. Artificial Life, 15(2):185–212, 04 2009.

Matthew Tancik, Ben Mildenhall, Terrance Wang, Divi Schmidt, Pratul P. Srinivasan, Jonathan T. Barron, and Ren Ng. Learned initializations for optimizing coordinate-based neural representations. In CVPR, 2021.

Yusuke Tashiro, Jiaming Song, Yang Song, and Stefano Ermon. Csdi: Conditional score-based diffusion models for probabilistic time series imputation. NeurIPS, 34:24804–24816, 2021.

Johannes von Oswald, Christian Henning, Benjamin F. Grewe, and João Sacramento. Continual learning with hypernetworks. In ICLR, 2020.

Chris Zhang, Mengye Ren, and Raquel Urtasun. Graph hypernetworks for neural architecture search. In ICLR, 2019.

Dominic Zhao, Seijin Kobayashi, João Sacramento, and Johannes von Oswald. Meta-learning via hypernetworks. In 4th Workshop on Meta-Learning at NeurIPS 2020 (MetaLearn 2020). NeurIPS, 2020. doi: 10.3929/ethz-b-000465883.

