# SUPPLEMENTARY
## IHYPERTIME: INTERPRETABLE TIME SERIES GENERATION WITH IMPLICIT NEURAL REPRESENTATIONS

# 1 ADDITIONAL EXPERIMENTS

## 1.1 LINEAR INTERPOLATION OF LATENT SPACE

In Figures 1 to 4, we show reconstructions of interpolations between pairs of embeddings generated by our architecture, trained on datasets of different lengths (72 and 360). In the top row, we show a smooth transition from $Z_1$ to $Z_2$, in terms of the fully reconstructed time series. Our architecture also performs a TSR decomposition of the time series, and hence in the three bottom rows we display the smooth transition between $Z_1$ and $Z_2$ of each individual component of the time series (Trend, Seasonality, Residual).

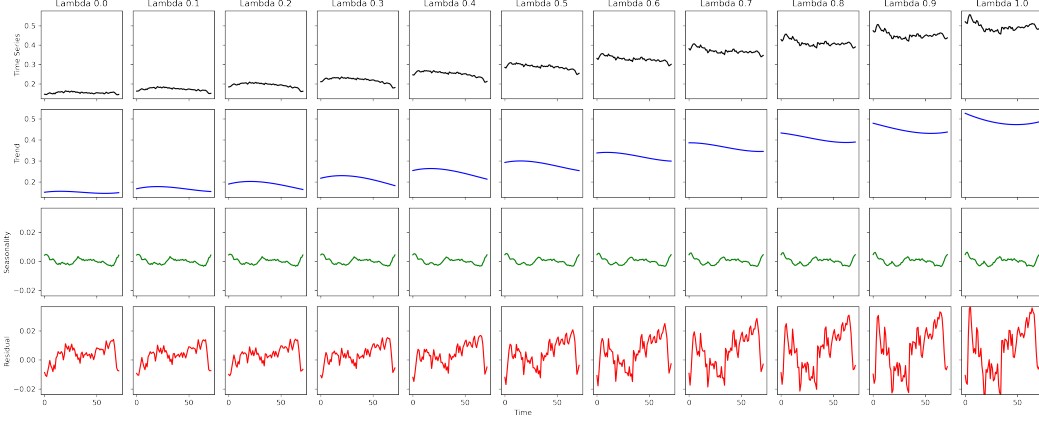

Figure 1: Example of interpolation of embeddings on Stock 72, top corresponds to the full time series and bottom to the reconstruction of each part of the embedding.

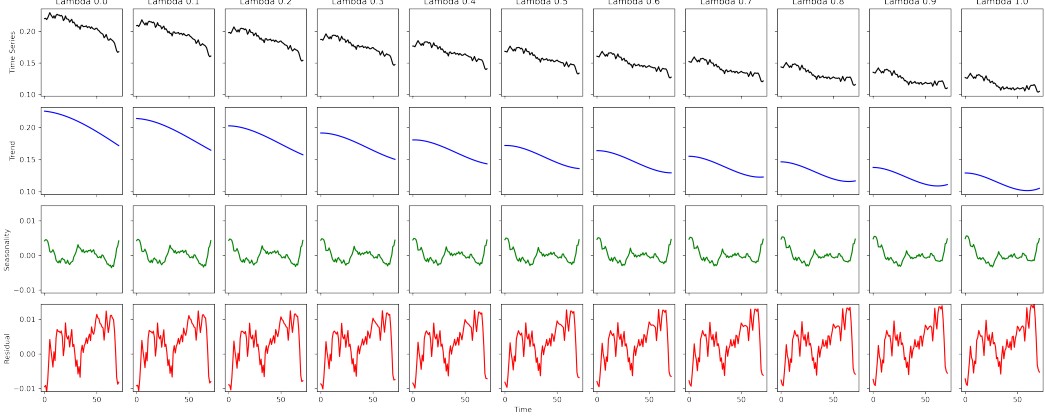

Figure 2: Example of interpolation of embeddings on Stock 72, top corresponds to the full time series and bottom to the reconstruction of each part of the embedding.

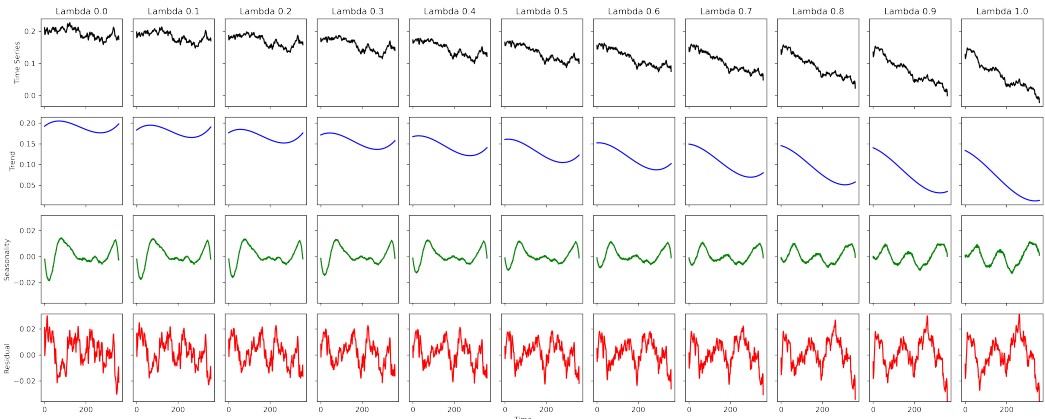

Figure 3: Example of interpolation of embeddings on Stock 360, top corresponds to the full time series and bottom to the reconstruction of each part of the embedding.

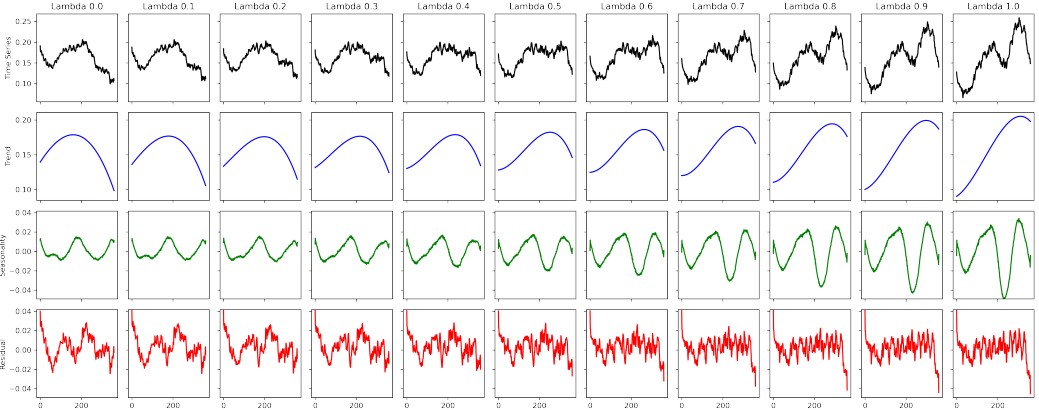

Figure 4: Example of interpolation of embeddings on Stock 360, top corresponds to the full time series and bottom to the reconstruction of each part of the embedding.

## 1.2 TRAIN AND TEST SPLIT ON STOCK DATA

In Table 1 we show the evaluation of the approaches while randomly splitting data between training set (i.e., 80% of dataset) and test set (i.e., 20% of dataset). The table reports the quantitative metrics for the regular Stock data with 24 length, while Figure 5 shows the t-SNE visualization (Top) and the data distribution (Bottom). These results confirm the results from the our main paper.

Table 1: Generation performance on regular Stock24 dataset

| Metric | iHT (Ours) | GT-GAN | TimeGAN | DiffTime | LS4 | FFlows |
|---|---|---|---|---|---|---|
| Discr-score ↓ | 0.054±0.028 | 0.273±0.046 | 0.068±0.018 | 0.079±0.014 | 0.154±0.080 | 0.426±0.032 |
| Pred-score ↓ | 0.037±0.000 | 0.046±0.001 | 0.043±0.001 | 0.044±0.001 | 0.039±0.000 | 0.055±0.003 |
| Marginal Score ↓ | 0.355 | 0.403 | 0.434 | 0.359 | 0.513 | 0.335 |
| $sym$-Recall ↑ | 0.594 | 0.500 | 0.413 | 0.787 | 0.206 | 0.000 |

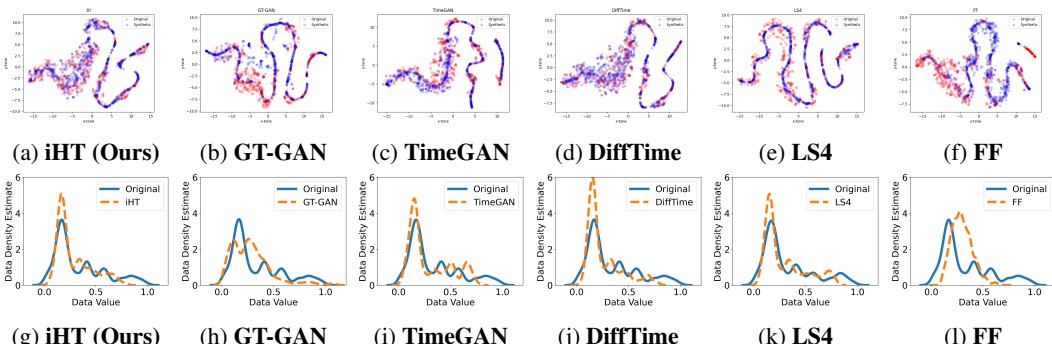

(a) **iHT (Ours)**    (b) **GT-GAN**    (c) **TimeGAN**    (d) **DiffTime**    (e) **LS4**    (f) **FF**

(g) **iHT (Ours)**    (h) **GT-GAN**    (i) **TimeGAN**    (j) **DiffTime**    (k) **LS4**    (l) **FF**

Figure 5: (Top) t-SNE visualizations on regular Stock24 data where a greater overlap of blue and red dots shows a better distributional-similarity between the generated data and original data. (Bottom) Data distribution on regular Stock24 data. Our approach shows the best performance.

## 1.3 COMPUTATIONAL TIMES

Our experiments use the same synthetic data-set of 8192000 data-points (e.g. number of samples $\times$ sequence length), organized in 5 datasets of different sequence lengths $l \in \{80, 320, 1280, 5120, 20480\}$. Thus, we obtain 5 datasets of $\{102400, 25600, 6400, 1600, 400\}$ number of samples, which we process using $\{1024, 256, 64, 16, 4\}$ as batch-size, respectively. Thanks to this batch-size, we can fairly train each model for 100 iteration and always loading to GPU the same amount of data-points, for each sequence length and iteration.For inference, we only sample a single batch of data.

**Impact of data loading** All the times are computed directly in the training and testing loop, after the model has been already loaded in the GPU memory, using a NVIDIA T4 GPU. Moreover, we found that the data loading (both for training and inference) is negligible w.r.t. model computational time. We measure the loading being around $0.0145$ sec for training (100 iterations) and inference $0.001$ sec (1 iteration).

**Fourier Flow** Finally, we add in Table 2 the computational times Fourier Flows, which trend still has some convexing. We believe the slightly drop likely comes from not optimized code to handle large batch-sizes: the longer time-series (e.g., 320 and 1280) worsen the computational time, but the model is also relieved by the smaller batch-size (e.g., 256 and 64).

Table 2: Left: Characteristics of datasets to assess computational time. Right: Details of training and inference times of Fourier Flows.

| Time series length | Batch size | FFlows train time (sec) | FFlows inference time (sec) |
|---|---|---|---|
| 80 | 1024 | 12.5 | 0.34 |
| 320 | 256 | 8.14 | 0.10 |
| 1280 | 64 | 6.24 | 0.08 |
| 5120 | 16 | 10.19 | 1.15 |
| 20480 | 4 | 93.72 | 39.24 |

## 1.4 ANALYSIS OF THE DIMENSION OF $Z_T$, $Z_S$ AND $Z_R$

The results of varying the value of the three latent dimensions are shown in Table 3 and Table 4, where the we report $Z\_T\_S\_R$ (e.g., $Z\_4\_6\_8$ represents $Z_T = 4$, $Z_S = 6$ and $Z_R = 8$) .

Table 3: Results for different values of $Z$ on Stock 24 dataset.

|  | $Z\_2\_3\_3$ | $Z\_4\_6\_6$ | $Z\_10\_15\_15$ | $Z\_40\_60\_60$ | $Z\_100\_150\_150$ |
|---|---|---|---|---|---|
| *Reconstruction* |  |  |  |  |  |
| MAE (train) | 0.0091 | 0.0096 | 0.0069 | 0.0063 | 0.0067 |
| MAE (test) | 0.0093 | 0.0104 | 0.0082 | 0.0076 | 0.0080 |
| *Generation* |  |  |  |  |  |
| Pred score | 0.037±0.000 | 0.038±0.000 | 0.037±0.000 | 0.036±0.000 | 0.037±0.000 |
| Discr score | 0.061±0.034 | 0.143±0.079 | 0.054±0.028 | 0.215±0.032 | 0.382±0.034 |

Table 4: Results for different values of Z on Stock 72 dataset.

|  | $Z\_2\_3\_3$ | $Z\_4\_6\_6$ | $Z\_10\_15\_15$ | $Z\_40\_60\_60$ | $Z\_100\_150\_150$ |
|---|---|---|---|---|---|
| *Reconstruction* |  |  |  |  |  |
| MAE (train) | 0.0089 | 0.0088 | 0.0093 | 0.0072 | 0.0073 |
| MAE (test) | 0.0090 | 0.0089 | 0.0096 | 0.0074 | 0.0074 |
| *Generation* |  |  |  |  |  |
| Discr-score | 0.046±0.032 | 0.044±0.017 | 0.028±0.027 | 0.025±0.019 | 0.035±0.034 |
| Pred-score | 0.186±0.002 | 0.184±0.001 | 0.185±0.001 | 0.185±0.001 | 0.185±0.001 |

## 1.5 RECONSTRUCTION EVALUATION

Below we report the reconstruction error on train and test set for iHyper-Time in Table 5, and the plots for some qualitative results in Figure 6 and Figure 7.

Table 5: Reconstruction errors for iHT on train and test splits.

|              | Energy24 | Stock72 | Stock24 | Stock360 |
| ------------ | -------- | ------- | ------- | -------- |
| MAE (train)  | 0.0273   | 0.0093  | 0.0069  | 0.0112   |
| MAE (test)   | 0.0370   | 0.0096  | 0.0082  | 0.0114   |

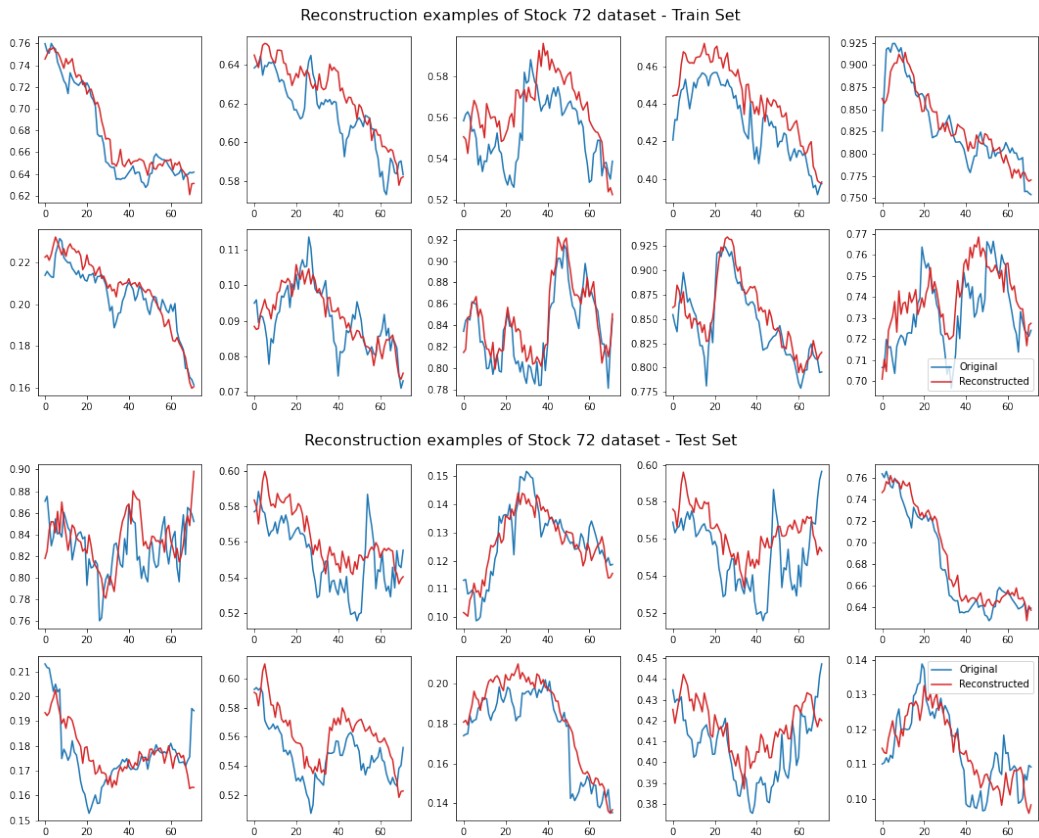

Figure 6: Reconstruction plots for Stock72 dataset. Top corresponds to train set and bottom to test set.

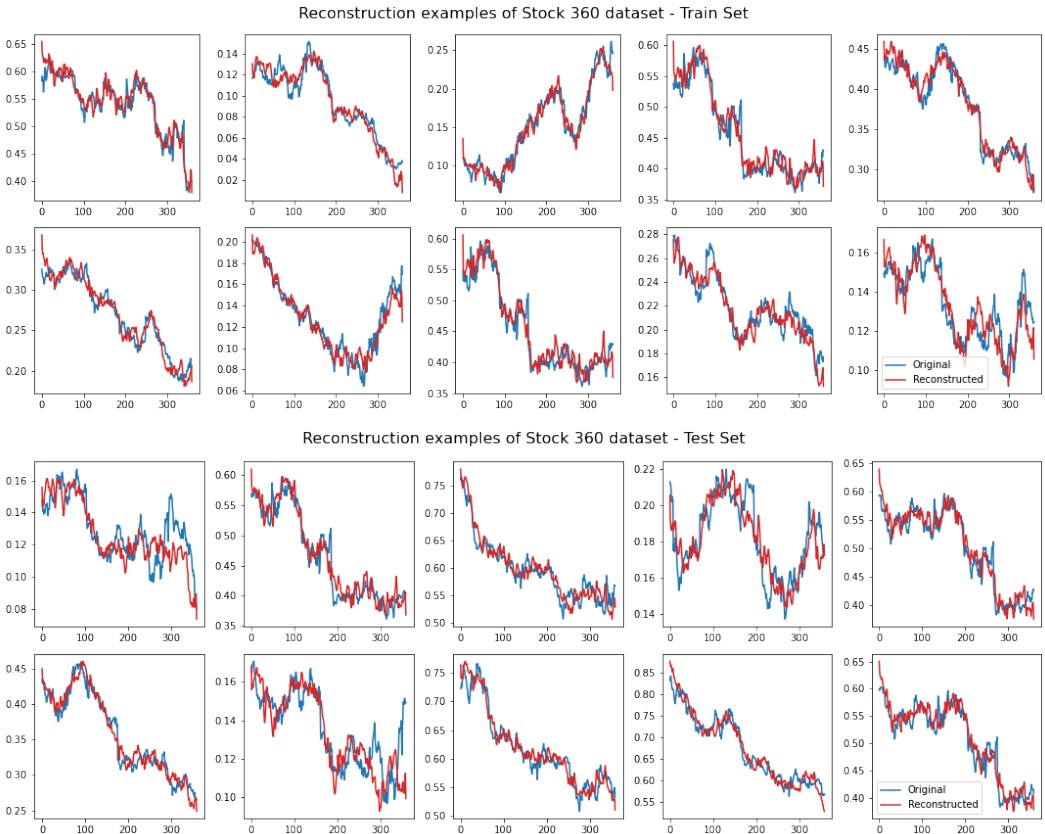

Figure 7: Reconstruction plots for Stock360 dataset. Top corresponds to train set and bottom to test set.

## 1.6 REAL-DATA INTERPRETABLE DECOMPOSITION

In this section we report an analysis on three real world datasets to show that iHT is able to disambiguate between trend, season and residual components after training. We used two well known real-world time series datasets that exhibit strong seasonality and trend, Atmospheric CO2, and Airline Passenger, and used the Stock72 dataset described in section B of the supplementary material that exhibits strong trend and weak seasonality.

We used two well known real-world time series datasets that exhibit strong seasonality and trend: Atmospheric CO2 and Airline Passenger, and used the Stock72 dataset described in section B of the supplementary material that exhibits strong trend but low seasonality.

The Atmospheric CO2 dataset corresponds to monthly data (January 1959 to December 1987) with a total of 348 observations with a seasonality period of 12 [1]. We sliced the data into time series of sequence length 60, obtaining a total of 288 time series.

The Airline Passenger dataset corresponds to monthly data (January 1948 to December 1960) with a total of 144 observations with a seasonality period of 12 [2]. We sliced the data into time series of sequence length 36, obtaining a total of 108 time series.

To estimate the presence of trend and seasonality of each dataset we use two metrics defined in [3]: the strength of trend and strength of seasonality. As these datasets do not include ground-truth trend and seasonality, we estimate them using STL [1]. Given a time series with additive decomposition in trend, seasonality and residual: $y_t = T_t + S_t + R_t$ , we can define the strength of trend as:

$$F_T = \max\left(0, 1 - \frac{\text{Var}(R_t)}{\text{Var}(T_t + R_t)}\right)$$

This measures the relative variance of the remainder component $R_t$ to the variance of the trend with the remainder component. The value ranges from 0 to 1, with 0 indicating no trend and 1 indicating a strong trend.

Strength of seasonality is defined by:

$$F_S = \max\left(0, 1 - \frac{\text{Var}(R_t)}{\text{Var}(S_t + R_t)}\right)$$

Which measures the relative variance of the remainder component $R_t$ to the variance of the seasonality with the remainder component. A value close to 0 indicates little to no seasonality, and a value close to 1 indicates strong seasonality.

Table 6 shows these metrics for the three datasets. As expected, Stock72 doesn't exhibit strong seasonality, while the other two datasets exhibit strong trends and seasonality, with all values on average above 0.98.

Table 6: Strength of trend and seasonality in the CO2, Air Passanger and Stock72 datasets.

|  | CO2 | Air Pass | Stock72 |
| --- | --- | --- | --- |
| Trend strength | $0.99 \pm 0.01$ | $0.98 \pm 0.01$ | $0.99 \pm 0.01$ |
| Seasonal strength | $0.99 \pm 0.00$ | $0.99 \pm 0.01$ | $0.32 \pm 0.09$ |

We trained iHT on each dataset and we show in Figures 8 and 9 of the Supplementary the reconstructed time series and the output of each of the individual blocks for the CO2 and Air Passanger datasets, respectively. For the output of the trend, seasonality and residual block, we compare with the STL method. We observe that there is a good agreement in the trend and seasonality components in both datasets. The STL method requires as parameter the period of the seasonality, which is known for both datasets, while our approach is able to estimate the seasonality from the data. In the case of the Stock72 dataset, for the STL comparison, we estimated the period using a Fourier transform and retrieving the dominant frequency. Figure 10 of the supplementary shows the decomposition generated by iHT and compared against STL. We can see in this case that the seasonal component is quite small in magnitude compared to the residuals.

[1] Cleveland, R. B., Cleveland, W. S., McRae, J. E. & Terpenning, I. (1990). STL: A Seasonal-Trend Decomposition Procedure Based on Loess (with Discussion). Journal of Official Statistics.

[2] Downloaded from `https://www.kaggle.com/datasets/rakannimer/air-passengers`

[3] Wang, X., Smith, K. A., & Hyndman, R. J. (2006). Characteristic-based clustering for time series data. Data Mining and Knowledge Discovery, 13(3), 335–364.

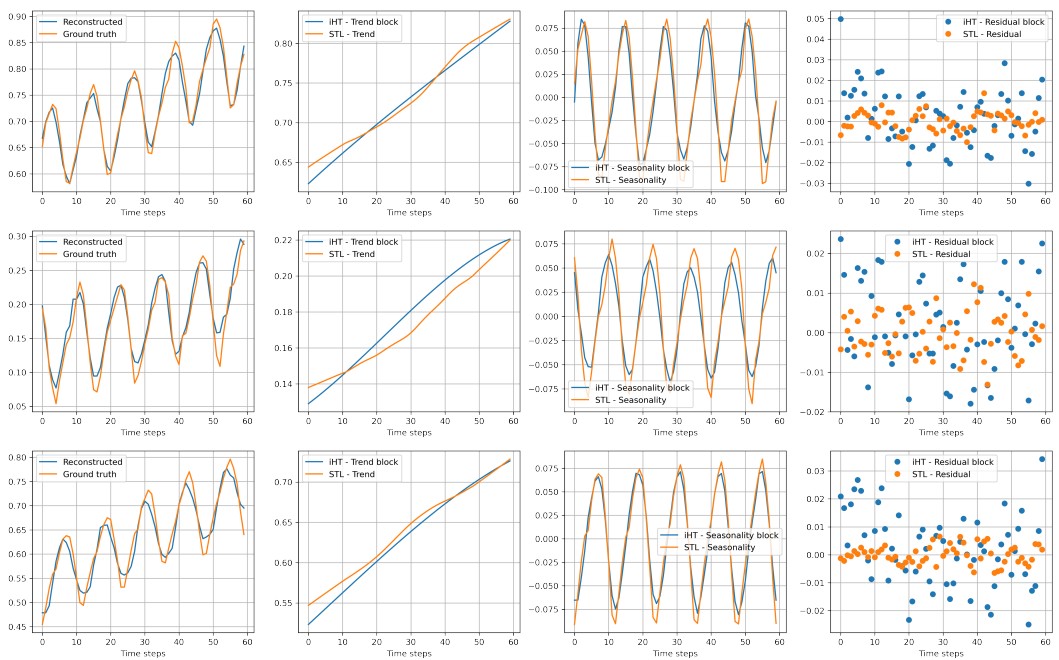

Figure 8: iHT decomposition on CO2 dataset.

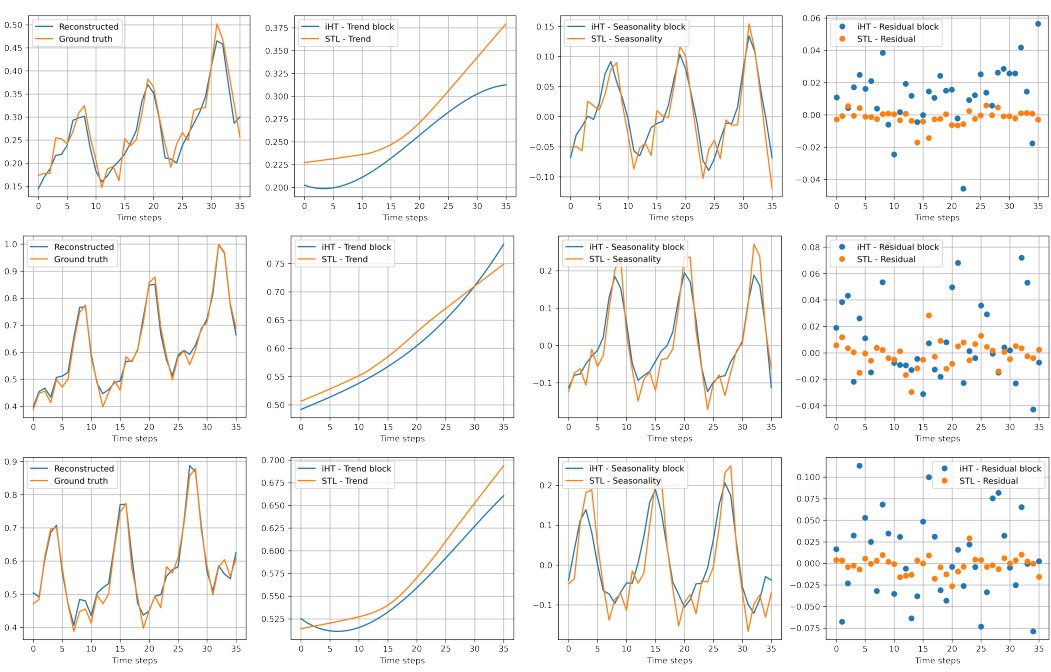

Figure 9: iHT decomposition on Air travel dataset.

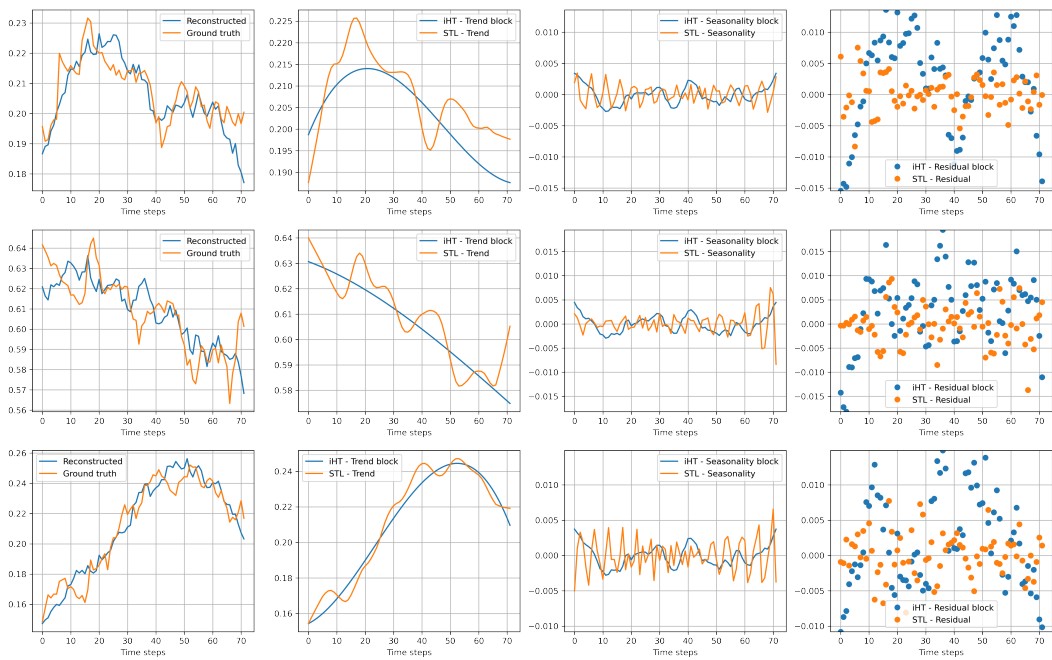

Figure 10: iHT decomposition on Stock72 dataset.

## 1.7 ANALYSIS OF $w0$

In Tables 7 and 8, we report the reconstruction and generation metrics for the Stock24 and Stock72 datasets. In Stock24, the best reconstruction and generation metrics are obtained with $w_0 = 30$. In the case of Stock72, the best reconstruction metric is also obtained with $w_0 = 30$, but the discriminative and predictive scores show a slight improvement for $w_0 = 100$.

Table 7: iHT with different values of $w_0 \in [5, 10, 30, 100, 300]$ for Stock24 dataset.

|  | $w_0 = 5$ | $w_0 = 10$ | $w_0 = 30$ | $w_0 = 100$ | $w_0 = 300$ |
|---|---|---|---|---|---|
| MAE (train) | 0.0119 | 0.0119 | 0.0069 | 0.0243 | 0.1851 |
| MAE (test) | 0.0120 | 0.0121 | 0.0082 | 0.0245 | 0.1788 |
| Discr-score | 0.151±0.059 | 0.212±0.044 | 0.054±0.028 | 0.087±0.066 | 0.489±0.007 |
| Pred-score | 0.041±0.001 | 0.040±0.001 | 0.037±0.000 | 0.037±0.000 | 0.059±0.002 |

Table 8: iHT with different values of $w_0 \in [5, 10, 30, 100, 300]$ for Stock72 dataset.

|  | $w_0 = 5$ | $w_0 = 10$ | $w_0 = 30$ | $w_0 = 100$ | $w_0 = 300$ |
|---|---|---|---|---|---|
| MAE (train) | 0.0156 | 0.0115 | 0.0093 | 0.0182 | 0.2052 |
| MAE (test) | 0.0157 | 0.0116 | 0.0096 | 0.0187 | 0.1963 |
| Discr-score | 0.045±0.034 | 0.058±0.038 | 0.028±0.027 | 0.026±0.011 | 0.250±0.037 |
| Pred-score | 0.188±0.002 | 0.186±0.001 | 0.185±0.001 | 0.183±0.001 | 0.196±0.000 |

## 1.8 ANALYSIS OF FFN

We conducted an ablation study on iHT using Random Fourier Features (FFN) instead of a SIREN network for the residuals. Given the importance of the scale factor in FFN as explained in [1], we evaluated FFN with multiple scale factor values: 5, 10, 100. Tables 9 and 10 show the reconstruction and generation metrics for the Stock24 and Stock72 datasets, respectively. For the Stock24 dataset,

SIREN achieves the best reconstruction performance as well as the lowest discriminative score. In the case of Stock72, FFN with scale factor 10 achieves better reconstruction errors, and shows slightly better performance in discriminative score than SIREN network.

Table 9: iHT with SIREN layers and with FFN with different scales ($\sigma \in [5, 10, 100]$) for Stock24 dataset.

|  | SIREN ($w_0 = 30$) | FFN ($\sigma = 5$) | FFN ($\sigma = 10$) | FFN ($\sigma = 100$) |
|---|---|---|---|---|
| MAE (train) | 0.0069 | 0.0094 | 0.0095 | 0.0101 |
| MAE (test) | 0.0082 | 0.0094 | 0.0095 | 0.0100 |
| Discr-score | 0.054±0.028 | 0.244±0.162 | 0.346±0.169 | 0.36±0.183 |
| Pred-score | 0.037±0.000 | 0.036±0.000 | 0.036±0.000 | 0.036±0.000 |

Table 10: iHT with SIREN layers and with FFN with different scales ($\sigma \in [5, 10, 100]$) for Stock72 dataset.

|  | SIREN ($w_0 = 30$) | FFN ($\sigma = 5$) | FFN ($\sigma = 10$) | FFN ($\sigma = 100$) |
|---|---|---|---|---|
| MAE (train) | 0.0093 | 0.0078 | 0.0085 | 0.0083 |
| MAE (test) | 0.0096 | 0.0079 | 0.0086 | 0.0082 |
| Discr-score | 0.028±0.027 | 0.028±0.014 | 0.024±0.026 | 0.054±0.029 |
| Pred-score | 0.185±0.001 | 0.184±0.001 | 0.185±0.002 | 0.184±0.001 |

[1] Woo, Gerald et al. "Learning Deep Time-index Models for Time Series Forecasting." International Conference on Machine Learning (2022).

## 1.9 UNCONDITIONAL GENERATION

In this section we discuss a possible extension of our current model to a purely unconditional one based on the VAE framework. To accomplish this, we assume a prior distribution over the embeddings $Z$ that is Gaussian:

$$Z \sim \mathcal{N}(Z|\mathbf{0}, \boldsymbol{I}) \tag{1}$$

Let $\mathcal{D} = \{(t_i, \mathbf{y}_i)\}_{i=1}^N$ denote an observed time series. In the variational autoencoder framework, model parameters are learned by maximizing the evidence lower-bound (ELBO):

$$\log p(\mathcal{D}) \geq \text{ELBO}(\mathcal{D}) = -\text{D}_{KL}(q_\phi(Z|\mathcal{D})\|p(Z)) + \mathbb{E}_{q_\phi(Z|\mathcal{D})}\left[\log p_\alpha(\mathcal{D}|Z)\right], \tag{2}$$

where $\text{D}_{\text{KL}}(q_\phi(Z|\mathcal{D})\|p(Z))$ denotes the Kullback-Leibler divergence between the variational posterior $q_\phi(Z|\mathcal{D})$ and the prior $p(Z)$ and $\mathbb{E}_{q_\phi(Z|\mathcal{D})}[\log p_\alpha(\mathcal{D}|Z)]$ denotes expected log likelihood of the observations given the embeddings (referred to as reconstruction error when one is minimizing the negative ELBO). The variational posterior is assumed to belong to the family of diagonal Gaussian distributions and can be parameterized as follows:

$$Z_i = g(t_i, \mathbf{y}_i) \tag{3}$$

$$\mu_Z = h_\mu\left(\bigoplus_{i=1}^N Z_i\right), \qquad \Sigma_Z = \text{diag}\left(h_{\sigma^2}\left(\bigoplus_{i=1}^N Z_i\right)\right) \tag{4}$$

$$Z|\mathcal{D} \sim \mathcal{N}(\mu_Z, \Sigma_Z), \qquad \text{(variational approximation)} \tag{5}$$

where $g$ is used to obtain a latent embedding for each $(t_i, \mathbf{y}_i)$ pair, $\oplus$ denotes a symmetric operation on the embeddings to combine them into a single embedding. Together, $g$ and $\oplus$ form the set encoder. The function $h_\mu$ is a neural network that maps from the combined embedding to the mean of the variational posterior and $h_{\sigma^2}$ is a neural network that maps from the combined embedding to the (diagonal) covariance matrix of the variational posterior. The variational posterior is chosen to be Gaussian so that obtaining an analytical expression of the Kullback-Leibler divergence is possible.

Under homoscedastic Gaussian assumptions on the likelihood of $\mathcal{D}$ given $Z$ with covariance matrix $\gamma^2 \boldsymbol{I}$, the expected log-likelihood is given by:

$$\log p_\alpha(\mathcal{D}|Z) = \sum_{i=1}^{N} -\frac{m}{2}\log(2\pi\gamma^2) - \frac{1}{2\gamma^2}\|\mathbf{y}_i - \hat{f}(t_i)\|_2^2, \qquad (6)$$

where $\hat{f}$ is the hyponetwork obtained by passing the embedding $Z$ through the decoder network. For fixed $\gamma$, maximizing $\log p_\alpha(\mathcal{D}|Z)$ is equivalent to minimizing the squared error $\sum_{i=1}^{N}\|\mathbf{y}_i - \hat{f}(t_i)\|_2^2$.

Similar to other VAE frameworks, the encoder and decoder can be trained using amortized inference, where the weights of the networks are shared between different time series datasets $\mathcal{D}_1, \mathcal{D}_2, \dots, \mathcal{D}_J$, each of which can contain a variable number of data points $N_j$ for $j = 1, \dots, J$. In the end, the parameters of the network are learned by minimizing:

$$\text{Loss}(\mathcal{D}_1, \dots, \mathcal{D}_J) = -\sum_{j=1}^{J}\text{ELBO}(\mathcal{D}_j) \qquad (7)$$

At test time, unconditional generation can easily be accomplished by sampling an embedding from the prior $Z \sim \mathcal{N}(\mathbf{0}, \boldsymbol{I})$ and passing it through the decoder to obtain corresponding hyponetworks for the trend, seasonality, and residual components. The time series can then be generated by passing in chosen time inputs into the instantiated TSNet.

## 1.10 iHyperTime with attentive set encoder

In this section we present an ablation study focusing on the impact of different set encoders within our iHypertime model. Specifically, we compare the performance of iHypertime when employing our original set encoder versus using Set Functions for Time series with attention (SeFT-attn), an attentive set encoder proposed by Horn et al. [1]. The objective is to assess how the choice of set encoder impacts the reconstruction and generative capabilities of iHypertime.

Table 11: Results for two different configurations of iHT with original set encoder and with SeFT-attn using Stock72 data.

|  | iHT - Set Encoder | iHT - SeFT-attn |
| --- | --- | --- |
| MAE (train) | 0.0093 | 0.0118 |
| MAE (test) | 0.0096 | 0.0120 |
| Discr-score | 0.028±0.027 | 0.038±0.026 |
| Pred-score | 0.185±0.001 | 0.187±0.002 |

In Table 11 we show the evaluation of iHT using the original set encoder and SeFT-attn. The table reports the quantitative metrics for the regular Stock72 dataset. We can see that iHT with the original set encoder shows slightly better performance both in reconstruction and generation metrics, but this could be due to now using the optimal set of hyperparameters for SeFT-attn. In Figure 11 we show the t-SNE visualization and the data distribution, for iHT with each set encoder.

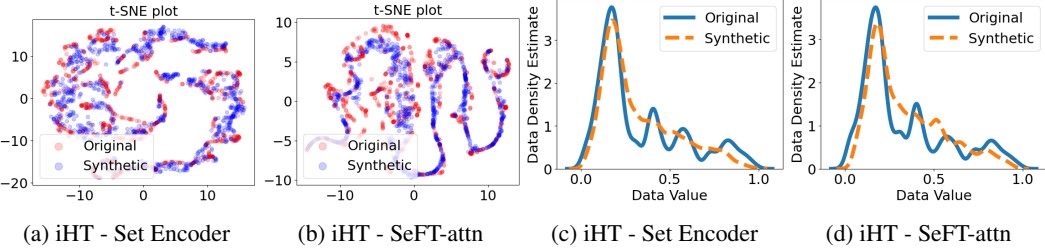

(a) iHT - Set Encoder     (b) iHT - SeFT-attn     (c) iHT - Set Encoder     (d) iHT - SeFT-attn

Figure 11: Left: t-SNE visualizations on Stock72 data. Right: Data distribution on Stock 72 data.

### 1.10.1 Implementation details

**Original set encoder** The original set encoder in iHT processes time series data by forming sets that aggregate time points and their corresponding multi-dimensional feature values. Specifically, for a time series with $N$ samples of $M$ features, the sets are of the form $S_{\text{original}} = \{(t_i, z_{i1}, \ldots, z_{im}) \mid i = 1 \ldots N, m = 1 \ldots M\}$. These sets are then input to a neural network composed of SIREN layers and each set produces an embedding. The embeddings from each set are averaged as explained in Section 3.2 in the paper. This averaging is a symmetric operation, meaning that the order of inputs does not affect the output, thus maintaining permutation invariance.

**SeFT-attn** The attentive set encoder first generates time embeddings of the time coordinate that are used to construct the sets and then uses an attention-based mechanism to aggregate the embeddings. Below are the main steps:
*Time Encoding*: Each time stamp $t_i$ is converted into a dense vector using a variation of positional encoding, allowing the model to interpret the temporal information in a multi-dimensional space.
*Set Construction*: The sets are constructed similarly to the original encoder but include the time-encoded vectors and the feature type. Each observation is treated as a tuple of the form $(t'_i, z_{im}, m)$, where $t'_i$ is the time-encoded representation, $z_{im}$ is the observed value, and $m$ is the feature index $m = 1 \ldots M$.
*Attention-based Aggregation*: Multi-head Attention is then employed to assign weights to each element in the set. The attention scores are used to compute a sum of the embeddings, instead of the averaging used in the original encoder.

For iHT we follow the same implementation as described in Section D.2 of the main paper. For the attentive set encoder (SeFT-attn), we based our implementation on the code publicly available in: `https://github.com/twkillian/EDICT`, and used the following hyper-parameters:

- num_timescales: 100
- encoder_mlp_width: 256
- encoder_mlp_dropout: 0.2
- encoder_mlp_depth: 3
- latent_width: 40
- attn_n_heads: 2
- attn_dropout: 0.2

[1] Horn, Max et al. "Set Functions for Time Series." International Conference on Machine Learning (2019).

## 1.11 Additional reconstruction plots

Figure 12 and Figure 13 show reconstruction results of iHT using the same y-axis scale for the Stock72 and Stock360 datasets, respectively.

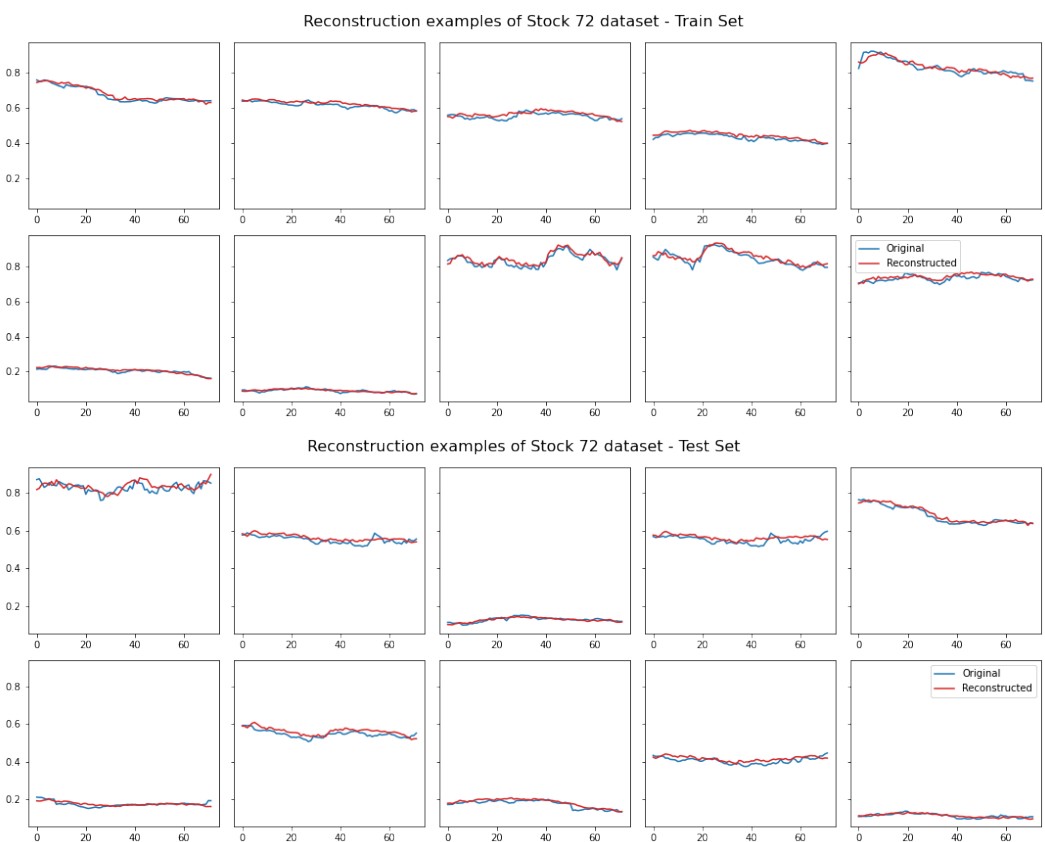

Figure 12: Reconstruction plots for Stock72 dataset. Top corresponds to train set and bottom to test set. All time series have the same y-axis.

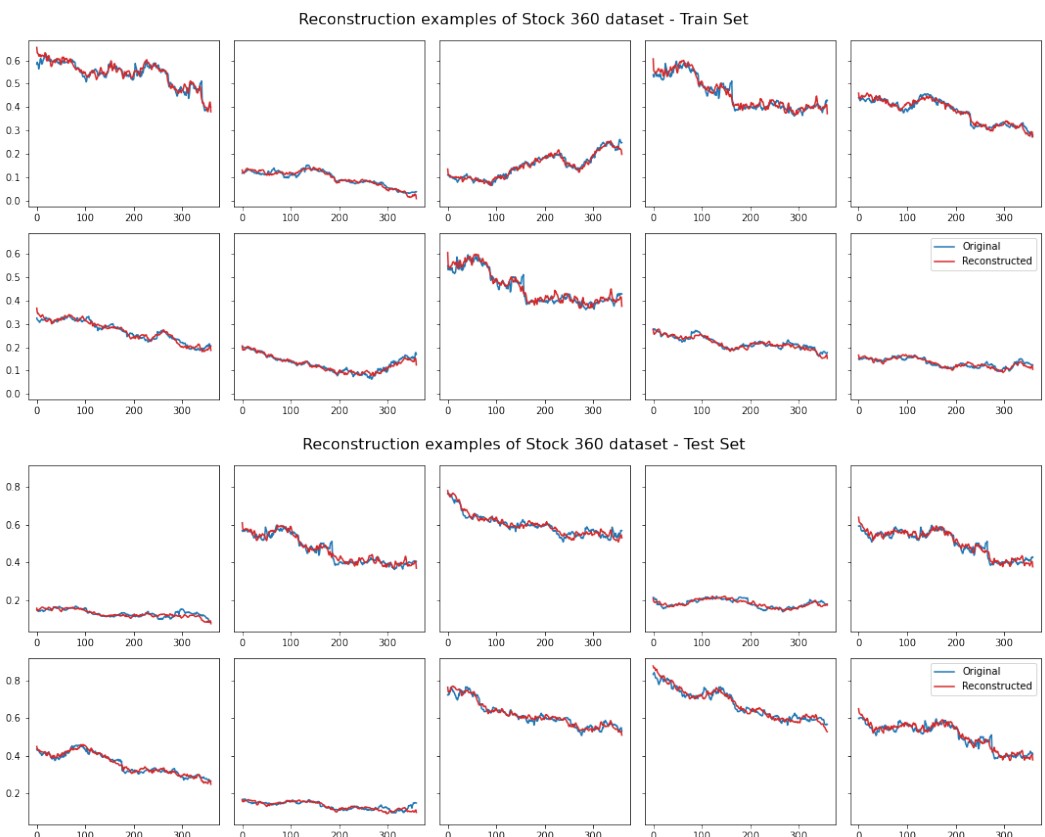

Figure 13: Reconstruction plots for Stock360 dataset. Top corresponds to train set and bottom to test set. All time series have the same y-axis.