# OpenReview forum: "iHyperTime: Interpretable Time Series Generation with Implicit Neural Representations"
_ICLR.cc/2024/Conference — Submitted to ICLR 2024_

### Official Review · Reviewer_y35Q · 2023-10-16

**Soundness:** 3 good
**Presentation:** 3 good
**Contribution:** 3 good
**Rating:** 5
**Confidence:** 4

**Summary:**

This paper uses INRs to synthesize time series data. The synthesized time series is an addition of three parts: trend, seasonality and residual components. A hyper network is used to generate parameters of each part given different data. The training of iHyperTime is performed in three stages, in order to improve stability

**Strengths:**

1. The idea is simple and should work well.
2. Experimental results are promising.
3. The model supports interpretability.

**Weaknesses:**

1. The author claims that "a single layer SIREN has been shown to be structurally similar to a Fourier mapped
perception (FMP)". I am not clear whether the author use SIREN (equation 4) or FMP (equation 5) to implement fr(t) in the paper.

2. If f(t) is multidimensional, does the author model each dimension as ftr(t) + fs(t) + fr(t) independently?

3. How does the proposed model achieve unconditional generation? The proposed model requires {t, f(t)} as input to synthesize time series. However, how can the model synthesize new samples without {t, f(t)}?

4. It would be better to cite Deep Sets when discussing the set encoder.

**Questions:**

None

---

> ### Author Response · Authors · 2023-11-18
>
> ## Weaknesses
> We thank the reviewer for their comments and suggestions. We hope they find our answers useful, and we are happy to answer any additional questions.
>
> ### **W.1.**
>
> We apologize for the lack of clarity in the description of the method. TSNet uses a SIREN to model the Residual component, and a Fourier decomposition to model the Seasonality component. We do not actually use a Fourier mapped perceptron (FMP), and we corrected the related section. In particular, we have rewritten all Section 3.1.1 to provide a more concise and clear explanation of the TSNet architecture.
>
> ### **W.2.**
>
> We thank the reviewer for raising this question. If f(t) is multidimensional, iHT models each dimension as $f_T(t) + f_S(t) + f_R(t)$ independently. Following an $m$-dimensional time series, the trend block outputs $m$ trends, the seasonality block outputs $m$ time series that correspond to the seasonality of each feature, and the same is for the residual block. Each of the features are then added together obtaining an $m$-dimensional time series.
>
> ### **W.3.**
>
> We apologize for the lack of clarity in the description of the generation mechanism. The model does require samples of time series to generate a new one. Generation of new time series is produced by randomly selecting pairs of time series, and performing linear a interpolation between their embeddings $Z^{(1)}$ and $Z^{(2)}$:
> $$
>         Z^{\text{gen}} = Z^{(1)} + \lambda (Z^{(2)} - Z^{(1)})
> $$
>     where $\\lambda$ is also sampled randomly. Optionally, the interpolation can be performed on individual components $Z_T$, $Z_S$, $Z_R$ of the embeddings, enabling the conditional generation of time series.
>
> In the current setting the model cannot synthesize new samples without {t, f(t)}. This could be achieved in a variational setting or by incorporating adversarial training, which we plan for a future work. We have now refactored Section 3.2 of the main paper to include more details on the generative process.
>
>
> ### **W.4.**
>
> We thank the reviewer for the suggested reference, we have now cited Deep Sets in the revised paper version, in the set encoder subsection in Section 3.2.

---

> > ### Comment · Reviewer_y35Q · 2023-11-20
> > **thanks for clarification**
> >
> > I believe W2 and W3 are two major weakness of the proposed method. For W2, this model doesn't consider the influence from other dimensions. For W3, this method can't achieve unconditional generation. Therefore I keep my score.

---

> > > ### Author Response · Authors · 2023-11-21
> > >
> > > We would like to thank the reviewer for the time dedicated to read our rebuttal.
> > >
> > >
> > > **Influence from Other Dimensions (W2).**
> > >
> > > We apologize for the lack of clarity in our previous reply. Across most of the network building blocks (embeddings, decoders, TSNet), the Trend-Seasonality-Residual (TSR) components $f_T(t)$, $f_S(t)$, $f_R(t)$ of the time series are processed separately, but the different dimensions (in the case of multivariate data) are computed and encoded together. Furthermore, in the first block of the architecture (set encoder) both the TSR components and the channels are processed together. This enables the network to generate embeddings that encode interactions between the different components, as well as between different signal channels. This is in contrast to other SOTA methods such as Fourier Flows [1], where individual channels are processed independently in multivariate time series. We will add a clarification of this point in the final version of the paper.
> > >
> > > [1] A. Alaa, A. Chan, M. van der Schaar. Generative time-series modeling with fourier flows. In ICLR, 2021.
> > >
> > > **Unconditional Generation Limitation (W3).**
> > >
> > > While we acknowledge the limitations of our work, we would like to emphasize that the conditional nature of our method allows us to construct interpretable counterfactuals of existing time series by interpolating between the sub-vectors of the embeddings (trend, seasonality and residuals). This kind of generation is analyzed in Appendix H, where we show the synthesis of new time series with a constraint on the trend component, and can be useful in a variety of settings.
> > >
> > > For example, in the domain of finance, algorithmic trading strategies are typically evaluated using back-testing, which involves running the strategy against observed historical data. Our method would allow the generation of new scenarios by generating new histories (e.g., COVID-19 seasonality patterns with alternative trends that have been observed in historical data), which can be used to better evaluate the robustness and performance of algorithmic trading strategies.
> > >
> > > We understand, however, that this important generation scenario is not stated clearly in the manuscript and that the capability of unconditional generation should be discussed more clearly. We will make the following two changes to the manuscript and appendix:
> > >
> > > - _Clarification for conditional generation:_ We will update the introduction and Section 3.2 to emphasize the conditional nature of the generation method. We will also include a limitations section to discuss the implications of conditional generation.
> > > - _Extension to variational autoencoder framework:_  In the supplemental material, we have included a discussion of a possible change of the current model to a purely generative (unconditional) one based on the VAE framework. We will incorporate this discussion as part of the appendix in the final version.
> > >
> > > While we acknowledge these limitations, we believe our work contributes significantly to the field of time series generation and we would like to highlight that our approach provides realistic time series samples with interpretable trend, seasonality, and residual. In particular, we demonstrated that our generation is not just interpretable, but we also provide generated time series that excel in terms of realism, computational time, and usefulness, w.r.t. state-of-art approaches, in a broad spectrum of scenarios.

---

### Official Review · Reviewer_3ERW · 2023-10-30

**Soundness:** 3 good
**Presentation:** 2 fair
**Contribution:** 3 good
**Rating:** 8
**Confidence:** 4

**Summary:**

The paper proposes to use implicit neural representations / coordinate-based neural networks to model time series data. The proposed method decomposes time series into season, trend, and residual components, and leverages a hypernetwork to predict the coefficients of the season, trend, and residual basis functions. The proposed method is applied on time series generation tasks.

**Strengths:**

The paper introduces an interesting new method for time series generation based on implicit neural representations with the added benefit of seasonal-trend-residual decomposition. Empirical and qualitative experiments show significant improvement over baselines.

**Weaknesses:**

The writing is currently very unclear and is unable to convey the critical ideas of the paper.

1. How is the model used to perform generation? Based on the writing, I am unable to understand how this formulation, which requires taking data as input with the set encoder, can be used for unconditional generation of new time series.
2. The paragraph on __Residual blocks__ in Section 3.1.1 does not make sense at all. Why is it describing the seasonal component, "In this work, we propose to model the seasonal component of the time series as a ...", which was already addressed in the previous paragraph? The notation is suddenly changed from $t$ to $x$? Why talk about random Fourier features if the paper proposes to use SIREN? I do not understand what is the implemented/proposed model for the residual component after reading this paragraph.
3. What are the details of the hypernetwork component/iHyperTime architecture? More details with mathematical description needs to be given for this part. How is the Set Encoder and SIREN layers used here? Details of the hypernetwork are not given.
4. Important details on problem formulation and training are not included. Concretely, what are the time coordinates $t_i$? For long sequences, is the whole time series encoded into a set, or divided into windows? Section 4.2 states "Performance results for regularly sampled time series are provided in Table 1, for univariate and multivariate datasets of varying lengths." -- which datasets are univariate, which are multivariate? What are the varying lengths? How do I interpret 24/72/360 in Table 1?

I would be happy to increase my score to "marginal accept" if writing is improved, and to "accept" if more evidence is presented on interpretability of real world data.

**Questions:**

1. Is the model actually able to disambiguate between season, trend, and residual in an interpretable manner for real world data? Can you visualize the seasonal-trend decomposition on real world datasets with strong seasonality/trend?

---

> ### Author Response · Authors · 2023-11-18
>
> We thank the reviewer for the suggestions. We acknowledge that the paper presentation was not clear enough. In the new version of the paper we extensively rewrite section 3, and we have colored the rewritten parts in blue to highlight the modifications/additions.
> As suggested, we have also evaluated our approach on three real datasets, showing that the model is able to disambiguate between season, trend, and residual in an interpretable manner for real world data.
>
> I hope the reviewer finds our results useful in assessing the quality of our work. We are happy to answer any additional questions.
>
> ## **Weaknesses**
>
> ### **W.1**
>   We apologize for the lack of clarity in the description of the generative method. In the main paper we have now added a subsection at the end of section 3.2 explaining how we perform generation. Generation of new time series is produced by randomly selecting pairs of time series from the training-set, and performing a linear interpolation between their embeddings $Z^{(1)}$ and $Z^{(2)}$:
>     $$ Z^{\\text{gen}} = Z^{(1)} + \\lambda (Z^{(2)} - Z^{(1)}) $$
> where $\\lambda$ is also sampled randomly. Optionally, the interpolation can be performed on individual components $Z_T$, $Z_S$, $Z_R$ of the embeddings, enabling the conditional generation of time series.
>
>
> ### **W.2**
> We apologize for the mistake in explaination of the seasonal component: when we said "seasonal" we should have said "residual".
>
> **Change of notation** In the trend and seasonality networks, we used $t$ because there is a single layer. And in residual, we used $x$ to indicate the outputs of the successive layers. We should have clarified this, also stating that $x_0 = t$. We have now rewritten this section ensuring consistency of the notation.
>
> **Random Fourier Features** We apologize for the lack of clarity in this aspect. Our initial goal was to convey that using SIRENs or random Fourier Features was equivalent w.r.t. having a good representation of frequencies in the signal, following [1]. However, we agree that this leads to confusion. We have now revised Section 3.1.1 to clarify this and we removed any misleading statements about Fourier Features Networks (FFNs).
>
> [1] Benbarka, Nuri et al. “Seeing Implicit Neural Representations as Fourier Series.” 2022 IEEE/CVF Winter Conference on Applications of Computer Vision (WACV) (2021).
>
>
> ### **W.3**
> We thank the reviewer for pointing out an important missing aspect. We have now included two subsections in Section 3.2 detailing how the set encoder and hypernetworks are used. For the Set encoder we have provided a mathematical description and how it is parameterized using SIREN layers. We have provided implementation details in Section D.2 of the appendix, which is now referred in the main paper. Furthermore, in section D.2 we have also added more information on how the Set Encoder is used, and provided more details about the hypernetwork architecture.

---

> ### Author Response · Authors · 2023-11-18
>
> ### **W.4.**
>
> We thank the reviewer for pointing out aspects that were not clearly explained in the paper.
>
> **Time coordinate.** In our model, we use a single floating point value as temporal coordinate (time $t$). As data pre-processing, all the values are scaled to the interval $[-1, 1]$, with a common global factor for all time series of a dataset. In all cases, regardless of sequence length, the time series is fed to the set encoder as a single set, and is hence converted into a single embedding $Z$. We have now included this information in Appendix D.2.
>
> **Datasets.** Following previous methods [1, 2, 3], we used the Stock and Energy datasets [5] sliced into 24 time-steps windows. These datasets are multivariate, with 6 and 28 features, respectively. Furthermore, in order to make a fair comparison with  [3] and  [4], who show performances in longer time series of Stock data, we generated an univariate version of the Stock dataset by selecting one single feature (Close Price), and slicing it in two ways: using a window of 72 time steps (Stock72), and a window of 360 time steps (Stock 360).
>
> Additionally, in Section 4.4 we consider 4 univariate datasets with longer real-world time-series from Monash repository [6], namely FRED-MD, NN5 Daily, Temperature Rain, and Solar Weekly. The first three datasets have time series of approx. $750$ time steps, while the last one contains time series of length $52$. This information is summarized in Appendix B, where Table 8 presents the main characteristics of each dataset. We have improved this table to clarify the difference between the different datasets, in terms of sequence length and number of channels. And we refer this table in Section 4.1 of the main paper.
>
> Finally, in Section 4.3 we evaluate our method on irregular data. Following existing literature [2], we create irregular time-series datasets by randomly dropping 30, 50, 70\% of observations from the original datasets. We select each observation to drop uniformly at random, and independently for each time series $\{(t_i, y_i)\}_{i=0}^N$ in the original dataset.
>
> - [1] J. Yoon, D. Jarrett, M. van der Schaar. Time-series generative adversarial networks. In NeurIPS, 2019.
>
> - [2] J. Jeon, J. Kim, H. Song, S. Cho, N. Park. GT-GAN: General purpose time series synthesis with generative adversarial networks. In NeurIPS, 2022.
>
> - [3] A. Coletta, S. Gopalakrishan, D. Borrajo, S. Vyetrenko. On the constrained time-series generation problem. In NeurIPS, 2023.
>
> - [4] A. Alaa, A. Chan, M. van der Schaar. Generative time-series modeling with fourier flows. In ICLR, 2021.
>
> - [5] L. Candanedo, V. Feldheim, D. Deramaix. Data driven prediction models of energy use of appliances in a low-energy house. Energy and buildings, 140:81–97, 2017.
>
> - [6] R. Godahewa, C. Bergmeir, G. Webb, R. Hyndman, P. Montero-Manso. Monash time series forecasting archive. In NeurIPS, 2021.

---

> ### Author Response · Authors · 2023-11-18
>
> ## Questions
>
> ### **Q.1**
>
> We thank the reviewer for suggesting an interesting new experiment. We have now conducted an analysis on three real world datasets to show that iHT is able to disambiguate between trend, season and residual components after training.
>
> The new experiments are included in Section 1.6 of a new supplementary material, submitted along the main paper. We will add the experiments in the final version of the paper.
>
> **Datasets**
>
> We used two well known real-world time series datasets that exhibit strong seasonality and trend: Atmospheric CO2 and Airline Passenger, and used the Stock72 dataset described in section B of the supplementary material that exhibits strong trend but low seasonality.
>
> The Atmospheric CO2 dataset corresponds to monthly data (January 1959 to December 1987) with a total of 348 observations with a seasonality period of 12 [1]. We sliced the data into time series of sequence length 60, obtaining a total of 288 time series.
>
> The Airline Passenger dataset corresponds to monthly data (January 1948 to December 1960) with a total of 144 observations with a seasonality period of 12 [2]. We sliced the data into time series of sequence length 36, obtaining a total of 108 time series.
>
> To estimate the presence of trend and seasonality of each dataset we use two metrics defined in [3]: the strength of trend and strength of seasonality. As these datasets do not include ground-truth trend and seasonality, we estimate them using STL [1]. Given a time series with additive decomposition in trend, seasonality and residual:
>     $
>         y_t = T_t+ S_t + R_t
>    $
>  , we can define the strength of trend as:
> $$
>         F_T = \max\left(0, 1 - \frac{\text{Var}(R_t)}{\text{Var}(T_t+R_t)}\right)
> $$
>
> This measures the relative variance of the residual component $R_t$ to the variance of the trend with the residual component. The value ranges from 0 to 1, with 0 indicating no trend and 1 indicating a strong trend.
>
> Strength of seasonality is defined by:
> $$
>         F_S = \\max\\left(0, 1 - \\frac{\\text{Var}(R_t)}{\\text{Var}(S_t+R_t)}\\right)
> $$
>  Which measures the relative variance of the residual component $R_t$ to the variance of the seasonality with the residual component. A value close to 0 indicates little to no seasonality, and a value close to 1 indicates strong seasonality.
>
> Table below shows these metrics for the three datasets. As expected, Stock72 doesn't exhibit strong seasonality (as stock data usually have not such component), while the other two datasets exhibit strong trends and seasonality, with all values on average above 0.98.
>
> $$
> \\begin{array}{cccc}
> \\hline
> \\text{Component} & \\text{CO2} & \\text{Air Pass} & \\text{Stock 72} \\\\ \\hline
>     \\text{Trend strength}   & 0.99 \\pm 0.01 & 0.98 \\pm0.01 & 0.99 \\pm 0.01 \\\\
>         \\text{Seasonal strength} &  0.99 \\pm 0.00 & 0.99 \\pm0.01  & 0.32 \\pm 0.09 \\\\
> \\hline
> \\end{array}
> $$
>
> **Results**
>
> We trained iHT on each dataset and we show in Figures 8 and 9 of the Supplementary the reconstructed time series and the output of each of the individual blocks for the CO2 and Air Passanger datasets, respectively. For the output of the trend, seasonality and residual block, we compare with the STL method. We observe that there is a good agreement in the trend and seasonality components in both datasets. The STL method requires as parameter the period of the seasonality, which is known for both datasets, while our approach is able to estimate the seasonality from the data. In the case of the Stock72 dataset, for the STL comparison, we estimated the period using a Fourier transform and retrieving the dominant frequency. Figure 10 of the supplementary shows the decomposition generated by iHT and compared against STL. We can see in this case that the seasonal component is quite small in magnitude compared to the residuals.
>
> We will include this new analysis on real-world data in the revised version of the manuscript.
>
> [1] Cleveland, R. B., Cleveland, W. S., McRae, J. E. & Terpenning, I. (1990). STL: A Seasonal-Trend Decomposition Procedure Based on Loess (with Discussion). Journal of Official Statistics.
>
> [2] Downloaded from https://www.kaggle.com/datasets/rakannimer/air-passengers
>
> [3] Wang, X., Smith, K. A., & Hyndman, R. J. (2006). Characteristic-based clustering for time series data. Data Mining and Knowledge Discovery, 13(3), 335–364.

---

> ### Comment · Reviewer_3ERW · 2023-11-20
>
> Thanks for the extensive response. I have raised my score since my queries have been addressed, and I think this work introduces a novel method with good performance.
>
> However I do have a few comments:
> 1. Section 3.2 is titled "TIME SERIES GENERATION WITH IHYPERTIME" but is still addressing architecture and training, only addressing generation in the last paragraph. This section/title should be changed to something more appropriate.
> 2. It turns out the model can only do conditional generation. This was not explained in the original manuscript, and in the updated manuscript, it is only mentioned once in the middle - this needs to be highlighted more clearly in the introduction, and preferably the abstract as well.
> 3. Furthermore, it is clear that the proposed method would outperform baselines on the evaluation metrics being used, due to the fact that it is a conditional generation method. This should be highlighted in a limitations section.

---

> > ### Author Response · Authors · 2023-11-21
> >
> > We would like to sincerely thank the reviewer for re-evaluating our manuscript and for the constructive feedback. We are pleased to see that our responses and clarifications have positively impacted the assessment of our work.
> >
> > **Title of Section 3.2.** We appreciate the observation regarding the content and title of Section 3.2. We will change the title of the section accordingly.
> >
> > **Conditional Generation Clarification.** We will update the abstract and introduction of the manuscript to highlight the conditional nature of our method. We will also include a limitations section where we discuss the implications of conditional generation on our method’s performance. This section will also outline potential areas for future work to extend the capabilities of our method beyond conditional generation.
> >
> > We believe these changes will enhance the clarity and completeness of our manuscript. We would like to thank the reviewer again for the feedback, which has significantly contributed to improve our work.

---

### Official Review · Reviewer_h6ka · 2023-10-30

**Soundness:** 2 fair
**Presentation:** 2 fair
**Contribution:** 3 good
**Rating:** 5
**Confidence:** 4

**Summary:**

This paper deals with the generation of time series by employing implicit neural representation (INR) networks. The main advantage of INR lies in handling irregularly sampled grids of different sizes. To avoid training one INR per instance, the INR weights are modulated through a hypernetwork based on a permutation invariant set encoder. The INR consists of three blocks (trend, seasonality, and residual blocks) that are additively recombined to reconstruct the time series. Each block is modulated by a different portion of the latent vector (the output of the set encoder), which is passed through its corresponding decoder. Once the model is trained, a new time series can be generated by linearly interpolating two existing latent vectors (corresponding to two different samples). Then, this new latent vector $z^{(gen)}$ modulates the trained INR to characterize a new time series $x^{(gen)}_{t}$ that can be queried for any timestamp $t$. The approach is tested against several baselines on regular, irregular, and long-time series datasets, and quantitative and qualitative evaluations are presented. In addition, the model allows for a trend-seasonality decomposition analysis.

**Strengths:**

- S1. IHyperTime can handle irregularly sampled time series allowing the model to learn robust representations across series with different grid measurement. Once trained, iHyperTime can generate new time series,  at any timestamp $t$, opening interesting applications.

- S2. The model follows a classic time series decomposition, which allows for some degree of interpretability and for some constrained generation on some of the factors, for instance the trend.

- S3. Extensive experiments are conducted in the paper. In addition, plenty of qualitative analysis of the model are available in the appendix, leading to a better comprehension of the model behavior. In addition, the experiments show competitive results compared to the baselines.

**Weaknesses:**

Major weaknesses

- W.1. Time series generation is done by interpolating linearly latent codes of two existing series, thus limiting the diversity and expressiveness of the generated time series. Moreover, it would be interesting to understand qualitatively what happened in the latent space when interpolating linearly between two instances (please see Q.1.).

- W.2. The results are good at highlighting that the generated time series is faithful to the original distribution through the \textit{predictive}, \textit{discriminative}, and \textit{marginal} scores, but there is no qualitative or quantitative experiment on the diversity of the generated time series. My concerns comes from the generation procedure. If $z^{(gen)} = 0.9 z^{(1)} + 0.1 z^{(2)}$, the new time series $x^{(gen)}$ would be very likely $x^{(1)}$ leading to good fidelity scores but not really generating novel time series. Moreover it is standard in generation to split the time series into train/test datasets before the generation phase. Then, generate some new $x^{(gen)}$ according to $z^{(i)}$'s from the train dataset and then compute the \textit{predictive}, \textit{discriminative}, and \textit{marginal} scores between the $x^{(gen)}$ and the $x^{(j)}$'s from the test dataset. From the evaluation metrics paragraph, it is unclear if the dataset is split before generation.

Minor weaknesses

- MW1. The point arguing that SIREN outperforms Fourier Features Network (FFN) for learning and representing high frequencies is at my knowledge not true for time series. Indeed, classical Fourier Features where frequencies are sampled in the linear scale can suffer from spectral bias. But if the frequencies are sampled in the logarithm scale or drawn from a Gaussian distribution as in [2] the spectral bias doesn't stand anymore.
    In the context of DeepTime [3], the authors demonstrate the superiority of FFN over SIREN in time series forecasting.
- MW2. Figure 4 is not completely clear. More details are needed to understand how this figure was generated. At first glance, even if the method is efficient for different sequence lengths, the training and inference time should increase with the sequence length, at least due to the loading of the data. Moreover, it is quite peculiar to see a method, Fourier Flows, having a convex curve in time with the sequence length. This makes the argument on the independence with respect to the sequence length less convincing, even though the model still is efficient at training time as shown in Table 12.
- MW3. In Figures 14-20, the distribution of the original data is different for each baseline, this limits the conclusion that can be drawn from these experiments.
- MW4. The interpolation process for generating new time series is clear when reading the code, however it is not really described in the paper. This would be important for better clarity of the paper to have this description at least in the appendix.

- W.3. One key feature of the IHyperTime architecture is the permutation invariant set encoder which allows to encode time series with different sampled grids. However the set encoder is known for underfitting [1] because of the naive aggregation mechanism. And there is no metrics in the experiments allowing to understand the quality of the reconstruction. Moreover, the dimension of $Z_{T}, Z_{S}$ and $Z_{R}$ seems to be crucial and are not discussed in the paper (they are set to 10, 15, 15). It would be interesting to understand the trade-off between the quality of reconstruction and the quality of generation according to the dimension of the $Z's$ (please see Q.3).

[1]: Kim, H., Mnih, A., Schwarz, J., Garnelo, M., Eslami, A., Rosenbaum, D., ... Teh, Y. W. (2019). Attentive neural processes. arXiv preprint arXiv:1901.05761.

[2]: Tancik, M., Srinivasan, P., Mildenhall, B., Fridovich-Keil, S., Raghavan, N., Singhal, U., ... Ng, R. (2020). Fourier features let networks learn high frequency functions in low dimensional domains. Advances in Neural Information Processing Systems, 33, 7537-7547.

[3]: Gerald Woo, Chenghao Liu, Doyen Sahoo, Akshat Kumar, and Steven Hoi. Deeptime: Deep time-
index meta-learning for non-stationary time-series forecasting. arXiv preprint arXiv:2207.06046,
2022.

**Questions:**

- Q.1. It would be interesting for a given latent vector $z^{(1)}$ and another latent vector $z^{(2)}$ to compute  $z_{λ}^{(gen)}$ for $λ$ in $\{0, 0.1, 0.2, 0.3, ..., 0.9, 1\}$ and then to visualize $x^{(gen)}_{λ}$ for each $λ$.

- Q.2. Please refer to W.2.

- Q.3. It would also be nice to have an ablation on the modulation mechanism (other than the classical set encoder) for instance the attentive set encoder proposed by Max Horn et al. in [set functions for times series]. In addition, it would be interesting to see the effect of the dimension of $Z_{T}, Z_{S}$ and $Z_{R}$ on the reconstruction loss and on the generation quality.

- Q.4. As describe in MW1, the argument of using SIREN over FFN is not theoretically true. It would then be interesting to see an ablation study on SIREN vs FFN. In the same spirit, no ablation study is performed on $w_0$, which is a crucial and sensitive hyper-parameter of SIREN.

- Q.5. Could you give some quantitative and qualitative results on the reconstruction quality using IHyperTime?

- Q.6. Why is the marginal score is used only on the four Monash datasets ?

- Q.7. Could you comment the results of Table 5 for the Temp Rain dataset? It seems strange that LS4 has such a low marginal score while IHyperTime outperforms greatly LS4 in the other two metrics. Overall this table needs to be more commented. %Why not use the same notations for the metrics in Tables 1-4 and Table 5?

- Q.8. In Tables 1-2, could you explain why DiffTime discriminative score greatly improves with $30\%$ missing data ? and then decreases again with $50\%$  and $70\%$ missing data. This looks like an anomaly.

- Q9. Why is DiffTime not implemented on the four Monash datasets (cf Table 11)?

---

> ### Author Response · Authors · 2023-11-16
>
> We would like to sincerely thank the reviewer for their effort and interesting comments. We now included an additional supplementary material along the main paper. The supplementary contains our new experiments to address reviewer's concerns.
>
> ## **Major Weaknesses**
>
> ### **W.1. / Q.1.**
>
> In Figure 1 and Figure 2 we show the reconstructed time-series from interpolations between pairs of embeddings generated by our architecture on the Stock72 dataset, and Figures 3 and 4 correspond to the Stock 360 dataset.
> The figures show the time-series reconstruction and the embeddings interpolation for a given latent vector z1 and another latent vector z2, at varying $\lambda \in [0, \ldots, 1]$ (as requested in Q.1).
>
> In the top row, we show a smooth transition from $z_1$ to $z_2$, in terms of the fully reconstructed time series. Our architecture also performs a TSR decomposition of the time series, and hence in the three bottom rows we display the smooth transition between $z_1$ and $z_2$ of each individual component of the time series (Trend, Seasonality, Residual).
>
> We will add these plots in the main paper. We better discuss the diversity of generated time-series in the next comment.
>
>
> ### **W.2. / Q.2.**
>
> **Train-Test Split.**
>  We agree that splitting the time series into train/test is the best approach for a fair evaluation of the generative models. However, we found that some of state-of-art approaches train and test the model on the whole dataset for Stock and Energy data[1,2,3]; and to perform a consistent comparison w.r.t. the performance reported in their original paper, we decided to keep the same approach in our paper.
> We however agree with the reviewer's concern, and thus we are happy to report in the table below (also Table 1 of supplementary) our initial effort in evaluating approaches while randomly splitting data between training set (i.e., 80$\\%$ of dataset) and test set (i.e., 20$\\%$ of dataset). The Table reports the quantitative metrics for the regular Stock data with length 24. While Figure 5 in the supplementary material shows the t-SNE visualization and the data distribution. These results confirm the findings from the our paper. Notice that, for the Monash datasets we already divided the dataset in training ($80\%$) and test ($20\%$) set, similar to the benchmark model LS4 [4].
>
> **Diversity.** Finally, we also agree that diversity of generated time-series plays a key role in the evaluation of generative models. We can qualitatively evaluate the diversity and coverage from t-SNE of Figure 5 (as well as in t-SNE plots of the main paper). The t-SNE plots show that iHyperTime synthetic data mostly overlap with real data: i.e., iHyperTime is able to have enough diversity to reconstruct the real data distribution. Following reviewer suggestion, we also use a novel metric to quantitatively evaluate the diversity of time-series, namely $sym$-Recall[6] which extends the $\\beta$-Recall from[5]. Such metric quantifies the diversity of time-series as the extent to which synthetic samples cover the full variability of real samples. Initial results are reported in the Table below.
>
> $\\begin{array}{l|ccccccc}
> \\hline
> \\text{Metric} & \\text{iHT (Ours)} & \\text{GT-GAN} & \\text{TimeGAN} &  \\text{DiffTime} & \\text{LS4} & \\text{FFlows} \\\\ \\hline
> \\text{Discr-score} \\downarrow & 0.054\\pm0.028 &  0.273\\pm0.046 & 0.068\\pm0.018 & 0.079\\pm0.014 & 0.154\\pm0.080 & 0.426\\pm0.032 \\\\
> \\text{Pred-score} \\downarrow & 0.037\\pm0.000 & 0.046\\pm0.001 & 0.043\\pm0.001& 0.044\\pm0.001 & 0.039\\pm0.000 & 0.055\\pm0.003 \\\\
> \\text{Marginal Score} \\downarrow  &  0.355 & 0.403 & 0.434 & 0.359 & 0.513 & 0.335 \\\\
> \\text{sym-Recall} \\uparrow &  0.594 & 0.500 & 0.413 & 0.787 & 0.206 & 0.000 \\\\
> \\hline
> \\end{array}$
>
>
> We will add a more extensive discussion, and new experiments in the final version of the paper.
>
>
> **References**
>
> [1] - Yoon, Jinsung, Daniel Jarrett, and Mihaela Van der Schaar. "Time-series generative adversarial networks." Advances in neural information processing systems 32 (2019).
>
>
> [2] - Jeon, Jinsung, et al. "GT-GAN: General Purpose Time Series Synthesis with Generative Adversarial Networks." Advances in Neural Information Processing Systems 35 (2022): 36999-37010.
>
> [3] - Alaa, Ahmed, Alex James Chan, and Mihaela van der Schaar. "Generative time-series modeling with fourier flows." International Conference on Learning Representations. 2020.
>
> [4] - Zhou, Linqi, et al. "Deep latent state space models for time-series generation." International Conference on Machine Learning. PMLR, 2023.
>
> [5] - Alaa, Ahmed, et al. "How faithful is your synthetic data? sample-level metrics for evaluating and auditing generative models." International Conference on Machine Learning. PMLR, 2022.
>
> [6] - Khayatkhoei, Mahyar, and Wael AbdAlmageed. "Emergent Asymmetry of Precision and Recall for Measuring Fidelity and Diversity of Generative Models in High Dimensions." arXiv preprint arXiv:2306.09618 (2023).

---

> ### Author Response · Authors · 2023-11-16
>
> ## **Minor Weaknesses**
>
> ### **MW1 /. Q.4.**
>
> We have now revised Section 3.1.1 to clarify and remove any misleading statements about SIREN outperforming Fourier Features Networks (FFNs) in terms of spectral bias. Following Q4, we will provide an ablation study shortly comparing SIREN and FFN and will add a more extensive discussion in the final version of the paper.
> **[UPDATE]** We have added the additional ablation studies in the supplementary material, and we discuss them in a comment below.
>
>
> ### **MW2**
>
> We acknowledge that Figure 4 is missing essential details and a more accurate discussion of the results. We also agree with the reviewer that the main goal of our analysis is showing a relatively low computational time for our approach w.r.t. existing methods, as also shown in Table 12. In re-writing the final paper, we are making sure of not claiming any independence with respect to the sequence length.
>
> **More details** Our experiments use the same synthetic data-set of 8192000 data-points (e.g. number of samples $\times$ sequence length), organized in 5 datasets of different sequence lengths $l \\in \\{80, 320, 1280, 5120, 20480\\}$. Thus, we obtain 5 datasets of $\\{102400, 25600, 6400, 1600, 400\\}$ number of samples, which we process using $\\{1024, 256, 64, 16, 4\\}$ as batch-size, respectively.  Thanks to this batch-size, we can fairly train each model for 100 iteration and always loading to GPU the same amount of data-points, for each sequence length and iteration.For inference, we only sample a single batch of data.
>
> **Impact of data loading** All the times are computed directly in the training and testing loop, after the model has been already loaded in the GPU memory, using a NVIDIA T4 GPU. Moreover, we found that the data loading (both for training and inference) is negligible w.r.t. model computational time. We measure the loading being around $ 0.0145$ sec for training (100 iterations) and inference $0.001$ sec (1 epoch).
>
> **Fourier Flow** Finally, we repeated the experiments with Fourier Flows with the new computational times reported in Table 2 of supplementary. However, we still have some convexing. We believe the slightly drop likely comes from not optimized code to handle large batch-sizes: longer time-series (e.g., 320 and 1280) worsen the computational time, but the model is also relieved by the smaller batch-size (e.g., 256 and 64).
>
> ### **MW3**
>
> We thank the reviewer for spotting this issue. We are improving the charts in the final version of the paper with the same scale for both the x and y axes. Figure 5 of the supplementary material already shows some of the new charts.
>
>
> ### **MW4**
>
> We apologize for the lack of clarity in the description of the method. We are incorporating a new detailed description of our time series generation procedure.
>
> ### **MW5 / W.3 / Q.3.**
> We thank the reviewer for the suggestion.
>
> **Reconstruction.** In the supplementary material we now report both the quantitative and qualitative metrics to assess the quality of the reconstruction in train and test set. In Table 5 of supplementary we report the reconstruction error for the iHyperTime for the Energy24, Stock24, Stock72, and Stock360 datasets. While Figure 6 and Figure 7 show the actually quality of reconstructed time-series for Stock 72 and Stock 360, respectively.
>
> **Latent dimension**
> The model results at varying of the latent dimensions $Z_T$, $Z_S$ and $Z_R$ are shown in the supplementary material on Table 3 and Table 4 for stock 24 and stock 72, respectively. As suggested we evaluate both the reconstruction error and the generation metrics, for different z\_{T}\_{S}\_{R} values (e.g., z\_4\_6\_8 represents $Z_T=4$, $Z_S=6$ and $Z_R=8$). The results show comparable results for Stock 72. For Stock 24 we have slightly better generative performance for small dimensions at the cost of higher reconstruction error, and vice versa.
>
> ## **Other Questions**
>
> ### **Q.5.**
>
> In Table 5 of supplementary we report quantitative metrics to evaluate the reconstruction quality using iHyperTime for  Energy24, Stock24, Stock72, and Stock360 datasets. Figure 6 and Figure 7 report the qualitative results, showing the reconstructed time-series for Stock 72 and Stock 360, respectively.
>
> ### **Q.6.**
> We apologize for not introducing the Marginal Score for all the datasets. For simplicity, for each dataset we decided to use only the most common metrics used in the related benchmark papers. Therefore, only Monash uses the marginal score[1]. In our current effort we are in integrating Marginal score also for the other datasets. Table 1 of supplementary material shows the marginal score results for Stock 24.
>
> [1] - Zhou, Linqi, et al. "Deep latent state space models for time-series generation." International Conference on Machine Learning. PMLR, 2023.

---

> ### Author Response · Authors · 2023-11-16
>
> ## **Other Questions**
>
> ### **Q.7.**
>
> **Temp Rain.** The Temp Rain dataset has more complex temporal dynamics, which make the generative process more difficult. In particular, Temp Rain data has stiff transitions: i.e., most of the data points lie around on the x-axis (y=0) with very sharp spikes [1]. Therefore, a model could achieve very low marginal score by just generating data closely to the x-axis. Such data however are also easily distinguishable from the real data, for a good predictive model.  Therefore, we believe LS4[2] generates good data, but with less sharp spikes than the real data, achieving better marginal score but lower Classification score w.r.t. our approach. This would also explain the high predictive score in LS4, where less sharp spikes would also hinder the predictive power of a model trained on synthetic data but tested on real.
>
>   We will add more detailed discussion in the final version of the paper.
>
> **Notation.**  Regarding the different notations for the metrics in Tables 1-4 and Table 5, in order to make a fair comparison with the state-of-the-art benchmarks, Tables 1-4 follow the predictive and discriminative scores as they are set up in TimeGAN, GT-GAN, DiffTime, etc, whilst in Table 5 we follow the same set up for Classification and predictive score as LS4. Moreover, in Table 5 we show results in the Monash dataset, where three of the datasets have time series with sequence length greater than 700. Given that the predictive and discriminative scores in Tables 1-4 are RNN-based, the time to compute the metrics would be impractical. In addition, the RNN-based metrics might not be able to capture the more complex dynamics from the Monash datasets.
>
>
> [1] - Temperature Rain Dataset without Missing Values, https://zenodo.org/records/5129091
>
> [2] - Zhou, Linqi, et al. "Deep latent state space models for time-series generation." International Conference on Machine Learning. PMLR, 2023.
>
>
> ### **Q.8. and Q.9.**
>
> We thank the reviewer for the good suggestion. We are actually training DiffTime for Monash, which we initially omitted due to the longer training times. We also agree that the performance seems to have an anomaly for 30 Stock missing data. We will add the new results soon, and we will integrate them in the final version of the paper.

---

> ### Author Response · Authors · 2023-11-18
>
> ### **Q.4**
>
> We appreciate the reviewer's suggestion for these important ablation studies. We have included both analyses in the supplementary material, and discuss them below. We will include these additional results in the final version of the paper.
>
> **Analysis of $w_0$.** In Section 1.7 of the supplementary material, we have included the performance of iHT using different values of $w_0$ in the SIREN layers. In the Tables A and B below, we report the reconstruction and generation metrics for the Stock24 and Stock72 datasets. In Stock24, the best reconstruction and generation metrics are obtained with $w_0=30$. In the case of Stock72, the best reconstruction metric is also obtained with $w_0=30$, but the discriminative and predictive scores show a slight improvement for $w_0=100$.
>
> _Table A: Results for iHT with different values of $w_0$ for the Stock24 dataset._
> $$
> \\begin{array}{cccccc}
> \\hline
>  &    w_0=5 &   w_0=10 &   w_0=30 &  w_0=100 &  w_0=300  \\\\
> \\hline
> \\text{MAE (train)} &  0.0119 &  0.0119 &  0.0069 &  0.0243 &  0.1851 \\\\
> \\text{MAE (test)}  &  0.0120 &  0.0121 &  0.0082 &  0.0245 &  0.1788 \\\\
> \\hline
> \\text{Discr-score} &  0.151 \\pm 0.059 &  0.212 \\pm 0.044 &  0.054 \\pm0.028 & 0.087 \\pm 0.066 &  0.489 \\pm 0.007 \\\\
> \\text{Pred-score}  &  0.041 \\pm 0.001 &   0.040 \\pm 0.001 &  0.037 \\pm 0.000  & 0.037 \\pm 0.000 &  0.059 \\pm 0.002 \\\\
> \\hline
> \\end{array}
> $$
>
>
> _Table B: Results for iHT with different values of $w_0$ for the Stock72 dataset._
> $$
> \\begin{array}{cccccc}
> \\hline
>  &    w_0=5 &   w_0=10 &   w_0=30 &  w_0=100 &  w_0=300  \\\\
> \\hline
> \\text{MAE (train)} &  0.0156 &  0.0115 &  0.0093 &  0.0182 &  0.2052  \\\\
> \\text{MAE (test)}  &  0.0157 &  0.0116 &  0.0096 &  0.0187 &  0.1963 \\\\
> \\hline
> \\text{Discr-score} &  0.045 \\pm 0.034 &  0.058 \\pm 0.038 & 0.028 \\pm 0.027 & 0.026 \\pm 0.011 &  0.250 \\pm 0.037 \\\\
> \\text{Pred-score}  &  0.188 \\pm 0.002 &  0.186 \\pm 0.001 & 0.185 \\pm 0.001 & 0.183 \\pm 0.001 &   0.196 \\pm 0.000  \\\\
> \\hline
> \\end{array}
> $$
>
> **Comparison with FFN.**  In Section 1.8 of the supplementary material, we now report the performance of a variant of iHT that uses Random Fourier Features (FFN) instead of SIREN for the representation of residuals. Given the importance of the scale factor in FFN as explained in [1], we evaluated FFN with multiple scale factor values: 5, 10, 100.
> The Tables C and D below show the reconstruction and generation metrics for the Stock24 and Stock72 datasets, respectively. For the Stock24 dataset, SIREN achieves the best reconstruction performance as well as the lowest discriminative score. In the case of Stock72, FFN with scale factor 10 achieves better reconstruction errors, and shows slightly better performance in discriminative score than SIREN network.
>
> _Table C: Results for with SIREN layers and with FFN for Stock24 dataset._
>
> $$
> \\begin{array}{ccccc}
> \\hline
>  &    \\text{SIREN} (w_0=30) & \\text{FFN} (\sigma=5) &  \\text{FFN} (\sigma=10) & \\text{FFN} (\sigma=100) \\\\
> \\hline
> \\text{MAE (train)} &  0.0069   &     0.0094 &        0.0095 &         0.0101   \\\\
> \\text{MAE (test)}  &  0.0082  &     0.0094 &        0.0095 &         0.0100 \\\\
> \\hline
> \\text{Discr-score} &  0.054 \\pm 0.028   &  0.244 \\pm 0.162 &  0.346 \\pm 0.169 &  0.36 \\pm 0.183 \\\\
> \\text{Pred-score}  &  0.037 \\pm 0.000  &    0.036 \\pm 0.000 &    0.036 \\pm 0.000 &   0.036 \\pm 0.000   \\\\
> \\hline
> \\end{array}
> $$
>
> _Table D: Results for with SIREN layers and with FFN for Stock72 dataset._
>
> $$
> \\begin{array}{ccccc}
> \\hline
>  &    \\text{SIREN} (w_0=30) & \\text{FFN} (\sigma=5) &  \\text{FFN} (\sigma=10) & \\text{FFN} (\sigma=100) \\\\
> \\hline
> \\text{MAE (train)} &  0.0093 &      0.0078 &        0.0085 &         0.0083   \\\\
> \\text{MAE (test)}  &  0.0096 &     0.0079 &        0.0086 &         0.0082 \\\\
> \\hline
> \\text{Discr-score} & 0.028 \\pm 0.027    &  0.028 \\pm 0.014 &  0.024 \\pm 0.026 &  0.054 \\pm 0.029 \\\\
> \\text{Pred-score}  &  0.185 \\pm 0.001  &  0.184 \\pm 0.001 &  0.185 \\pm 0.002 &  0.184 \\pm 0.001   \\\\
> \\hline
> \\end{array}
> $$
>
> [1] Woo, Gerald et al. “Learning Deep Time-index Models for Time Series Forecasting.” International Conference on Machine Learning (2022).

---

> > ### Comment · Reviewer_h6ka · 2023-11-20
> > **Thank you for your answers and details.**
> >
> > I would first like want to thank the authors for their thorough and honest responses about the different raised issues.
> >
> > **W1**. As illustrated in Figures 1 and 2, with the current generation approach, the generated time series is very similar to the closest one it is created from. This illustrates my main concern about the approach, that the generated time series are just small alterations of the training data.
> >
> > **W2**. Thank you for clarifying this point. This is critical as with the current generation mechanism, generated time series are very close to the training set. It is good also to have added a diversity metric.
> >
> > **MW1**. Thank you for the ablation study, and as also stated by reviewers y35Q and 3ERW, it is a good thing that the residual blocks paragraph be more clearly written.
> >
> > **MW2**. Thank you for the details.
> >
> > **MW5**. Thank you for the experiments. They confirm that the reconstruction is not of very good quality, especially for Stock 72. Also, I don't understand why the reconstruction scores for Stock 72 are better than for Stock 240, whereas the qualitative results show otherwise.
> >
> > The ablation on the dimension of the latent space is interesting because it illustrates the trade-off between reconstruction and generation. iHyperTime probably has good performances for generation because it underfits when learning to reconstruct. I am not convinced by this approach. I think that a well-trained INR with a more interesting generation mechanism would be more valuable. In general, a way to improve reconstruction would be to use something less prone to underfitting than the current set encoder.
> >
> > **Q7**. Thank you for the explanation. Regarding the notations, it would be important to have a small paragraph in the appendix to clarify this point as you just did.
> >
> > Overall, I think that this is a serious work with many experiments. However, I do not believe that the approach is sound enough, and the new experiments support this claim, with new generated time series being very close to the learned ones and the generative properties of the model probably being mostly due to the reconstruction error. Thus I will keep my score.

---

> > > ### Author Response · Authors · 2023-11-22
> > >
> > > We would like to thank the reviewer for their continued engagement with our work. Below we answer to the recent feedback:
> > >
> > > **W.1**
> > >
> > > Regarding the similarity of the generated time series, in our work we randomly sampled lambda values from the interval [0.0, 0.3]. This range was selected to ensure a balance between diversity and fidelity to the original data characteristics.     We believe that the chosen range strikes a good trade-off between realism and diversity. Quantitative and qualitative metrics show among the best fidelity and realism, and initial results in the supplementary show that we are second best in terms of diversity (Figure 5 and Table 1). Moreover, it's worth noting that the user can adjust lambda (e.g., 0.5), if more diversity is preferred.
> > >
> > > We would also like to provide additional context regarding the interpolation of embeddings. A key aspect of our model is the division of embeddings into three distinct components: trend, seasonality, and residuals. This representation allows for a more granular and targeted approach to time series generation, wherein each aspect of a time series is learned and encoded separately. This separation provides a unique advantage in terms of interpolation flexibility. For instance, we can interpolate between two time series while keeping one of the embeddings (e.g. trend) fixed.
> > > This gives place to a constrained generation process which maintains certain characteristics of the original series fixed (i.e. trend), while varying others (i.e. seasonality and residuals).We can selectively alter specific characteristics of the time series, which adds a layer of control to the generation process. We refer to Appendix H for a demonstration and analysis of this kind of constrained generation, which holds direct relevance across a broad spectrum of time series scenarios.
> > >
> > >
> > > **Q.7**
> > >
> > > We will include a short paragraph in the appendix in the final version of the paper to clarify the notation.

---

> > > > ### Author Response · Authors · 2023-11-22
> > > >
> > > > **MW5**
> > > >
> > > > **Reconstruction for Stock72 and Stock 360.** We thank the reviewer for spotting such discrepancies between qualitative and quantitative reconstruction. And we apologize for not having precisely report the qualitative metric. Specifically, for Stock 360 and Stock 72, we maintained different axis scales, which we believe could be misleading. For instance, a low-volatility time series fluctuating between 0 and 0.1 might inaccurately indicate a significant reconstruction error compared to a highly volatile time series defined between 0 and 1. We do apologize for any confusion caused by this oversight, we have updated such Figures (in Figure 12 and Figure 13 of supplementary).
> > > >
> > > >
> > > > **Reconstruction and Generation Performance.** While we agree that there is a trade-off between reconstruction and generation performance (especially in terms of diversity), we believe that our generative performance are also the result of a great effort in designing a good architecture, training procedure, and training loss with both time series amplitudes and spectrum (as detailed in Eqn 6) taken into account.
> > > >
> > > > **Implementation of Attentive Set Encoder.** In response to the concerns raised about underfitting with our original set encoder, we have implemented the attentive set encoder SeFT-attn [1] in our architecture. This encoder, as suggested, is known for its potential to provide a more nuanced and detailed representation of the input data. We show new results following this implementation in Section 1.10 of the supplementary material. In Figure 11 in the supplementary material we show the t-SNE visualization and the data distribution, for iHT with each set encoder. In the table below we show the evaluation of iHT using the original set encoder and SeFT-attn:
> > > >
> > > > $\\begin{array}{lrr}
> > > > \\hline
> > > > \\text{Metric} & \\text{iHT - Set Encoder} &  \\text{iHT - SeFT-attn} \\\\ \\hline
> > > > \\text{MAE (train)} &   0.0093 & 0.0118  \\\\
> > > > \\text{MAE (test)}  &  0.0096 & 0.0120 \\\\
> > > > \\text{Discr-score}  &   0.028\\pm0.027   & 0.038\\pm0.026  \\\\
> > > > \\text{Pred-score}    &  0.185\\pm0.001  & 0.187\\pm0.002  \\\\
> > > > \\hline
> > > > \\end{array}$
> > > >
> > > > We observe that iHT with the original set encoder shows slightly better performance both in reconstruction and generation metrics, although we believe this could be improved by further tuning of hyperparameters. We are currently in the process of performing extensive hyperparameter tuning to achieve better performance with the SeFT-attn encoder.
> > > >
> > > > **Generation Mechanism.** To achieve more diversity, we could extend our method by incorporating a convex combination of multiple embeddings, e.g., random sampling in the convex hull. This strategy would allow for a broader exploration within the embedding space, while maintaining a degree of control and consistency with the training data. This is a similar approach to what has been proposed in [2], where the authors generate new signals (in their case BRDFs) by encoding their data using PCA and interpolating in the convex hull of the reduced space.
> > > >
> > > > A different strategy beyond embedding interpolation could be to change our current model to an unconditional one. In Section 1.9 of the supplemental material, we have incorporated a discussion of a direct modification to our model, to adapt it towards an entirely generative (unconditional) approach based on the Variational Autoencoder (VAE) framework. We will incorporate this discussion as part of the appendix in the final version.
> > > >
> > > > [1] M. Horn et al. Set Functions for Time Series. ICML 2020.
> > > >
> > > > [2] Serrano, Ana et al. “An intuitive control space for material appearance.” ACM Transactions on Graphics (TOG) 2016.
> > > >
> > > > **Novelty**
> > > > Finally, we would like to highlight that our major contribution is the proposal of a novel framework for time-series generation with interpretable components.  In particular, we showed state-of-art generation results for missing data and very long time-series, which we believe is a limitation in many existing generation models. We really thank the reviewer for all suggestions, we also acknowledge the current limitations but we hope that this contribution could be of broader interested. For example, we agree that it could be extended for pure unconditional generation, or for adopting different set encoders.

---

> > > > > ### Comment · Reviewer_h6ka · 2023-11-23
> > > > >
> > > > > Thank you for your responses. I still believe that this work could be improved and that the approach is not sound enough to be accepted yet. More specifically, I think that the reconstruction error needs to be reduced (for instance with a better encoder) and that the generation mechanism could be more sound (for instance with the ideas that you propose here). Overall I appreciated our exchanges and I think that with these further improvements (that could not be done in the rebuttal due to the time constraint), this work would be valuable for the community.
> > > > >
> > > > > PS: Thank you for the clarification for the differences between Stock 72 and stock 360, it is clear now for me.
> > > > >
> > > > > PS2: You stated ‘in our work we randomly sampled lambda values from the interval [0.0, 0.3]’ but in the code there is ‘lambd = np.random.uniform(low=0.0, high=0.5)’, which mechanism is the one that you used in the additional experiments ?

---

> > > > > > ### Author Response · Authors · 2023-11-23
> > > > > >
> > > > > > Thank you for your continued engagement. We confirm that in our experiments we sampled lambda values from the interval [0.0, 0.3]. We have now updated the code in the repository with a more complete version of the code, including the implementation of the attentive set encoder (SeFT). As part of the new commit, we have updated the README.md and corrected the value of lambda to 0.3. The latest version of the code can be found in the folder "ihypertime_rebuttal". Thank you again for your consideration.

---

### Official Review · Reviewer_3NpH · 2023-11-10

**Soundness:** 3 good
**Presentation:** 3 good
**Contribution:** 3 good
**Rating:** 6
**Confidence:** 3

**Summary:**

The proposed work is concerned with time series generation and introduces an approach into encode the time series in the form of implicit neural representation through a TSNET,  a trend-seasonality-residual representation. The framework is applied to several regular and irregular time-series generation tasks with various percentages of missing data and compared with other approaches.

**Strengths:**

1. iHyperTime is able to synthesize time series and decompose the representation into configurable feature sets: trends- slow components of the signal, seasonality - periods of the signal and residuals.

2. Results show more accurate generation ability than compared GAN and other networks.

3. Training and inference times of ihyperTime appear to be more optimal than other compared approaches.

**Weaknesses:**

1. It is unclear which data is missing in experiments with irregular timeseries. Is this missing data in training or testing? The procedure of removing data needs to be defined.

2. It is unclear how the method performs in terms of clustering/classification score.

3. The results are not compared to GNN based generation methods.

**Questions:**

1. Recent works show that RNN encoder-decoder with various training strategies such as weak decoder or contrastive or attractive losses can both generate clustered interpretable latent representation and generate sequences. How these models compare with iHyperTime?

---

> ### Author Response · Authors · 2023-11-14
>
> We thank the reviewer for pointing out aspects that were not clearly explained and discussed in the paper.
>
> ### **Weaknesses:**
>
> **W1**
>
> Following existing literature [1,2,3] we create irregular time-series datasets by randomly dropping 30, 50, 70\% of observations from the original datasets. We select each observation to drop uniformly at random, and independently for each time series  {$(t_i, y_i)$}$_{i=0}^N$ in the original dataset.  The randomly removed data is the same for every model and every repeat, and used both in training and testing. We will add citations and more details for the irregular time-series dataset generation in the final version of the paper.
>
> **W2**
>
> We thank the reviewer for pointing out two interesting tasks that were not sufficiently discussed.  While our work primarily focuses on learning an interpretable trend-seasonality time series representation for time-series generation, we agree with the reviewer that this latent representation could be easily used and leveraged in broader applications, like cluster or classification. In particular, these applications could extensively benefit from the learned interpretable decomposition that separates trend, seasonality and residuals. We will discuss the possible different applications in the final version of the paper.
>
> **W3**
>
> We thank the reviewer for the suggestion. In our current effort, we are improving the state-of-art section discussing related GNN work [4,5,6,7]. Among this work we note that:
>
> - the work in [4] is the only one directly addressing the generation of time-series using a Graph-based model. Although contemporaneous, we plan to compare our proposed approach against this model, and we contacted the authors to access their code.
>
> - the work in [5] proposes an interesting graph neural network approach to embed irregularly sampled and multivariate time-series. While the work focuses on forecasting, it can be extended to other applications including generation. We are evaluating the complexity and effort of adapting such model for a comparison.
>
>
> ### **References:**
>
> [1] - Jeon, Jinsung, et al. "GT-GAN: General Purpose Time Series Synthesis with Generative Adversarial Networks." Advances in Neural Information Processing Systems 35 (2022): 36999-37010.
>
> [2] - Kidger, Patrick, et al. "Neural controlled differential equations for irregular time series." Advances in Neural Information Processing Systems 33 (2020): 6696-6707.
>
> [3] - Tang, Xianfeng, et al. "Joint modeling of local and global temporal dynamics for multivariate time series forecasting with missing values." Proceedings of the AAAI Conference on Artificial Intelligence. Vol. 34. No. 04. 2020.
>
> [4] - Iyer, Srikrishna, and Teng Teck Hou. "GAT-GAN: A Graph-Attention-based Time-Series Generative Adversarial Network." arXiv preprint arXiv:2306.01999 (2023).
>
> [5] -Zhang, Xiang, et al. "Graph-Guided Network for Irregularly Sampled Multivariate Time Series." International Conference on Learning Representations. 2022.
>
> [6] - Andrea, Cini, Marisca Ivan, and Cesare Alippi. "Filling the Gaps: Multivariate Time Series Imputation by Graph Neural Networks." ICLR 2022. 2021. 1-20.
>
> [7] - Jin, Ming, et al. "A survey on graph neural networks for time series: Forecasting, classification, imputation, and anomaly detection." arXiv preprint arXiv:2307.03759 (2023).

---

> > ### Author Response · Authors · 2023-11-14
> >
> > ### **Questions**
> >
> > **Q1**
> >
> > We thank the reviewer for suggesting an interesting discussion on RNN-based architectures for intepretable latent representations and sequence generation. In our current effort we are improving the state-of-art review to better discuss RNN-based approaches against iHyperTime.
> >
> > To the best of our knowledge, classical RNN approaches have shown lower performance in generative tasks (e.g., against GAN-based approaches like TimeGAN [1] used in our comparison). Even when they are trained using teacher-forcing (T-Forcing) [2] as well as professor-forcing (P-Forcing) [3] to regularize the generation. In fact, autoregressive approaches condition the model using previously generated samples, and even small prediction errors can compound over the sequences.  Therefore, we believe that a first major difference w.r.t. iHyperTime is that our approach can generate realistic time-series for longer sequences over 700 time steps.
> >
> > As suggested by reviewer we are also surveying approaches using contrastive loss, attractive losses, and weak decoder which have shown to improve performance compared with other losses [4, 5]. We will discuss the main differences between our intepretable latent representation for trend, seasonality, and residuals, against clustered interpretable latent representation [4, 6, 7, 8].
> >
> > Finally, to the best of our knowledge, existing work do not target directly time-series representation for generation. Instead, our approach achieves state-of-art performance in such generation task with also an interpretable time-series representation. We are happy to include any useful references the reviewer may suggest.
> >
> > [1] - Yoon, Jinsung, Daniel Jarrett, and Mihaela Van der Schaar. "Time-series generative adversarial networks." Advances in neural information processing systems 32 (2019).
> >
> > [2] - Alex Graves. "Generating sequences with recurrent neural networks." arXiv preprint arXiv:1308.0850, 2013.
> >
> > [3] - Alex M Lamb, et al. "Professor forcing: A new algorithm for training recurrent networks." In Advances In Neural Information Processing Systems, pages 4601–4609, 2016.
> >
> > [4] - Wanyan, Tingyi, et al. "Contrastive learning improves critical event prediction in COVID-19 patients." Patterns 2.12 (2021).
> >
> > [5] - Woo, Gerald, et al. "CoST: Contrastive learning of disentangled seasonal-trend representations for time series forecasting." arXiv preprint arXiv:2202.01575 (2022).
> >
> > [6] - Zhong, Ying, Dong Huang, and Chang-Dong Wang. "Deep Temporal Contrastive Clustering." Neural Processing Letters (2023): 1-17.
> >
> > [7] - Franceschi, Jean-Yves, Aymeric Dieuleveut, and Martin Jaggi. "Unsupervised scalable representation learning for multivariate time series." Advances in neural information processing systems 32 (2019).
> >
> > [8] - Eldele, Emadeldeen, et al. "Time-series representation learning via temporal and contextual contrasting." arXiv preprint arXiv:2106.14112 (2021).

---

> ### Comment · Reviewer_3NpH · 2023-11-21
> **Thank you for clarifications**
>
> I would like to thank the authors for clarifying the weaknesses and questions that I have raised. While W1 & W2 have been addressed by the authors, W3 and Q1 remain to be a concern since comparison appears to lack important methods/baselines.

---

> > ### Author Response · Authors · 2023-11-21
> >
> > We thank the reviewer for the time spent on reviewing our manuscript, and we are pleased to hear that our efforts to address W1 and W2 have been useful. Regarding W3 and Q1, we are making our best to integrate the requested approaches for the final version of the paper.

---

### Author Response · Authors · 2023-11-21

We would like to thank all the reviewers for their constructive and valuable comments, which helped us to substantially improve our work. We also thank the reviewers for recognizing such effort, and the improvements we made.

For convenience, we summarize here the main changes we introduced:

$\\bullet$ We have revised the entire Section 3, to enhance the presentation of our contribution, clarifying how the generative process works. We have also included more implementation details on the Set Encoder and the Decoder in the Appendix D.2.

$\\bullet$ We are improving the state-of-art comparison. We are now splitting the dataset into training and test set, and we are also evaluating the diversity of time-series both qualitatively and quantitatively, measured by a _sym-Recall_ metric [1].

$\\bullet$ We experimentally evaluate the quality of the reconstruction for our method, using both quantitative and qualitative metrics.

$\\bullet$ We conducted an ablation study to evaluate the quality of the reconstruction and generation w.r.t. $\\omega_0$ in SIREN layers, for both Stock72 dataset and Stock24 dataset.

$\\bullet$ We have introduced an experimental study to evaluate the model at varying of the latent dimensions $Z_T$ , $Z_S$ and $Z_R$. Our evaluation illustrates the trade-off between reconstruction and generation quality.

$\\bullet$ We conducted an ablation study on iHT using Random Fourier Features (FFN) instead of a SIREN network for the residuals. We show that SIREN achieves the best reconstruction performance as well as the lowest discriminative score.

$\\bullet$ We introduced an experiment to qualitatively evaluate the reconstructed time-series from interpolations between pairs of embeddings. We show a smooth reconstruction of time-series as the embeddings are interpolated using two latent vectors z1 and z2 at varying $\\lambda \\in [0, \\ldots, 1]$.

$\\bullet$ We demonstrate how our method allows interpretable decomposition on three real-data datasets, namely Atmospheric CO2, Airline Passenger, and Stock72. The first two datasets exhibit strong trend and seasonality, while the latter only exhibits a strong trend.

We will incorporate all the rebuttal's answers and the new supplementary material in the final version of the paper.

**Novelty:**
Finally, we would like to highlight that our approach provides realistic time-series generation with interpretable trend, seasonality, and residuals. With respect to state-of-art approaches, we demonstrated that our generation does not just provide interpretable components, but we also are superior in terms of realism, computational time, and usefulness.  Additionally, our method's inherent flexibility effectively addresses both irregular and very long time series, demonstrating its potential for a wider range of scenarios.

[1] - Khayatkhoei, Mahyar, and Wael AbdAlmageed. "Emergent Asymmetry of Precision and Recall for Measuring Fidelity and Diversity of Generative Models in High Dimensions."  In International Conference on Machine Learning (ICML 2023)

---

### Meta-Review · Area_Chair_61gS · 2023-12-13

**Metareview:**

The paper addresses the challenge of generating realistic time series. The proposed system comprises three key components: a set encoder that generates three latent representations from a sequence of tuples (t, x(t)), encoded using implicit Neural representation networks. Subsequently, hypernetworks utilize these latent representations as input to predict the weights of three modules dedicated to modeling the trend, seasonal, and residual components of the series, respectively. The outputs from these three modules are then combined to forecast the time series. After training the system on real data, it becomes capable of generating synthetic series by sampling in the latent space. This is accomplished through linear interpolation between two latent representations learned from real series. The paper presents extensive experiments and comparisons with baselines, demonstrating the strong performance of the proposed generation model.

The reviewers identified several issues related to the clarity of the technical description and the experimental setting. In response, the authors offered comprehensive explanations to address the concerns and augmented the manuscript with additional experiments and analyses, as per the reviewers' suggestions. These extensive responses and modifications greatly enhance the manuscript. Following the rebuttal, two out of four reviewers raised their scores. However, despite acknowledging the authors' diligent efforts, the remaining two reviewers asserted that significant technical concerns persisted. One main problem highlighted by all the reviewers is that the model exclusively performs conditional time series generation while most baselines operate in an unconditional setting. Consequently, the objectives of the generation tasks are somewhat divergent. As emphasized by the reviewers, this conditional setting introduces bias in the performance evaluation metrics, favoring the proposed method over the baselines. Although this limitation is now acknowledged in the discussion of the proposed method, concerns persist regarding the fairness of the comparisons.

**Justification For Why Not Higher Score:**

Some problems remaining in the comparison with baselines

**Justification For Why Not Lower Score:**

a

---

### Decision · Program_Chairs · 2024-01-16

Reject